# RoME: A Robust Mixed-Effects Bandit Algorithm for Optimizing Mobile Health Interventions

**Easton Huch**
Department of Statistics
University of Michigan
Ann Arbor, MI, USA
`ekhuch@umich.edu`

**Jieru Shi**
Statistical Laboratory, DPMMS
University of Cambridge
Cambridge, England, UK
`js2882@cam.ac.uk`

**Madeline R. Abbott**
Department of Biostatistics
University of Michigan
Ann Arbor, MI, USA
`mrabbott@umich.edu`

**Jessica R. Golbus**
Michigan Medicine
Ann Arbor, MI, USA
`jgolbus@umich.edu`

**Alexander Moreno**
Independent Researcher[*]
`alexander.f.moreno@gmail.com`

**Walter H. Dempsey**
Department of Biostatistics
University of Michigan
Ann Arbor, MI, USA
`wdem@umich.edu`

## Abstract

Mobile health leverages personalized and contextually tailored interventions optimized through bandit and reinforcement learning algorithms. In practice, however, challenges such as participant heterogeneity, nonstationarity, and nonlinear relationships hinder algorithm performance. We propose RoME, a **Ro**bust **M**ixed-**E**ffects contextual bandit algorithm that simultaneously addresses these challenges via (1) modeling the differential reward with user- and time-specific random effects, (2) network cohesion penalties, and (3) debiased machine learning for flexible estimation of baseline rewards. We establish a high-probability regret bound that depends solely on the dimension of the differential-reward model, enabling us to achieve robust regret bounds even when the baseline reward is highly complex. We demonstrate the superior performance of the RoME algorithm in a simulation and two off-policy evaluation studies.

## 1 Introduction

Mobile health (mHealth) uses smart devices to deliver digital notification interventions to users. These notifications nudge users toward healthier attitudes and behaviors. Because mHealth applications can monitor and react to users and their environment, they offer the promise of personalized, contextually tailored interventions. In practice, achieving this promise requires bandit or reinforcement learning (RL) algorithms that can accurately learn mHealth intervention effects, including how they vary by user, over time, and based on context. In this regard, contextual bandit algorithms are appealing due to strong empirical performance in other settings, their ability to customize intervention decisions based on changing contexts, and their simplicity and extensibility relative to full RL algorithms.

One unique aspect of mHealth (Greenewald et al., 2017) is the presence of a "do-nothing" action in which the mHealth app does not intervene. As mHealth rewards typically involve human decision making (e.g., whether the user chooses to exercise), the distribution of the corresponding *baseline reward*—the reward under the do-nothing action—typically evolves in a complex, nonstationary fashion as time progresses and contexts change. In contrast, the *treatment effects*—the difference in the expected value of the reward relative to the do-nothing action—tend to be much more stable

---

[*]A portion of this work was performed while Alexander was employed at Luminous Computing.

and can be adequately modeled using classical stationary models, such as linear models (Robinson, 1988).

Greenewald et al. (2017) introduced an action-centered contextual bandit algorithm with sublinear regret in this setting by replacing the observed reward, $R_t$, with a "pseudo-reward," $\tilde{R}_t$, where $t$ indexes the time points. Denoting the binary action as $A_t \in \{0, 1\}$ with $A_t = 0$ corresponding to the do-nothing action, the pseudo-reward for $A_t$ can be written as

$$\tilde{R}_t := (A_t - \pi_t)R_t = \pi_t(1 - \pi_t)R_t \left( \frac{A_t}{\pi_t} - \frac{1 - A_t}{1 - \pi_t} \right),$$

where $\pi_t$ is the (known) probability of treatment assignment from the Thompson-sampling algorithm. We took an additional step to derive an equivalent expression (second equality), which reveals that $\tilde{R}_t$ is proportional to an inverse-probability-weighted (IPW) estimator. Crucially, $\tilde{R}_t$ provides an unbiased estimate of the treatment effect *regardless* of the nonstationarity of the baseline outcome.

Although Greenewald et al. (2017)'s analysis shows sublinear regret in nonstationary settings, our simulations show that other methods can achieve lower regret over finite time horizons. The reasons include its failure to address treatment effect heterogeneity, pool information across users, and model the baseline outcome, which leads to highly variable pseudo-rewards and slow learning. We propose to generalize their method to overcome these limitations by

1. imposing hierarchical structure in the treatment effect model with shared fixed effects and random effects for users and time (i.e., a mixed-effects model),

2. efficiently pooling across users and time via nearest-neighbor regularization (NNR), and

3. denoising rewards with flexible supervised learning models via the debiased machine learning (DML) framework of Chernozhukov et al. (2018).

We name the resulting method RoME to highlight that it is a **Ro**bust **M**ixed-**E**ffects algorithm. The primary contributions are (a) the RoME method, (b) a high-probability regret bound that relies solely on the dimension of the differential-reward model, which is typically much smaller than that of the complex baseline reward model, and (c) empirical comparisons demonstrating the superior performance of RoME in simulation and two real-world mHealth studies.

## 2 Related Work

Related work in statistics considers the estimation of treatment effects with longitudinal data. Cho et al. (2017) present a semiparametric random-effects method for estimating subject-specific *static* treatment effects. Qian et al. (2020) discusses the statistical challenges of applying linear mixed-effects models to mHealth data and shows that the resulting estimates may lack a causal interpretation.

In the bandit literature, several works address the challenge of heterogeneous users. Ma et al. (2011) introduced NNR in recommender systems using homophily principles—similar nodes are more likely to be connected than dissimilar ones (McPherson et al., 2001). Cesa-Bianchi et al. (2013) adapted NNR for bandit settings, improving regret. Subsequent improvements focused on scalability and algorithm modifications for stronger regret bounds (Vaswani et al., 2017; Yang et al., 2020).

Other bandit approaches explicitly address the longitudinal setting in which treatment effects may evolve over time. The intelligentpooling method of Tomkins et al. (2021) employs a Gaussian linear mixed-effects model with both user- and time-specific random effects. Hu et al. (2021) provides a related approach based on generalized linear mixed effects models. Unlike our approach, both methods require the (generalized) linear conditional mean model to be correctly specified.

Aouali et al. (2023) and Lee et al. (2024) also propose bandit algorithms with mixed effects; methodologically, however, their use of mixed effects differs from the above approaches and our own. Aouali et al. (2023) employs random effects to relate treatments within categories (e.g., movies within a genre), and Lee et al. (2024) considers the case in which a single, discrete action leads to a multivariate reward that is affected by a user-specific random effect. In contrast, the other approaches listed above (and our own) do not assume any relationships among the actions, and they associate each discrete action with a single scalar-valued reward in an iterative fashion.

Other work develops semiparametric methods that lead to sublinear regret without requiring correct specification of the conditional mean reward. Krishnamurthy et al. (2018) and Kim & Paik (2019) propose improvements to the approach of Greenewald et al. (2017) that lead to stronger regret bounds. Kim et al. (2021, 2023) develop a doubly robust approach for both linear and generalized linear contextual bandits. However, in contrast to Tomkins et al. (2021), Hu et al. (2021), and our approach, these semiparametric methods are not immediately applicable to longitudinal settings.

## 3 Setting

### 3.1 Problem Statement

Recognizing the connection between mHealth policy learning and contextual bandit methods, we now consider a contextual multi-armed bandit environment with one control arm (the do-nothing action) denoted by $a = 0$ and $q$ non-baseline arms corresponding to different actions or treatments. Users are indexed by $i = 1, 2, \ldots$ and decision points are indexed by $t = 1, 2, \ldots$. For each user $i$ at time $t$, the algorithm observes a context vector $S_{i,t} \in \mathcal{S}$, chooses an action $A_{i,t} \in [q] \coloneqq \{0, \ldots, q\}$, and receives a reward $R_{i,t} \in \mathbb{R}$. The *differential reward* describes the difference in expected rewards between choosing a non-baseline arm and the control arm, defined as $\Delta_{i,t}(s, a) \coloneqq \mathbb{E}[R_{i,t}|S_{i,t} = s, A_{i,t} = a] - \mathbb{E}[R_{i,t}|S_{i,t} = s, A_{i,t} = 0]$.

We denote the observation history up to time $t$ for user $i$ as $\mathcal{H}_{i,t} \coloneqq (S_{i1}, A_{i1}, R_{i1}, \ldots, S_{i,t})$. Actions are selected according to stochastic policies, $\pi_{i,t} : \mathcal{H}_{i,t} \times \mathcal{S} \to \mathcal{P}([q])$. We use $\pi_{i,t}(a|s)$ to denote the probability of action $a \in [q]$ given context $s \in \mathcal{S}$ for a fixed (implicit) history. As in Greenewald et al. (2017), we assume that the probability of the control action is bounded:

**Assumption 1.** *There exists $0 < \pi_{\min}, \pi_{\max} < 1$ such that $\pi_{\min} < \pi_{i,t}(0|a) < \pi_{\max}$ for all $i, t, a$.*

As noted in Greenewald et al. (2017), this assignment probability controls the number of messages that participants receive. It ensures that participants receive sufficient messages that they do not disengage, but not so many that they are overwhelmed or fatigued. We now define

$$A_{i,t}^\star = \underset{a \in [q]}{\operatorname{argmax}} \, \mathbb{E}[R_{i,t}|S_{i,t}, A_{i,t} = a], \quad \bar{A}_{i,t}^\star = \underset{a \in [q] \setminus \{0\}}{\operatorname{argmax}} \, \mathbb{E}[R_{i,t}|S_{i,t}, A_{i,t} = a],$$

the optimal arm (including control) and the optimal non-baseline arm, respectively. Respecting the restrictions in Assumption 1, we can now express the optimal policy, $\pi_{i,t}^\star$, in two cases:

1. $A_{i,t}^\star = 0$: $\pi_{i,t}^\star(0|S_{i,t}) = \pi_{\max}$ and $\pi_{i,t}^\star(\bar{A}_{i,t}^\star|S_{i,t}) = 1 - \pi_{\max}$.
2. $A_{i,t}^\star > 0$: $\pi_{i,t}^\star(0|S_{i,t}) = \pi_{\min}$ and $\pi_{i,t}^\star(\bar{A}_{i,t}^\star|S_{i,t}) = \pi_{i,t}^\star(A_{i,t}^\star|S_{i,t}) = 1 - \pi_{\min}$.

These cases represent the appropriate boundary condition given the optimal action. We maximize the probability of the control action if $A_{i,t}^\star = 0$ and minimize it if $A_{i,t}^\star \neq 0$, selecting the next-best action with the highest possible probability; we set the assignment probabilities for the remaining arms to zero. In Section 5.2, we define regret relative to this optimal policy, $\pi_{i,t}^\star$.

Lastly, we consider a study design that progresses in *stages*, where users enter the study sequentially (Friedman et al., 2010), as illustrated in Figure 1. Staged recruitment serves two purposes in our analysis: it increases the fidelity of the theoretical setting by mimicking real-world recruitment strategies, and it allows us to estimate changes in the differential reward over time. Concretely, we first observe user $i = 1$ at time $t = 1$. Then we observe users $i = (1, 2)$ at times $t = (2, 1)$. Subsequent stages $k$ involve users $i \leq k$, each having had $k - i + 1$ observations, respectively. Define $\mathcal{O}_k = \{(i, t) : i \leq k, \, t \leq k + 1 - i\}$ as the set of all observations in stage $k$. Appendix F provides a notation summary.

### 3.2 Doubly Robust Differential Reward

Following Greenewald et al. (2017), we assume that the differential reward is linear: $\Delta_{i,t}(s, a) = x(s, a)^\top \theta_{i,t}$, where $x(s, a) \in \mathbb{R}^d$ is a feature vector depending on the context and action. We allow the baseline reward, $g_t(s)$, to be a nonlinear and nonstationary function of context and time. Combined, these two assumptions imply a partially linear model for the conditional mean reward:

$$\mathbb{E}[R_{i,t}|S_{i,t} = s, A_{i,t} = a] = g_t(s) + x(s, a)^\top \theta_{i,t} \delta_{a>0},$$

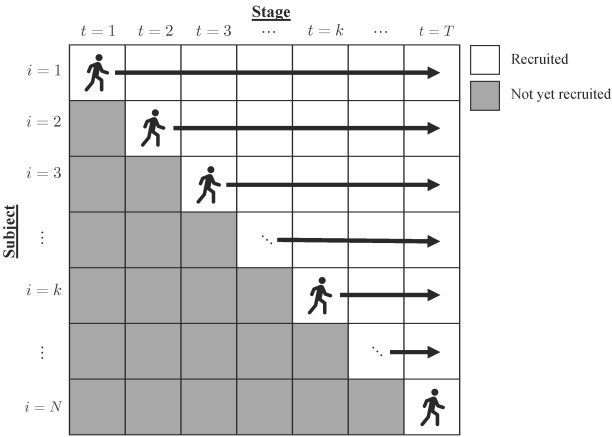

Figure 1: Illustration of the staged recruitment scheme. At each recruitment stage (each time point), a new participant is recruited and observed. At the same time, all participants who were recruited prior to the current stage are also observed again. Observations are not collected from participants who have yet to be recruited. For simplicity, we assume one participant is recruited at each stage.

where $\delta_{a>0}$ is an indicator of a non-baseline action. By incorporating both the linear predictor $x(s,a)^\top \theta_{i,t} \delta_{a>0}$ and the nonlinear baseline $g_t(s)$, we can maintain interpretability of the action effects while also allowing flexibility in specifying the observed rewards. The individual rewards are then realized according to the model

$$R_{i,t} = g_t(s) + x(s,a)^\top \theta_{i,t} \delta_{a>0} + \epsilon_{i,t},$$

where we make the following standard assumption (Abeille & Lazaric, 2017) regarding $\epsilon_{i,t}$:

**Assumption 2.** *The error $\epsilon_{i,t}$ is conditionally mean zero (i.e., $\mathbb{E}[\epsilon_{i,t}|\mathcal{H}_{i,t}] = 0$) and sub-Gaussian.*

Assumption 2 ensures that large errors are sufficiently rare and that $\tilde{R}_t$ is proportional to an unbiased estimate of $\Delta_{i,t}(s,a)$. However, $\tilde{R}_t$ suffers from high variance, which leads to suboptimal performance in practice. Our proposed algorithm, RoME, relies on an improved estimator, $\tilde{R}_{i,t}^f(s,\bar{a})$, that is also unbiased but achieves lower variance by denoising the rewards according to a working model, $f_{i,t}(s,a)$, for the conditional mean $r_{i,t}(s,a) := \mathbb{E}[R_{i,t}|S_{i,t} = s, A_{i,t} = a]$:

$$\tilde{R}_{i,t}^f(s,\bar{a}) := \underbrace{f_{i,t}(s,\bar{a}) - f_{i,t}(s,0)}_{\text{Model prediction}} + \underbrace{\frac{R_{i,t} - f_{i,t}(s, A_{i,t})}{\delta_{A_{i,t}=\bar{a}} - \pi_{i,t}(0|s)}}_{\text{Debiasing term}}, \tag{1}$$

where $\bar{a}$ is a preselected non-baseline arm; i.e., we condition on $A_{i,t} \in \{0, \bar{a}\}$. The first component is a model-based prediction of the differential reward, and the second component is a debiasing term that uses IPW to correct for potential errors in the specification of $f$. This improved estimator is a form of pseudo-outcome, a common technique in the causal inference literature (Bang & Robins, 2005; Nie & Wager, 2021; Kennedy, 2023). We refer to it as a *Doubly Robust Differential Reward* because it produces an unbiased estimate of $\Delta_{i,t}(s,a)$ as long as either $\pi_{i,t}$ or $f_{i,t}$ is correctly specified (see proof in Appendix E.1); though, in practice, the benefit stems from (a) the robustness to misspecification of $f$ (because $\pi_{i,t}$ is known) and (b) the variance reduction introduced by denoising the outcomes (see Remark 1 in Appendix E.2). In the following, we abbreviate $\tilde{R}_{i,t}^f(s,a)$ as $\tilde{R}_{i,t}^f$, with the context and action implied.

## 4 Algorithm Components

We now construct $\tilde{R}_{i,t}^f$, introduce the weighted least-squares loss function used by RoME to estimate parameters, and use nearest-neighbor regularization to pool information across users.

## 4.1 Debiased Machine Learning

To construct $\tilde{R}_{i,t}^f$, we fit a working model $f_{i,t}$ for the conditional mean reward. Because $\tilde{R}_{i,t}^f$ is robust to misspecification of $f_{i,t}$, our regret bound in Theorem 1 does not require $f_{i,t}$ to be correctly specified. However, we show in Appendix E.2 that the following assumption results in a lower asymptotic variance bound for $\tilde{R}_{i,t}^f$.

**Assumption 3.** *The working model, $f_{i,t}$, satisfies $|r_{i,t}(s,a) - f_{i,t}(s,a)| \overset{a.s.}{\to} 0$ for all $s$, $a$.*

Assumption 3 states that $f_{i,t}$ is a strongly consistent estimator of $r_{i,t}$, but it does not impose any specific rate requirement on the convergence. Although our regret bound does not require Assumption 3, it does require two standard conditions within the DML framework: (a) Neyman orthogonality and (b) independence of $f_{i,t}$ and the observed rewards. The construction of $\tilde{R}_{i,t}^f$ ensures that (a) holds, making $\tilde{R}_{i,t}^f$ robust to misspecification of $f$. We can enforce (b) via sample-splitting techniques. We present two such options below.

**Option 1:** The first option exploits the fact that outcomes across different users are independent. Consequently, we can perform sample splitting as follows. **Step 1**: Randomly assign each user $i$ to one of $J$ folds. Let $W_j$ denote the set containing user indexes for the $j$-th fold and let $W_j^{\complement}$ denote its complement. **Step 2**: For each fold, use a supervised learning algorithm to estimate the working model for $r_{i,t}(s,a)$ denoted $\hat{f}_{i,t}^{(j)}(s,a)$ using $W_j^{\complement}$. **Step 3**: Construct the pseudo-outcomes using (1).

**Option 2:** Because we do not expect an adversarial environment in mHealth, we may be willing to assume that the errors, $\epsilon_{i,t}$, are independent across time. In this case, we can adapt the procedure given above by randomly assigning data to folds at the level of $(i,t)$. In this case, $W_j$, contains the $(i,t)$ pairs assigned to fold $j$. This option is more powerful than option 1 in that it enables us to learn heterogeneity across users in the nuisance function, $f_{i,t}$, leading to greater variance reduction.

Lastly, we note two strategies that can reduce the computational demands of sample splitting. The first is to employ estimators that satisfy a leave-one-out stability condition, such as bagged estimators that use subsampling (Chen et al., 2022); these estimators eliminate the need to perform sample splitting. The second is to fit $f_{i,t}$ in an online fashion.

## 4.2 Estimation via Weighted Least Squares

Having formed $\tilde{R}_{i,t}^f(s,\bar{a})$, we now explain how to estimate the parameters, $\theta_{i,t} \in \mathbb{R}^d$, determining the differential reward, $\Delta_{i,t}(s,a)$. We assume that $\theta_{i,t}$ consists of three components:

**Assumption 4.** $\theta_{i,t} = \theta^{shared} + \theta_i^{user} + \theta_t^{time}$ *for all $i,t$.*

The parameter $\theta^{\text{shared}}$ is a fixed effect shared across all $i,t$. In contrast, $\theta_i^{\text{user}}$ and $\theta_t^{\text{time}}$ are random effects specific to a given user and time point, respectively. We place these parameters in a single column vector, $\theta$, as $\text{vec}(\theta^{\text{shared}}, \theta_1^{\text{user}}, \ldots, \theta_K^{\text{user}}, \theta_1^{\text{time}}, \ldots, \theta_K^{\text{time}})$, where $K$ is the total number of stages. We then form a corresponding feature vector, $\phi_{i,t} = \phi(x_{i,t}) \in \mathbb{R}^{(2K+1)d}$, where $\phi$ inserts $x_{i,t}$ into the positions corresponding to $\theta^{\text{shared}}$, $\theta_i^{\text{user}}$, and $\theta_t^{\text{time}}$ with zeros in all other locations; i.e., $\phi(x_{i,t}) = C_{i,t}^\top x_{i,t}$, where

$$C_{i,t} := [1, \, 0_{i-1}^\top, \, 1, \, 0_{K-i+t-1}^\top, \, 1, \, 0_{K-t}^\top] \otimes I_d$$

so that $C_{i,t} \in \mathbb{R}^{d \times (2K+1)d}$. We could then estimate $\theta_{i,t}$ via a weighted least squares (WLS) regression that minimizes the following objective function:

$$\ell_{\text{WLS}}(\theta) = \sum_{i=1}^n \sum_{t=1}^{K-i+1} \tilde{\sigma}_{i,t}^2 \big( \tilde{R}_{i,t}^f - \phi_{i,t}^\top \theta \big)^2,$$

where $\tilde{\sigma}_{i,t}^2 = \pi_{i,t}(0|S_{i,t}) \cdot \{1 - \pi_{i,t}(0|S_{i,t})\}$. However, $\ell_{\text{WLS}}(\theta)$ is over-parameterized and does not take advantage of available network information. The next section modifies $\ell_{\text{WLS}}(\theta)$ by adding regularization, producing the final loss function used in RoME.

## 4.3 Nearest-Neighbor Regularization

We assume access to network information relating neighboring users and time points. These networks are defined by graphs $G_{\text{user}} = (V_{\text{user}}, E_{\text{user}})$ and $G_{\text{time}} = (V_{\text{time}}, E_{\text{time}})$, where $V$ denotes vertices and $E$, edges. In practice, these networks can be formed via known clusters (e.g., shared schools, companies, geographic locations) or similar covariates. In the case of $G_{\text{time}}$, a reasonable choice is to construct a graph of neighboring sequential time points; i.e., $t = 1 \leftrightarrow t = 2 \leftrightarrow t = 3 \ldots$. Because we expect neighboring users and time points to have similar parameters, we may be able to improve our estimates of $\theta$ by regularizing neighboring values of $\theta_i^{\text{user}}$ and $\theta_t^{\text{time}}$ toward each other. This can be accomplished by using an $L^2$ Laplacian penalty. We illustrate the idea using $G_{\text{user}}$; the application to $G_{\text{time}}$ is similar.

First, we form the incidence matrix, $Q$, with the entry $Q_{v,e}$ corresponding to the $v$-th vertex (user) and the $e$-th edge. Denote the vertices of this edge as $v_i$ and $v_j$ with $i > j$. $Q_{v,e}$ is then equal to 1 if $v = v_i$, -1 if $v = v_j$, and 0 otherwise. The Laplacian matrix is then defined as $L = QQ^\top \in \mathbb{R}^{K \times K}$. Similar to Yang et al. (2020), we form a *network cohesion* penalty across users as follows:

$$\text{tr}(\Theta_{\text{user}}^\top L_{\text{user}} \Theta_{\text{user}}) = \sum_{(i,j) \in E_{\text{user}}} \|\theta_i^{\text{user}} - \theta_j^{\text{user}}\|_2^2,$$

where $\Theta_{\text{user}} := (\theta_1^{\text{user}}, \ldots, \theta_K^{\text{user}})^\top \in \mathbb{R}^{K \times d}$ and $L_{\text{user}}$ is the Laplacian matrix for users. The penalty is small when $\theta_i^{\text{user}}$ and $\theta_j^{\text{user}}$ are close for connected users. We employ a shared regularization hyperparameter $\lambda$, for $G_{\text{user}}$ and $G_{\text{time}}$, and include a standard $L^2$ penalty on $\theta$ with hyperparameter $\gamma$. The full penalization matrix $V_0 \in \mathbb{R}^{(2K+1)d \times (2K+1)d}$ is

$$V_0 = \text{diag}(\gamma I_d, \gamma I_{Kd} + \lambda L_\otimes^{\text{user}}, \gamma I_{Kd} + \lambda L_\otimes^{\text{time}}), \tag{2}$$

where $L_\otimes^{\text{user}} := L_{\text{user}} \otimes I_d$, the Kronecker product of $L_{\text{user}}$ with $I_d$ ($L_\otimes^{\text{time}}$ is defined similarly). We then adapt $\ell_{\text{WLS}}(\theta)$ to include a corresponding penalty term:

$$\ell(\theta) = \sum_{i=1}^{n} \sum_{t=1}^{K-i+1} \tilde{\sigma}_{i,t}^2 \big(\tilde{R}_{i,t}^f - \phi_{i,t}^\top \theta\big)^2 + \theta^\top V_0 \theta. \tag{3}$$

In Section 5, we show that (3) leads to a Thompson sampling algorithm for action selection.

## 5   Thompson Sampling Algorithm

We now describe the Thompson sampling procedure and provide a high-probability regret bound.

### 5.1   Algorithm Description

The minimizer of the weighted regularized least squares loss in (3) is $\hat{\theta} = V^{-1}b$, where

$$V = V_0 + \sum_{i=1}^{n} \sum_{t=1}^{K-i+1} \tilde{\sigma}_{i,t}^2 \phi_{i,t} \phi_{i,t}^\top, \quad b = \sum_{i=1}^{n} \sum_{t=1}^{K-i+1} \tilde{R}_{i,t}^f \tilde{\sigma}_{i,t}^2 \phi_{i,t}.$$

In analogy to Bayesian linear regression, the penalized and weighted Gram matrix, $V$, plays the role of a (scaled) posterior precision matrix under a multivariate Gaussian prior for $\theta$. This motivates an algorithm in which we randomly perturb the point estimate, $\hat{\theta}$. Specifically, we define $\underline{V}_{i,t}^{-1} := C_{i,t} V^{-1} C_{i,t}^\top$ and set $\tilde{\theta}_{i,t} = C_{i,t} \hat{\theta} + \beta_{i,t}(\delta) \underline{V}_{i,t}^{-1/2} \eta$, where $\eta$ is a mean-zero random variable and

$$\beta_{i,t}(\delta) := v \sqrt{2 \log \left\{ \frac{2K(K+1)}{\delta} \right\} + \log \left\{ \frac{\det(\underline{V}_{i,t}^{-1})}{\det(\Lambda_{i,t}^0)} \right\}} + \zeta \max\{\log^{3/4}(K), 1\}. \tag{4}$$

In the above expression, $\Lambda_{i,t}^0 := C_{i,t} V^{-1} V_0 V^{-1} C_{i,t}^\top$ and $K$ is the total number of stages. The value $v$ is the sub-Gaussian factor for $\tilde{R}_{i,t}^f$, and $\zeta$ is defined in Remark 2 in Appendix E; both can be set to sufficiently large constants. The hyperparameter $\delta$ determines the probability with which the regret bound holds. The random deviate, $\eta$, is drawn from a distribution $\mathcal{D}^{TS}$ satisfying the

technical conditions in Definition 1 in Appendix D. In practice, we recommend setting $\mathcal{D}^{TS}$ to a multivariate Gaussian distribution or t-distribution to simplify calculations. Given $\tilde{\theta}_{i,t}$, we then compute $\bar{A}_{i,t} = \operatorname{argmax}_{a \in [q] \backslash \{0\}} x_{i,t}^\top \tilde{\theta}_{i,t}$ and randomly select $A_{i,t} = 0$ with probability

$$\pi_{i,t}(0|\bar{A}_{i,t}) := \max \left[ \pi_{\min}, \min \left\{ \pi_{\max}, \Pr(x_{i,t}^\top \tilde{\theta}_{i,t} < 0) \right\} \right], \tag{5}$$

or $A_{i,t} = \bar{A}_{i,t}$ with probability $1 - \pi_{i,t}(0|\bar{A}_{i,t})$. The action selection is summarized in Algorithm 1.

---

**Algorithm 1** Action selection

---

**input** $V, b, C_{i,t}, \beta_{i,t}(\delta), x_{i,t}, \pi_{\min}, \pi_{\max}$
    Set $\hat{\theta} = V^{-1}b$ and $\underline{V}_{i,t}^{-1} = C_{i,t}V^{-1}C_{i,t}^\top$
    Sample $\eta \sim \mathcal{D}^{TS}$
    Set $\tilde{\theta}_{i,t} = C_{i,t}\hat{\theta} + \beta_{i,t}(\delta)\underline{V}_{i,t}^{-1/2}\eta$
    Set $\bar{A}_{i,t} = \operatorname{argmax}_{a \in [q] \backslash \{0\}} x_{i,t}^\top \tilde{\theta}_{i,t}$
    Calculate $\pi_{i,t}(0|\bar{A}_{i,t})$ according to (5)
    With probability $\pi_{i,t}(0|\bar{A}_{i,t})$, play action 0; otherwise, play action $\bar{A}_{i,t}$

---

Having detailed the action selection algorithm, we now summarize RoME in Algorithm 2. The algorithm includes an outer loop that iterates through the stages from $k = 1$ to $k = K$. Within each stage, we handle users sequentially, iteratively selecting actions according to Algorithm 1, observing rewards, and updating $V$ and $b$ accordingly.

---

**Algorithm 2** RoME

---

    Initialize $V_0$ according to (2)
    $V \leftarrow V_0$
    $b \leftarrow 0$
    **for** $k = 1, \ldots, K$ **do**
        **for** $(i, t) = (1, k), (2, k-1), \ldots, (k, 1)$ **do**
            Choose an arm according to Algorithm 1 using $V$ and $b$
            Observe reward, $R_{i,t}$
            Construct pseudo-reward, $\tilde{R}_{i,t}^f$, as explained in Section 4.1
            Set $\phi_{i,t} = C_{i,t}^\top x_{i,t}$
            Update weighted Gram matrix: $V \leftarrow V + \tilde{\sigma}_{i,t}^2 \phi_{i,t}\phi_{i,t}^\top$
            Update feature-outcome products: $b \leftarrow b + \tilde{\sigma}_{i,t}^2 \tilde{R}_{i,t}^f \phi_{i,t}$
        **end for**
    **end for**

---

### 5.2 Regret bound

We first define regret, $\mathbf{Regret}_K$, in terms of the following average across stages:

$$\sum_{k=1}^{K} \frac{1}{k} \sum_{(i,t) \in \mathcal{O}_k \backslash \mathcal{O}_{k-1}} \left\{ \pi_{i,t}^\star(\bar{A}_{i,t}^\star|S_{i,t}) \cdot x(S_{i,t}, \bar{A}_{i,t}^\star)^\top \theta_{i,t}^\star - \pi_{i,t}(\bar{A}_{i,t}|S_{i,t}) \cdot x(S_{i,t}, \bar{A}_{i,t})^\top \theta_{i,t}^\star \right\}.$$

The inner sum calculates regret (the difference between the expected reward of the optimal action and the chosen action) weighted by the probability of the action, following Greenewald et al. (2017), for all users at stage $k$. We then average these values to account for the growing number of pairs $(i, t)$ as the stages progress. The cumulative regret is the sum of these averaged values over all stages.

The rewards associated with the baseline action are excluded since $\Delta_{i,t}(S_{i,t}, 0) = 0$. $\mathbf{Regret}_K$ is a form of pseudo-regret because it eliminates the randomness due to $\{\epsilon_{i,t}\}_{t=1}^{K}$ (Audibert et al., 2003). The regret bound requires additional technical assumptions:

**Assumption 5.** *For all s and a, $\|x(s, a)\| \leq 1$.*

**Assumption 6.** *There exists a known value $M \in \mathbb{R}^+$ such that $|g_t(s)|, |f_{i,t}| \leq M$ for all i, t, and s.*

**Assumption 7.** *Let $e_K := \#E_K^{user} + \#E_K^{time}$ denote the total number of edges. Then $e_K = O(K^\omega)$ for some $\omega \in [0, \infty)$.*

**Assumption 8.** *Let $\theta_k^\star$ represent the vector of true parameter values at stage $k = i + t - 1$ and $V_{i,t}^{-1}$, the matrix $V$ prior to decision point $(i, t)$. Then $\sup_{(i,t)\in\mathcal{O}_K} \|V_0\theta_k^\star\|_{V_{i,t}^{-1}} = O_p\left\{\log^{3/4}(K)\right\}.$*

**Assumption 9.** *The matrix $\underline{V}_{i,t}$ is bounded below as follows: $\underline{V}_{i,t} \succ \alpha \min(i,t) I_d$ for some $\alpha > 0$ that may depend on $d$.*

Assumption 5 is a standard assumption that simplifies the proof (Abeille & Lazaric, 2017). Assumption 6 ensures that the (estimated) baseline rewards are bounded. Assumption 7 ensures that the number of edges grows no faster than a polynomial rate. Assumption 8 limits the contribution of the random effects to the regret bound. Lastly, Assumption 9 ensures sufficient exploration of the context space. See the proof of Theorem 1 in Appendix E.3 for additional discusion of Assumptions 8 and 9. We now state the regret bound.

**Theorem 1.** *Under Assumptions 1–2, 4–9, with probability at least $1 - \delta$, $\mathbf{Regret}_K$ is of order*

$$O\left[d\sqrt{K\log(K)\log(K^2d)}\left\{\sqrt{d} + \log^{1/4}(K)\right\}\right].$$

The exact bound (including constants) and proof are given in Appendix E. Following Greenewald et al. (2017), we decompose the regret into two terms. The first term depends on a summation of terms of the form $\{\pi_{i,t}^\star(A_{i,t}^\star|S_{i,t}) - \pi_{i,t}(\bar{A}_{i,t}|S_{i,t})\}x(S_{i,t}, \bar{A}_{i,t})^\top\theta_{i,t}^\star$. We employ a novel technique based on Markov's inequality to bound this term (see Lemmas 12 and 13 in Appendix E).

We bound the second term using the high-level proof technique of Abeille & Lazaric (2017). The primary difficulty in bounding this term is that the classical theory for the estimation error of RLS estimates (Theorem 2 in Abbasi-Yadkori et al. (2011)) is not directly applicable to our setting due to the increasing dimensionality. Lemma 6 in Appendix E generalizes this theory to our setting. This Lemma produces a bound involving $\|C_{i,t}V_{i,t}^{-1}V_0\theta_k^\star\|_{\underline{V}_{i,t}}$, which we subsequently bound by $\|V_0\theta_k^\star\|_{V_{i,t}^{-1}}$ in Lemma 7. This section of the proof requires Assumption 8 to ensure that the contribution of the new parameters to this bound is small enough that we incur no more than a logarithmic penalty.

The overall bound scales like $\tilde{O}(\sqrt{K}d^{3/2})$ up to log factors, matching the standard rate given in Agrawal & Goyal (2012) and Abeille & Lazaric (2017) for linear Thompson sampling with fixed dimension. In particular, the bound depends solely on the dimensionality of the differential reward— not the baseline reward, which may be much more complex.

# 6 Experiments

This section presents results from applying RoME in a simulated mHealth study with staggered recruitment and an off-policy comparison study for cardiac rehabilitation mHealth interventions. The simulations were implemented using Python and the results were generated using individual compute nodes with two 3.0 GHz Intel Xeon Gold 6154 processors and 180 GB of RAM. Case studies were implemented using R 4.2.2 and results were generated on a cluster composed of individual compute nodes with 2.10 GHz Intel Xeon Gold 6230 processors and 192 GB of RAM.

## 6.1 Competitor Comparison Simulation

In this section, we compare RoME to four competing methods in simulation. We implemented RoME using **Option 2** with a bagged ensemble of stochastic gradient trees (Gouk et al., 2019; Mastelini et al., 2021) trained online via the River library (Montiel et al., 2021). We programmed the linear algebra operations using SuiteSparse (Davis & Hu, 2011) to take advantage of the sparse nature of the Laplacian matrices and $\phi_{i,t}$ from Algorithm 2.

Our competing baselines are (a) Standard: Standard Thompson sampling for linear contextual bandits, (b) AC: the **A**ction-**C**entered contextual bandit algorithm (Greenewald et al., 2017), (c) IntelPooling: The intelligentpooling method of Tomkins et al. (2021) fixing the variance parameters close to their true values, and (d) Neural-Linear: a method that uses a pre-trained neural network to transform the feature space for the baseline reward (similar to the Neural Linear method of Riquelme et al. (2018)).

We compare these methods under three settings: Homogeneous Users, Heterogeneous Users, and Nonlinear. The first two involve a linear baseline model and time-homogeneous parameters, with the second setting having distinct user parameters. The third setting, designed to mirror the challenges of mHealth studies, includes a nonlinear baseline and both user- and time-specific parameters. For each setting, we simulate 200 stages following the staged recruitment regime depicted in Figure 1, and we repeat the full 200-stage simulation 50 times. Appendix A.1 provides details on the setup and a link to our implementation.

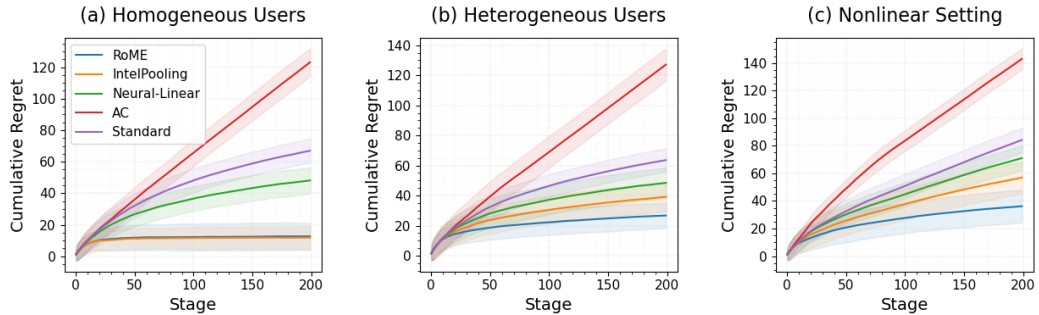

Figure 2: Cumulative regret in the (a) Homogeneous Users, (b) Heterogeneous Users, and (c) Nonlinear settings. RoME performs competitively in the first setting (the simplest), and it substantially outperforms the next-best method (IntelPooling) in the others.

Figure 2 displays the average cumulative regret for each method. RoME and IntelPooling have similar performance in the Homogeneous Users setting, but RoME excels in the Heterogeneous Users and Nonlinear settings, with significantly lower cumulative regret by stage 200. This is expected as RoME can utilize network information and model nonlinear baselines in these settings. However, in the Homogeneous Users setting, where no network information is available and a linear baseline is used, RoME does not outperform IntelPooling.

Appendices A.2–A.5 offer more results, including additional method comparisons, a rectangular data array simulation, hyperparameter sensitivity analyses, and pairwise statistical comparisons. The latter show that RoME outperformed each competitor in at least 48 of 50 repetitions in the Nonlinear setting, indicating even higher statistical confidence than Figure 2 suggests. RoME substantially outperforms the competing algorithms in the additional comparisons and is several orders of magnitude faster than would be required in a standard mHealth study, producing over 20,000 decisions in as little as 95 seconds (see Table 2 in Appendix A.2).

## 6.2 Valentine Study Analysis Results

In this section, we compare RoME to the above algorithms via off-policy evaluation using data from the Valentine Study (Jeganathan et al., 2022), a prospective, randomized-controlled, remotely administered trial designed to evaluate an mHealth intervention to supplement cardiac rehabilitation for low- and moderate-risk patients. In the analyzed dataset, participants were randomized to receive or not receive contextually tailored notifications promoting low-level physical activity and exercise throughout the day. The left panel of Figure 3 shows the estimated improvement in the average reward over the original constant randomization, averaged over stages (K = 120) and participants (N=108). We see that RoME achieved the highest average reward; its average performance (across bootstrap replications) is higher than the 75th percentile of all other methods.

To further analyze the improvement of RoME relative to the other methods, we test whether the proposed algorithm significantly improves cumulative rewards using paired t-tests with one-sided alternative hypotheses. The null hypothesis ($H_0$) for these tests is that the two algorithms being compared achieve the same average reward. The alternative hypothesis ($H_1$) is that the algorithm listed in the column achieves higher average rewards than the algorithm listed in the row. The right of Figure 3 displays the p-values obtained from these pairwise t-tests. The dark shade of the last column indicates that the proposed RoME algorithm achieves significantly higher rewards than the other five competing algorithms ($p \leq 0.01$ for all pairwise comparisons). The results imply that RoME would have achieved a 3.5% increase in step count relative to the constant randomization policy that was

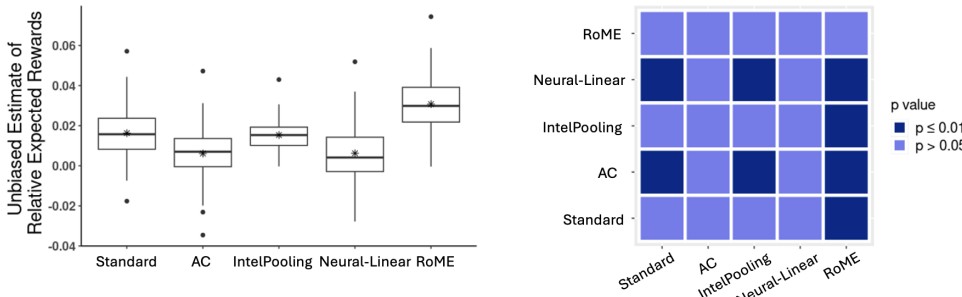

Figure 3: **(left)** Boxplot of unbiased estimates of the average per-trial reward for all five competing algorithms, relative to the reward obtained under the pre-specified Valentine randomization policy across 100 bootstrap samples. Within each box, the asterisk ($*$) indicates the mean value, while the mid-bar represents the median. **(right)** Heatmap of p-values from the pairwise paired t-tests.

actually implemented in this study. This effect translates to an increase of 140 steps per day, given a baseline of 1,000 steps per hour and four one-hour measurement windows. See Appendix B for additional implementation details.

We performed an additional off-policy comparison using data from the Intern Health Study (IHS) (NeCamp et al., 2020), as shown in Figure 11 in Appendix C.2. The results further demonstrate the competitive performance of the RoME algorithm, with the AC algorithm also showing comparable performance in this dataset. Further details on the analysis can be found in Appendix C.

## 7   Discussion

This paper introduces RoME, a robust mixed-effects contextual bandit algorithm for mHealth studies. RoME adapts DML and network cohesion penalties to dynamic settings, enabling researchers to efficiently pool information across users and over time. In addition to the methodological contribution, we also prove a high-probability regret bound in an asymptotic regime that involves a growing pool of users and time points. Our implementation of RoME is several orders of magnitude faster than would be required in practice and could easily be adapted for use in future mHealth studies.

We see several promising directions for improving RoME in future work. One such direction might consider improvements in the algorithm and theoretical analysis that enable us to replace Assumptions 8 and 9 with weaker and more primitive assumptions. In particular, these assumptions may not be plausible for $q > 1$ because most arm selections will be restricted to the same two arms (for large $K$). Other interesting directions include data-adaptive strategies for hyperparameter selection and addressing long-term impacts of mHealth treatments, such as accumulating treatment fatigue. Future work could also consider computational improvements that would enable large-scale deployment of RoME, which would be especially relevant in nonclinical settings.

**Acknowledgements**

The authors gratefully acknowledge the contributions of the researchers, administrators, and participants involved in the Valentine (`https://clinicaltrials.gov/study/NCT04587882`) and Intern Health (`https://clinicaltrials.gov/study/NCT03972293`) studies. This work was supported in part by grant P50 DA054039 provided through the NIH and NIDDK and by grant R01 GM152549 through the NIH and NIGMS. Dr. Golbus receives funding from the NIH (L30HL143700, 1K23HL168220). Initially while working on this paper, Dr. Moreno was supported by Luminous Computing, Inc.

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

# A   Additional Details for Simulation Study

This appendix provides details regarding the setup of our simulation study, additional results not presented in the main paper, and results from additional simulations employing different competitors and/or simulation designs. We summarize the differences between the competing algorithms in Table 1.

| | User-specific Parameters | Time-specific parameters | Pools Across Users | Enforces User Network Cohesion | Enforces Time Network Cohesion | Allows Nonlinear Baseline | Nonlinear Baseline Model |
|---|---|---|---|---|---|---|---|
| RoME | $\checkmark$ | $\checkmark$ | $\checkmark$ | $\checkmark$ | $\checkmark$ | $\checkmark$ | $\checkmark$ |
| RoME-BLM | $\checkmark$ | $\checkmark$ | $\checkmark$ | $\checkmark$ | $\checkmark$ | $\checkmark$ | $\times$ |
| RoME-SU | $\times$ | $\checkmark$ | $\checkmark$ | N/A | $\checkmark$ | $\checkmark$ | $\checkmark$ |
| NNR-Linear | $\checkmark$ | $\times$ | $\checkmark$ | $\checkmark$ | N/A | $\times$ | $\times$ |
| IntelPooling | $\checkmark$ | $\checkmark$ | $\checkmark$ | $\checkmark$ | $\times$ | $\times$ | $\times$ |
| Neural-Linear | $\checkmark$ | $\times$ | $\times$ | N/A | N/A | $\checkmark$ | $\checkmark$ |
| Standard | $\checkmark$ | $\times$ | $\times$ | N/A | N/A | $\times$ | $\times$ |
| AC | $\checkmark$ | $\times$ | $\times$ | N/A | N/A | $\checkmark$ | $\times$ |

Table 1: This table compares the methods tested in the simulation study based on the design components listed in the columns, with $\checkmark$ indicating that the method includes the component listed in the column, $\times$ indicating that it does not, and N/A indicating that the component is not applicable. This table can be used to investigate the effect of certain model components on cumulative regret; for example, comparing RoME and RoME-BLM helps us understand the impact of modeling the nonlinear baseline—the only column that differs between these two methods.

## A.1   Setup Details

The code for the simulation study is fully containerized and publicly available at `https://github.com/eastonhuch/RoME`. The simulation study involves a generative model of the following form for user $i$ at time $t$:

$$R_{it} = g(S_{i,t}) + x(S_{i,t}, A_{i,t})^\top \theta_{it} + \epsilon_{it}, \quad \epsilon_{it} \sim \mathcal{N}(0,1)$$

Here $S_{i,t} = (s_1, s_2) \in \mathbb{R}^2$ is a context vector, with both dimensions $\overset{iid}{\sim} U(-1,1)$. We set $x(s,a) = a(1, s_1, s_2)$. For simplicity, we set $g$ to a time-homogeneous function. The specific nature of the function varies across the following three settings mentioned in Section 6.1:

- Homogeneous Users: Standard contextual bandit assumptions with a linear baseline and no user- or time-specific parameters. The linear baseline is $g(S_{i,t}) = 2 - 2s_1 + 3s_2$, and the causal parameter is $\theta_{it} = (1, 0.5, -4)$ such that the optimal action varies across the context space.

- Heterogeneous Users: Same as the above but each user's causal parameter has iid $\mathcal{N}(0,1)$ noise added to it.

- Nonlinear: The general setting discussed in the paper with a nonlinear baseline, user-specific parameters, and time-specific parameters. The base causal parameter and user-specific parameters are the same as in the previous two settings. The nonlinear baseline and time-specific parameter are shown in Figure 4.

We assume that the data are observed via a staged recruitment scheme, as illustrated in Figure 1. For computational convenience, we update parameters and select actions in batches. If, for instance, we observe twenty users at a given stage, we update our estimates of the relevant causal parameters and select actions for all twenty users simultaneously. This strategy offers a slight computational advantage with limited implications in terms of statistical performance.

For simplicity, we assume that the nearest-neighbor network is known and set the relevant hyperparameters accordingly. We took care to set the hyperparameters such that RoME performs a similar

amount of shrinkage compared to other methods, especially IntelPooling, which effectively uses a separate penalty matrix for users and time. To accomplish this, we used the same penalty matrices for RoME and IntelPooling, effectively generalizing the scalar $\gamma$ penalty discussed in the main paper. We use 5 neighbors within the DML methods and set the other hyperparameters as follows: $\lambda = 1$, $\delta = 0.01$, $v = 1$, and $\zeta = 10$.

For the Neural-Linear method, we generate a $200 \times 200$ array of baseline rewards to train the neural network prior to running the bandit algorithm. Consequently, the results shown in the paper for Neural-Linear are better than would be observed in practice because we allowed the Neural-Linear method to leverage data that we did not make available to the other methods. This setup offers the computational benefit of not needing to update the neural network within bandit replications, which substantially reduces the necessary computation time.

Aside from the input features used, the Neural-Linear method has the same implementation as the Standard method. The Neural-Linear method uses the output from the last hidden layer of a neural network to model the baseline reward. However, we use the original features (the state vectors) to model the advantage function because the true advantage function is, in fact, linear in these features.

Our neural networks consisted of four hidden layers with 10, 20, 20, and 10 nodes, respectively. The first two employ the ReLU activation function (Nair & Hinton, 2010) while the latter two employ the hyperbolic tangent. We chose to use the hyperbolic tangent for the last two layers because Snoek et al. (2015) found that smooth activation functions such as the hyperbolic tangent were advantageous in their neural bandit algorithm. The loss function was the mean squared error between the neural networks' output and the baseline reward on a simulated data set. We trained our networks using the Adam optimizer (Kingma & Ba, 2014) with batch sizes of 200 for between 20 and 50 epochs. We simulated a separate validation data set to check that our model had converged and was generating accurate predictions.

Figure 5 compares the true baseline reward function in the nonlinear setting (left) to that estimated by the neural network (right). We see that the neural network produced an accurate approximation of the baseline reward, which helps explain the good performance of the Neural-Linear method relative to other baseline approaches, such as Standard.

We include the Neural-Linear method primarily to demonstrate that correctly modeling the baseline is not sufficient to ensure good performance. In the mHealth, algorithms should also be able to (a) efficiently pool data across users and time and (b) leverage network information. The Neural-Linear method satisfies neither of these criteria. Note that a neural network could be used to model the baseline rewards as part of our algorithm. Future work could consider allowing the differential rewards themselves to also be complex nonlinear functions, which could be accomplished by combining our method with Neural-Linear. We leave the details to future work.

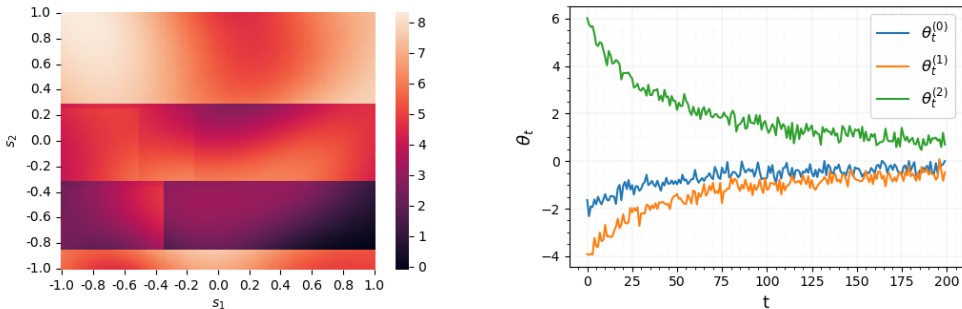

Figure 4: (left) The baseline reward function $g_t(S_{i,t})$ (constant across time in this case) used in the simulation study. The proposed method allows this function to be a nonlinear function of the context vectors. The baseline was generated using a combination of recursive partitioning and by summing scaled, shifted, and rotated Gaussian densities. (right) The time-specific parameters used in the simulation study. These parameters cause the advantage function to vary over time. We set them such that the advantage function changes quickly at the beginning of the study then stabilizes.

## A.2 Additional Method Comparisons

Our results in the main paper indicate that RoME outperforms existing methods, especially in the complex longitudinal setting that we are targeting. These positive results beg the question: What aspects of RoME contribute most to its superior performance?

In this section, we describe an additional simulation study designed to answer this question. In it, we compare RoME (as implemented in the main paper) to three additional comparison methods: RoME-BLM, RoME-SU, and NNR-Linear. Each comparison isolates particular aspects of our algorithm to understand its contribution to RoME's performance.

RoME-**BLM** is a version of RoME that employs **B**agged **L**inear **M**odels as the baseline supervised learning algorithm. Unlike the method implemented in the main paper (which uses bagged stochastic gradient trees), this baseline model is misspecified in the nonlinear setting; i.e., it cannot accurately model the nonlinear baseline reward function. Consequently, this comparison will allow us to gauge the impact of the supervised learning method used in our method. In particular, it should allow us to validate whether the algorithm is robust to baseline misspecification as the theory suggests.

RoME-**SU** is a **S**ingle-**U**ser version of our algorithm. Rather than having a separate $\theta_i$ for each user, this method estimates a single advantage function that is shared across all users. It does, however, employ DML, time effects, and nearest-neighbor regularization (for the time effects). Consequently, this comparison will help us understand the impact of the user-specific parameters in our simulation.

NNR-Linear is similar to the existing network-cohesion bandit algorithms described in the main paper. It includes a distinct $\theta_i$ for each user and regularizes them toward each other using a Laplacian penalty. It does not, however, use DML or time effects. As a result, this comparison isolates the impact of these two factors and should provide evidence that RoME is more broadly applicable than existing network-cohesion bandit algorithms.

Figure 6 displays the cumulative regret as a function of the stage for these methods across the three settings used in the main paper. We see that all methods perform similarly in panel (a), the setting with homogeneous users. However, in panel (b) we see that the regret achieved by RoME-SU begins to resemble a straight upward-sloping line because it cannot appropriately model the user-specific advantage functions.

In panel (c), we observe a much wider spread of performance, with RoME and RoME-BLM performing similarly and substantially outperforming the other methods. The fact that RoME and RoME-BLM perform similarly indicates that our algorithm is, in fact, robust to misspecification as expected. The comparison to NNR-Linear reveals that the DML and time effects—the new components of our method compared to existing network-cohesion bandit algorithms—do contribute meaningfully to RoME's superior performance. We found in additional simulations studies not reported here that the gap between NNR-Linear and our method widens as the magnitude of the time effects increases.

Table 2 displays the average computation time for each method across all three settings. Each run of the simulation produces $200 \times 201/2 = 20,100$ decisions. RoME and RoME-SU require the longest computation time at about seven minutes. RoME-BLM requires only 95–110 seconds, indicating

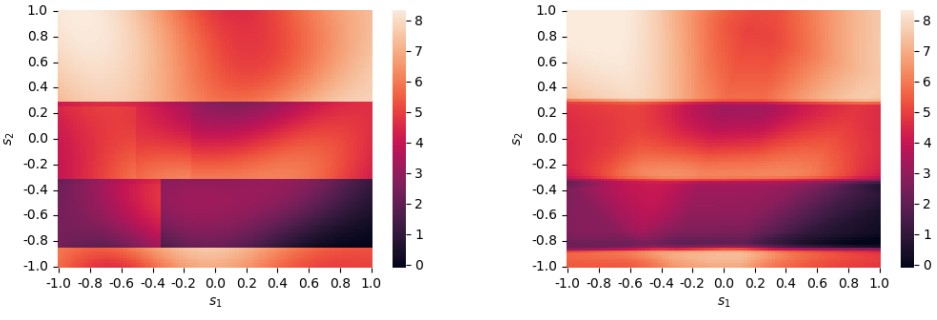

Figure 5: (left) The baseline reward function $g_t(S_{i,t})$ used in the simulation study compared to (right) the estimated baseline reward from our neural network in the nonlinear setting.

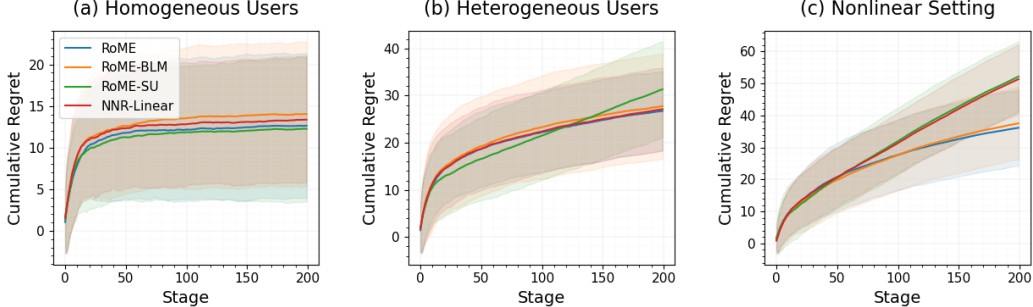

Figure 6: Cumulative regret in the (a) Homogeneous Users, (b) Heterogeneous Users, and (c) Nonlinear settings for the additional comparison methods. The fact that RoME outperforms RoME-SU demonstrates the importance of the user-specific parameters. The fact that RoME outperforms NNR-Linear shows that previous network cohesion approaches do not adequately address the nonlinear nature of the baseline reward and presence of time effects. The comparable performance of RoME-BLM to RoME highlights how our algorithm is robust to misspecification of the baseline reward model.

|  | Homogeneous Users | Heterogeneous Users | Nonlinear |
| --- | --- | --- | --- |
| RoME | 394.5 | 406.4 | 420.1 |
| RoME-BLM | 94.6 | 96.8 | 109.9 |
| RoME-SU | 383.5 | 390.5 | 396.5 |
| NNR-Linear | 29.7 | 31.3 | 31.5 |
| IntelPooling | 46.6 | 42.1 | 45.0 |
| Neural-Linear | 0.7 | 0.8 | 0.9 |
| Standard | 0.2 | 0.2 | 0.3 |
| AC | 0.1 | 0.1 | 0.3 |

Table 2: Average computation time across all methods and settings in the main simulation.

that the computational bottleneck for RoME is the ML model fitting. The required computation time for IntelPooling is about half that of RoME-BLM at 42–47 seconds. NNR-Linear requires 30–32 seconds, and the remaining methods run in less than one second. Because most clinical mHealth studies require fewer than 1,000 decisions per day, the speed of RoME is orders of magnitude faster than is required in practice. However, large-scale commercial deployment of RoME may require computational adjustments and approximations.

### A.3 Pairwise Comparisons

Table 3 shows pairwise comparisons between methods across the three settings. The individual cells indicate the percentage of repetitions (out of 50) in which the method listed in the row outperformed the method listed in the column. The asterisks indicate p-values below 0.05 from paired two-sided t-tests on the differences in final regret. The Avg column indicates the average pairwise win percentage.

RoME and RoME-BLM perform well across all three settings and the difference between them is statistically indistinguishable. These methods perform about equally well compared to IntelPooling and NNR-Linear in the first setting, but they perform better in the other two. In the Nonlinear setting in particular, RoME and RoME-BLM substantially outperform all other methods, and the pairwise differences are statistically significant at the 5% level.

### A.4 Simulation with Rectangular Data Array

The main simulation involves simulating data from a triangular data array. At the 200-th (final) stage, the algorithm has observed 200 rewards for user 1, 199 rewards for user 2, and so on.

## Homogeneous Users

|  | 1 | 2 | 3 | 4 | 5 | 6 | 7 | 8 | **Avg** |
|---|---|---|---|---|---|---|---|---|---|
| 1. RoME | - | 58%* | 46% | 66% | 40% | 100%* | 100%* | 100%* | 73% |
| 2. RoME-BLM | 42%* | - | 36%* | 44% | 32%* | 100%* | 100%* | 100%* | 65% |
| 3. RoME-SU | 54% | 64%* | - | 58% | 44% | 100%* | 100%* | 100%* | 74% |
| 4. NNR-Linear | 34% | 56% | 42% | - | 32%* | 100%* | 100%* | 100%* | 66% |
| 5. IntelPooling | 60% | 68%* | 56% | 68%* | - | 100%* | 100%* | 100%* | 79% |
| 6. Neural-Linear | 0%* | 0%* | 0%* | 0%* | 0%* | - | 100%* | 100%* | 29% |
| 7. Standard | 0%* | 0%* | 0%* | 0%* | 0%* | 0%* | - | 100%* | 14% |
| 8. AC | 0%* | 0%* | 0%* | 0%* | 0%* | 0%* | 0%* | - | 0% |

## Heterogeneous Users

|  | 1 | 2 | 3 | 4 | 5 | 6 | 7 | 8 | **Avg** |
|---|---|---|---|---|---|---|---|---|---|
| 1. RoME | - | 58% | 80%* | 56% | 100%* | 100%* | 100%* | 100%* | 85% |
| 2. RoME-BLM | 42% | - | 72%* | 52% | 100%* | 100%* | 100%* | 100%* | 81% |
| 3. RoME-SU | 20%* | 28%* | - | 22%* | 92%* | 100%* | 100%* | 100%* | 66% |
| 4. NNR-Linear | 44% | 48% | 78%* | - | 100%* | 100%* | 100%* | 100%* | 81% |
| 5. IntelPooling | 0%* | 0%* | 8%* | 0%* | - | 94%* | 100%* | 100%* | 43% |
| 6. Neural-Linear | 0%* | 0%* | 0%* | 0%* | 6%* | - | 98%* | 100%* | 29% |
| 7. Standard | 0%* | 0%* | 0%* | 0%* | 0%* | 2%* | - | 100%* | 15% |
| 8. AC | 0%* | 0%* | 0%* | 0%* | 0%* | 0%* | 0%* | - | 0% |

## Nonlinear

|  | 1 | 2 | 3 | 4 | 5 | 6 | 7 | 8 | **Avg** |
|---|---|---|---|---|---|---|---|---|---|
| 1. RoME | - | 56% | 100%* | 96%* | 98%* | 100%* | 100%* | 100%* | 93% |
| 2. RoME-BLM | 44% | - | 96%* | 98%* | 100%* | 100%* | 100%* | 100%* | 91% |
| 3. RoME-SU | 0%* | 4%* | - | 40% | 80%* | 100%* | 100%* | 100%* | 61% |
| 4. NNR-Linear | 4%* | 2%* | 60% | - | 82%* | 100%* | 100%* | 100%* | 64% |
| 5. IntelPooling | 2%* | 0%* | 20%* | 18%* | - | 98%* | 100%* | 100%* | 48% |
| 6. Neural-Linear | 0%* | 0%* | 0%* | 0%* | 2%* | - | 98%* | 100%* | 29% |
| 7. Standard | 0%* | 0%* | 0%* | 0%* | 0%* | 2%* | - | 100%* | 15% |
| 8. AC | 0%* | 0%* | 0%* | 0%* | 0%* | 0%* | 0%* | - | 0% |

Table 3: Pairwise comparisons between methods in the three settings of the main simulation. Each cell indicates the percent of repetitions (out of 50) in which the method listed in the row outperformed the method listed in the column in terms of final regret. Asterisks indicate p-values below 0.05 from paired two-sided t-tests on the differences in final regret. RoME and RoME-BLM perform well in all three settings, and their final regret is statistically indistinguishable. They substantially outperform all other methods in the Nonlinear setting.

In this section, we simulate actions and rewards under a rectangular array with 100 users and 100 time points. Although we still follow the staged recruitment regime depicted in Figure 1, at stage 100 we stop sampling actions and rewards for user 1; at stage 101 we stop sampling for user 2; and so on until we have sampled 100 time points for all 100 users. Aside from the shape of the data array, the setup is the same for this simulation as for the main simulation.

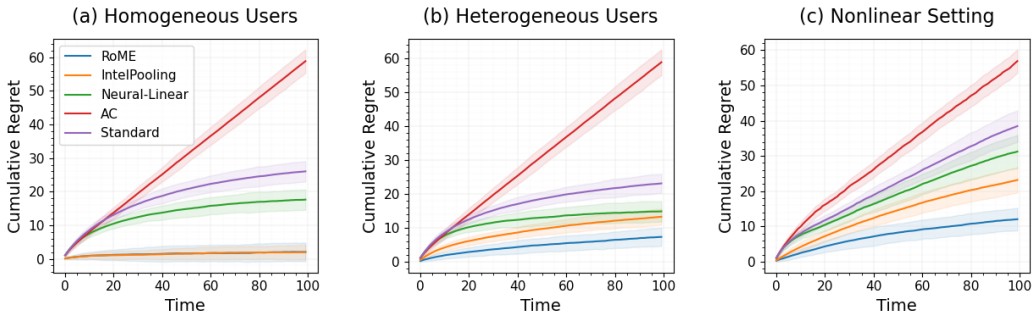

Figure 7: Cumulative regret in the (a) Homogeneous Users, (b) Heterogeneous Users, and (c) Nonlinear settings using a rectangular array of data in which we observe 100 time points for 100 users in a stagewise fashion as depicted in Figure 1. Similar to Figure 2, RoME is competitive in the first setting and substantially outperforms the competitors in the other settings.

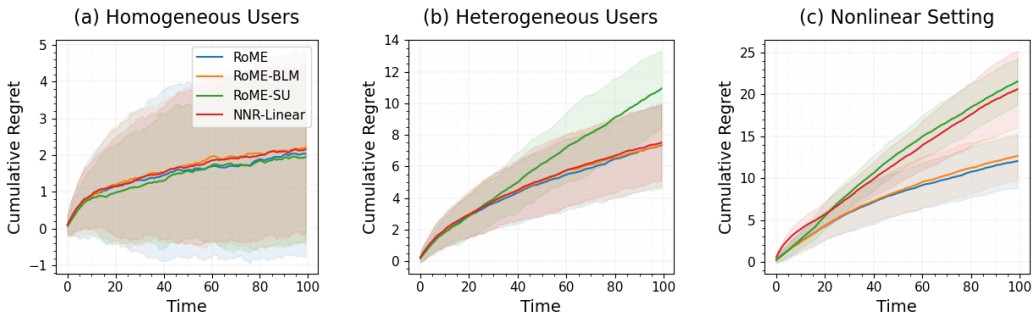

Figure 8: Cumulative regret in the (a) Homogeneous Users, (b) Heterogeneous Users, and (c) Nonlinear settings using a rectangular array of data in which we observe 100 time points for 100 users in a stagewise fashion as depicted in Figure 1. We observe the same performance ordering as in Figure 6, but here the relative differences are even larger.

The cumulative regret for these methods as a function of time—not stage—is shown in Figure 7. Qualitatively, the results are nearly identical to those from the main simulation (compare to Figure 2). The differences in final regret are even larger in this simulation than they were in the main simulation, presumably because this study exhibits greater variation in time effects due to the balance across time.

Figure 8 displays comparisons similar to those shown in Figure 6 but for the rectangular simulation. We again see that the results are qualitatively similar to the main simulation. The methods perform similarly in the Homogeneous Users setting. RoME-SU performs poorly in the Heterogeneous Users setting, but the other methods perform similarly. Finally, RoME and RoME-BLM substantially outperform the other methods in the nonlinear setting. Again, the differences in performance are even larger in this simulation than in the main simulation.

Table 4 displays results from pairwise method comparisons, similar to those shown in Table 3. Again, the results are qualitatively similar to but slightly more exaggerated than those from the main simulation. In particular, in this simulation study, RoME and RoME-BLM outperform all other methods in 100% of repetitions in the Nonlinear setting.

## Homogeneous Users

|  | 1 | 2 | 3 | 4 | 5 | 6 | 7 | 8 | **Avg** |
|---|---|---|---|---|---|---|---|---|---|
| 1. RoME | - | 58% | 44% | 54% | 48% | 100%* | 100%* | 100%* | 72% |
| 2. RoME-BLM | 42% | - | 42% | 48% | 40% | 100%* | 100%* | 100%* | 67% |
| 3. RoME-SU | 56% | 58% | - | 60% | 54% | 100%* | 100%* | 100%* | 75% |
| 4. NNR-Linear | 46% | 52% | 40% | - | 52% | 100%* | 100%* | 100%* | 70% |
| 5. IntelPooling | 52% | 60% | 46% | 48% | - | 100%* | 100%* | 100%* | 72% |
| 6. Neural-Linear | 0%* | 0%* | 0%* | 0%* | 0%* | - | 100%* | 100%* | 29% |
| 7. Standard | 0%* | 0%* | 0%* | 0%* | 0%* | 0%* | - | 100%* | 14% |
| 8. AC | 0%* | 0%* | 0%* | 0%* | 0%* | 0%* | 0%* | - | 0% |

## Heterogeneous Users

|  | 1 | 2 | 3 | 4 | 5 | 6 | 7 | 8 | **Avg** |
|---|---|---|---|---|---|---|---|---|---|
| 1. RoME | - | 52% | 100%* | 52% | 100%* | 100%* | 100%* | 100%* | 86% |
| 2. RoME-BLM | 48% | - | 100%* | 52% | 100%* | 100%* | 100%* | 100%* | 86% |
| 3. RoME-SU | 0%* | 0%* | - | 2%* | 92%* | 100%* | 100%* | 100%* | 56% |
| 4. NNR-Linear | 48% | 48% | 98%* | - | 100%* | 100%* | 100%* | 100%* | 85% |
| 5. IntelPooling | 0%* | 0%* | 8%* | 0%* | - | 88%* | 100%* | 100%* | 42% |
| 6. Neural-Linear | 0%* | 0%* | 0%* | 0%* | 12%* | - | 100%* | 100%* | 30% |
| 7. Standard | 0%* | 0%* | 0%* | 0%* | 0%* | 0%* | - | 100%* | 14% |
| 8. AC | 0%* | 0%* | 0%* | 0%* | 0%* | 0%* | 0%* | - | 0% |

## Nonlinear

|  | 1 | 2 | 3 | 4 | 5 | 6 | 7 | 8 | **Avg** |
|---|---|---|---|---|---|---|---|---|---|
| 1. RoME | - | 56%* | 100%* | 100%* | 100%* | 100%* | 100%* | 100%* | 94% |
| 2. RoME-BLM | 44%* | - | 100%* | 100%* | 100%* | 100%* | 100%* | 100%* | 92% |
| 3. RoME-SU | 0%* | 0%* | - | 28%* | 76%* | 100%* | 100%* | 100%* | 58% |
| 4. NNR-Linear | 0%* | 0%* | 72%* | - | 86%* | 100%* | 100%* | 100%* | 65% |
| 5. IntelPooling | 0%* | 0%* | 24%* | 14%* | - | 98%* | 100%* | 100%* | 48% |
| 6. Neural-Linear | 0%* | 0%* | 0%* | 0%* | 2%* | - | 100%* | 100%* | 29% |
| 7. Standard | 0%* | 0%* | 0%* | 0%* | 0%* | 0%* | - | 100%* | 14% |
| 8. AC | 0%* | 0%* | 0%* | 0%* | 0%* | 0%* | 0%* | - | 0% |

Table 4: Pairwise comparisons between methods in the three settings of the simulation with a rectangular array of data. As in Table 3, each cell indicates the percent of repetitions (out of 50) in which the method listed in the row outperformed the method listed in the column in terms of final regret. Asterisks indicate p-values below 0.05 from paired two-sided t-tests on the differences in final regret. The qualitative results are similar to those of Table 3. RoME and RoME-BLM perform well across all three settings and substantially outperform all other methods in the Nonlinear setting.

## A.5 Hyperparameter Sensitivity

In the simulation study, we chose hyperparameters for the fairest comparison possible, employing the same regularization parameters across methods. To assess robustness, we performed seven sensitivity analyses that alter hyperparameters for our methods while fixing those of the other methods; this approach gives the competing algorithms an advantage. Under these alternative hyperparameters, RoME and RoME-BLM still substantially outperform the other methods in the Nonlinear Setting.

Two of the sensitivity analyses led to no significant changes to Figure 2 and Table 3: setting $\delta = 0.05$ and $v = 10$. Below, we summarize the changes in the remaining analyses:

- Rescaling $\gamma$ by a factor of ten: This change resulted in RoME and RoME-BLM performing relatively better compared to the other methods in the Heterogeneous setting. In particular, RoME and RoME-BLM outperformed NNR-Linear in 70% and 72% of replications, compared to 56% and 52% using the original value of $\gamma$.

- Rescaling $\lambda$ by a factor of ten: Similar to the above results, this changed improved the performance of RoME and RoME-BLM compared to the other methods (especially RoME-SU and NNR-Linear) in the Heterogeneous Users setting.

- Adding (low and medium) noise to the nearest-neighbor network: RoME significantly outperformed RoME-BLM in the Nonlinear setting.

- Increasing the number of neighbors to 10: The performance gap between the full RoME algorithms (RoME and RoME-BLM) and RoME-SU increased to 88%, 76% compared to 80% and 72% under the original configuration, presumably due to increased network cohesion.

The results for the $\lambda$ sensitivity analysis are displayed in Table 5. Results for the remaining sensitivity analyses are available with our code.

# B Additional Details for Valentine Study

Personalizing treatment delivery in mobile health is a common application for online learning algorithms. We focus here on the Valentine study, a prospective, randomized-controlled, remotely administered trial designed to evaluate an mHealth intervention to supplement cardiac rehabilitation for low- and moderate-risk patients (Jeganathan et al., 2022; Golbus et al., 2023). We aim to use smartwatch data (Apple Watch and Fitbit) obtained from the Valentine study to learn the optimal timing of notification delivery given the users' current context.

## B.1 Data from the Valentine Study

Prior to the start of the trial, baseline data was collected from each of the participants (e.g., age, gender, baseline activity level, and health information). During the study, participants are randomized to either receive a notification ($A_t = 1$) or not ($A_t = 0$) at each of 4 daily time points (morning, lunchtime, mid-afternoon, evening), with probability 0.25. Contextual information was collected frequently (e.g., number of messages sent in prior week, step count variability in prior week, and pre-decision point step-counts).

Since the goal of the Valentine study is to increase participants' activity levels, we define the reward, $R_t$, as the step count for the 60 minutes following a decision point (log-transformed to eliminate skew). Our application also uses a subset of the baseline and contextual data; this subset contains the variables with the strongest association with the reward. Table 6 shows the features available to the bandit in the Valentine study data set.

For baseline variables, we use the participant's device model ($Z_1$, Fitbit coded as 1), the participant's step count variability in the prior week ($Z_2$), and a measure of the participant's pre-trial activity level based on an intake survey ($Z_3$, with larger values corresponding to higher activity levels).

At every decision point, before selecting an action, the learner sees two state variables: the participant's previous 30-minute step count ($S_1$, log-transformed) and the participant's phase of cardiac rehabilitation ($S_2$, dummy coded). The cardiac rehabilitation phase is defined based on a participant's

## Homogeneous Users

|  | 1 | 2 | 3 | 4 | 5 | 6 | 7 | 8 | **Avg** |
|---|---|---|---|---|---|---|---|---|---|
| 1. RoME | - | 58% | 48% | 64% | 38% | 100%* | 100%* | 100%* | 73% |
| 2. RoME-BLM | 42% | - | 38%* | 42% | 38%* | 100%* | 100%* | 100%* | 66% |
| 3. RoME-SU | 52% | 62%* | - | 58% | 46% | 100%* | 100%* | 100%* | 74% |
| 4. NNR-Linear | 36% | 58% | 42% | - | 32%* | 100%* | 100%* | 100%* | 67% |
| 5. IntelPooling | 62% | 62%* | 54% | 68%* | - | 100%* | 100%* | 100%* | 78% |
| 6. Neural-Linear | 0%* | 0%* | 0%* | 0%* | 0%* | - | 98%* | 100%* | 28% |
| 7. Standard | 0%* | 0%* | 0%* | 0%* | 0%* | 2%* | - | 100%* | 15% |
| 8. AC | 0%* | 0%* | 0%* | 0%* | 0%* | 0%* | 0%* | - | 0% |

## Heterogeneous Users

|  | 1 | 2 | 3 | 4 | 5 | 6 | 7 | 8 | **Avg** |
|---|---|---|---|---|---|---|---|---|---|
| 1. RoME | - | 52% | 92%* | 72%* | 100%* | 100%* | 100%* | 100%* | 88% |
| 2. RoME-BLM | 48% | - | 78%* | 68%* | 100%* | 100%* | 100%* | 100%* | 85% |
| 3. RoME-SU | 8%* | 22%* | - | 20%* | 92%* | 100%* | 100%* | 100%* | 63% |
| 4. NNR-Linear | 28%* | 32%* | 80%* | - | 100%* | 100%* | 100%* | 100%* | 77% |
| 5. IntelPooling | 0%* | 0%* | 8%* | 0%* | - | 96%* | 100%* | 100%* | 43% |
| 6. Neural-Linear | 0%* | 0%* | 0%* | 0%* | 4%* | - | 96%* | 100%* | 29% |
| 7. Standard | 0%* | 0%* | 0%* | 0%* | 0%* | 4%* | - | 100%* | 15% |
| 8. AC | 0%* | 0%* | 0%* | 0%* | 0%* | 0%* | 0%* | - | 0% |

## Nonlinear

|  | 1 | 2 | 3 | 4 | 5 | 6 | 7 | 8 | **Avg** |
|---|---|---|---|---|---|---|---|---|---|
| 1. RoME | - | 52% | 96%* | 96%* | 98%* | 100%* | 100%* | 100%* | 92% |
| 2. RoME-BLM | 48% | - | 92%* | 96%* | 96%* | 98%* | 100%* | 100%* | 90% |
| 3. RoME-SU | 4%* | 8%* | - | 60% | 90%* | 100%* | 100%* | 100%* | 66% |
| 4. NNR-Linear | 4%* | 4%* | 40% | - | 82%* | 100%* | 100%* | 100%* | 61% |
| 5. IntelPooling | 2%* | 4%* | 10%* | 18%* | - | 98%* | 100%* | 100%* | 47% |
| 6. Neural-Linear | 0%* | 2%* | 0%* | 0%* | 2%* | - | 98%* | 100%* | 29% |
| 7. Standard | 0%* | 0%* | 0%* | 0%* | 0%* | 2%* | - | 100%* | 15% |
| 8. AC | 0%* | 0%* | 0%* | 0%* | 0%* | 0%* | 0%* | - | 0% |

Table 5: Pairwise comparisons between methods in a sensitivity analysis in which we scale $\lambda$ by a factor of ten. As in Table 3, each cell indicates the percent of repetitions (out of 50) in which the method listed in the row outperformed the method listed in the column in terms of final regret. Asterisks indicate p-values below 0.05 from paired two-sided t-tests on the differences in final regret.

| Feature | Description | Interaction | Baseline |
|---|---|---|---|
| Phase II | 1 if in Phase II, 0 o.w. | $\checkmark$ | $\checkmark$ |
| Phase III | 1 if in Phase II, 0 o.w. | $\checkmark$ | $\checkmark$ |
| Steps in prior 30 minutes | log transformed | $\checkmark$ | $\checkmark$ |
| Pre-trial average daily steps | log transformed | $\times$ | $\checkmark$ |
| Device | 1 if Fitbit, 0 o.w. | $\times$ | $\checkmark$ |
| Prior week step count variability | SD of the rewards in the previous week | $\times$ | $\checkmark$ |

Table 6: List of features available to the bandit in the Valentine study. The features available to model the action interaction (effect of sending an anti-sedentary message) and to model the baseline (reward under no action) are denoted via a "$\checkmark$" in the corresponding column, otherwise $\times$.

|                     | Df   | Sum Sq | Mean Sq | F value | Pr(>F)  |     |
| ------------------- | ---- | ------ | ------- | ------- | ------- | --- |
| ParticipantIdentifier | 107  | 264    | 2.464   | 8.09    | <2e-16  | *** |
| Week                | 25   | 17     | 0.683   | 2.24    | 4e-04   | *** |
| Residuals           | 2265 | 690    | 0.305   |         |         |     |

Table 7: ANOVA analysis of the pseudo-outcomes in the Valentine study. The small p-values constitute strong evidence that the treatment effects differ by participant and over time.

time in the study: month 1 represents Phase I, month 2-4 represents Phase II, and month 5-6 represents Phase III.

## B.2    Justification of Assumption 4

The motivation for Assumption 4 arises from an exploratory analysis we performed using data from the Valentine study. We constructed the pseudo-reward for each observation as suggested by Equation (1) and then conducted an ANOVA test. The results show clear heterogeneity between users and across time, motivating the adoption of user- and time-specific random effects. The results of the ANOVA test are displayed in Table 7. Figure 9 shows the shape of the heterogeneity in the treatment effects over time.

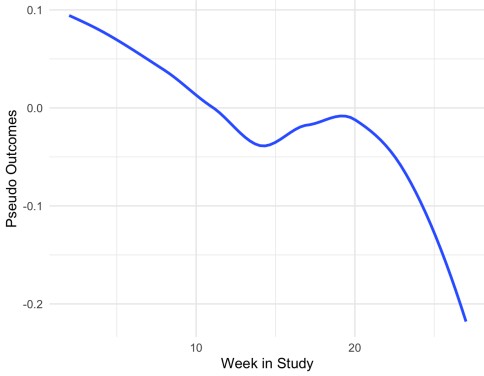

Figure 9: The time heterogeneity in the pseudo-outcomes. We calculated the pseudo-outcomes using (1), then averaged them across participants and plotted them over time. This exploratory analysis shows evidence that the causal effects vary substantially over time.

## B.3    Evaluation

The Valentine study collected the sensor-based features at 4 decision points per day for each study participant. The reward for each message was defined to be $\log(0.5 + x)$, where $x$ is the step count of the participant in the 60 minutes following the notification. As noted in the introduction, the baseline reward (the expected step count of a subject when no message is sent), not only depends on the state in a complex way but is likely dependent on a large number of time-varying observed variables. Both of these characteristics (complex, time-varying baseline reward function) suggest using our proposed approach.

We generated 100 bootstrap samples and ran our contextual bandit on them, considering the binary action of whether or not to send a message at a given decision point based on the contextual variables $S_1$ and $S_2$. Each user is considered independently and with a cohesion network, for maximum personalization and independence of results. To guarantee that messages have a positive probability of being sent, we only sample the observations with notification randomization probability between $0.01$ and $0.99$. In the case of the algorithm utilizing NNR, we chose four baseline characteristics (gender, age, device, and baseline average daily steps) to establish a measure of "distance" between users. For this analysis, the value of $k$ representing the number of nearest neighbors was set to 5. To utilize bootstrap sampling, we train the Neural-Linear method's neural network using out-of-bag samples. The neural network architecture comprises a single hidden layer with two hidden nodes.

The input contains both the baseline characteristics and the contextual variables and the activation function applied here is the *softplus* function, defined as softplus$(x) = \log(1 + \exp{(x)})$.

We performed an offline evaluation of the contextual bandit algorithms using an inverse propensity score (IPS) version of the method from Li et al. (2010), where the sequence of states, actions, and rewards in the data are used to form a near-unbiased estimate of the average expected reward achieved by each algorithm, averaging over all users.

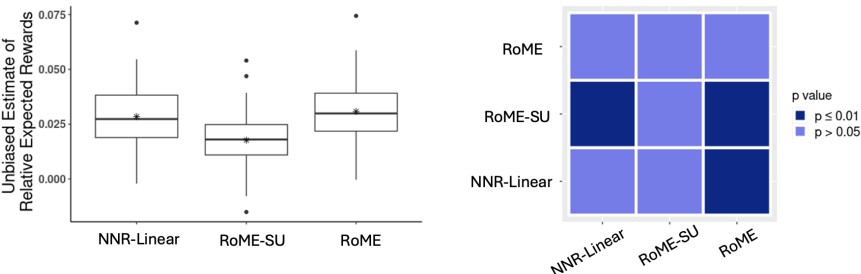

Figure 10: **(left)** Unbiased estimates of the average per-trial reward for all three ablation algorithms, relative to the reward obtained under the pre-specified Valentine Study randomization policy across 20 multiple-imputed data sets. And **(right)** p-values from the pairwise paired t-tests.

## B.4    An ablation study

We further investigate the primary contributors to the algorithm's performance, we conducted a parallel ablation study using real-world data analysis. In this study, we compared RoME (as detailed in the main paper) with two additional methods for reference: RoME-SU and NNR-Linear. Each comparison focuses on specific elements of our algorithm to discern their impact on RoME's overall performance.

The differences between these methods are illustrated in Figure. 10. In this particular dataset, NNR plays a more significant role in driving the overall superior performance of our algorithm.

## B.5    Inverse Propensity Score (IPS) offline evaluation

In the implemented Valentine study, the treatment was randomized with a constant probability $p_t = 0.25$ at each time $t$. To conduct off-policy evaluation using our proposed algorithm and the competing variations of the TS algorithm, we outline the IPS estimator for an unbiased estimate of the per-trial expected reward based on what has been studied in Li et al. (2010).

Given the logged data $\mathcal{D} = \{s_t = s_t, A_t = a_t, R_t = r_t\}_{t=1}^{T}$ collected under the policy $\mathbf{p} = \{p_t\}_{t=1}^{T}$, and the treatment policy being evaluated $\pi = \{\pi_t\}_{t=1}^{T}$, the objective of this offline estimator is to reweight the observed reward sequence $\{R_t\}_{t=1}^{T}$ to assign varying importance to actions based on the propensities of both the original and new policies in selecting them.

**Lemma 1** (Unbiasedness of the IPS estimator). *Assuming the positivity assumption in logging, which states that for any given $s$ and $a$, if $p_t(a|s) > 0$, then we also have $\pi_t(a|s) > 0$, we can obtain an unbiased per-trial expected reward using the following IPS estimator:*

$$\hat{R}_{IPS} = \frac{1}{T} \sum_{t=1}^{T} \frac{\pi_t(a_t|s_t)}{p_t(a_t|s_t)} r_t$$

As mentioned in the previous section, we restrict our sampling to observations with notification randomization probabilities ranging from 0.01 to 0.99. This selection criterion ensures the satisfaction of the positivity assumption. The proof essentially follows from definition, we have:

*Proof.*

$$\mathbb{E}[R_{\text{IPS}}] = \mathbb{E}_{\mathbf{p}}\left[\frac{1}{T}\sum_{t=1}^{T}\frac{\pi_t(a_t|s_t)}{p_t(a_t|s_t)}R_t(a_t, s_t)\right]$$

$$= \frac{1}{T}\sum_{t=1}^{T}\frac{\pi_t(a_t|s_t)}{p_t(a_t|s_t)}R_t(a_t, s_t) \times p_t(a_t|s_t)$$

$$= \frac{1}{T}\sum_{t=1}^{T}\pi_t(a_t|s_t)R_t(a_t, s_t)$$

$$= \mathbb{E}_{\pi}\left[\frac{1}{T}\sum_{t=1}^{T}R_t(a_t, s_t)\right]$$

$\square$

To address the instability issue caused by reweighting in some cases, we use a Self-Normalized Inverse Propensity Score (SNIPS) estimator. This estimator scales the results by the empirical mean of the importance weights, and still maintains the property of unbiasedness.

$$\hat{R}_{\text{SNIPS}} = \frac{\hat{R}_{\text{IPS}}}{\frac{1}{T}\sum_{t=1}^{T}\frac{\pi_t(a_t|s_t)}{p_t(a_t|s_t)}} = \frac{\sum_{t=1}^{T}\frac{\pi_t(a_t|s_t)}{p_t(a_t|s_t)}r_t}{\sum_{t=1}^{T}\frac{\pi_t(a_t|s_t)}{p_t(a_t|s_t)}} \tag{6}$$

## C   Additional Details for the Intern Health Study (IHS)

To further enhance the competitive performance of our proposed RoME algorithm, we performed an additional comparative analysis using a real-world data set from the Intern Health Study (IHS) (NeCamp et al., 2020). This micro-randomized trial investigated the use of mHealth interventions aimed at improving the behavior and mental health of individuals in stressful work environments. The estimates obtained represent the improvement in average reward relative to the original constant randomization, averaging across stages (K = 30) and participants (N = 1553). The available IHS data consist of 20 multiple-imputed data sets. We apply the algorithms to each imputed data set and perform a comparative analysis of the competing algorithms. The results presented in Figure 11 shows our proposed RoME algorithm achieved significantly higher rewards than the other three competing ones and demonstrated performance comparable to the AC algorithm. These findings further support the advantages of our proposed algorithm.

### C.1   Data from the IHS

Prior to the start of the trial, baseline data was collected on each of the participants (e.g., institution, specialty, gender, baseline activity level, and health information). During the study, participants are randomized to either receive a notification ($A_t = 1$) or not ($A_t = 0$) every day, with probability $3/8$. Contextual information was collected frequently (e.g., step count in prior five days, and current day in study).

We define the reward, $R_t$, as the step count on the following day (cubic root). Our application also uses a subset of the baseline and contextual data; this subset contains the variables with the strongest association to the reward. Table 8 shows the features available to the bandit in the IHS data set.

At every decision point, before selecting an action, the learner sees two state variables: the participant's previous 5-day average daily step count ($S_1$, cubic root) and the participant's day in the study ($S_2$, an integer from 1 to 30).

### C.2   Evaluation

We run our contextual bandit on the IHS data, considering the binary action of whether or not to send a message at a given decision point based on the contextual variables $S_1$ and $S_2$. Each user is considered independently and with a cohesion network, for maximum personalization and

| Feature | Description | Interaction | Baseline |
|---------|-------------|:-----------:|:--------:|
| Day in study | an integer from 1 to 30 | $\checkmark$ | $\checkmark$ |
| Average daily steps in prior five days | cubic root | $\checkmark$ | $\checkmark$ |
| Average daily sleep in prior five days | cubic root | $\times$ | $\checkmark$ |
| Average daily mood in prior five days | a Likert scale from $1-10$ | $\times$ | $\checkmark$ |
| Pre-intern average daily steps | cubic root | $\times$ | $\checkmark$ |
| Pre-intern average daily sleep | cubic root | $\times$ | $\checkmark$ |
| Pre-intern average daily mood | a Likert scale from $1-10$ | $\times$ | $\checkmark$ |
| Sex | Gender | $\times$ | $\checkmark$ |
| Week category | The theme of messages in a specific week (mood, sleep, activity, or none) | $\times$ | $\checkmark$ |
| PHQ score | PHQ total score | $\times$ | $\checkmark$ |
| Early family environment | higher score indicates higher level of adverse experience | $\times$ | $\checkmark$ |
| Personal history of depression | | $\times$ | $\checkmark$ |
| Neuroticism (Emotional experience) | higher score indicates higher level of neuroticism | $\times$ | $\checkmark$ |

Table 8: List of features available to the bandit in the IHS. The features available to model the action interaction (effect of sending a mobile prompt) and to model the baseline (reward under no action) are denoted via a "$\checkmark$" in the corresponding column, otherwise $\times$.

independence of results. To guarantee that messages have a positive probability of being sent, we only sample the observations with notification randomization probability between $0.01$ and $0.99$. For the algorithm employing NNR, we defined participants in the same institution as their own "neighbors". This definition enables the flexibility for the value of $k$, representing the number of nearest neighbors, to vary for each participant based on their specific institutional context. Furthermore, in our study setting, we assume that individuals from the same institution enter the study simultaneously as a group. Due to the limited access to prior data, we are unable to build the neural linear models as in the Valentine Study.

We utilized 20 multiple-imputed data sets and performed an offline evaluation of the contextual bandit algorithms on each data set. The result is presented below in Figure 11. Similar to Section B.4, here we also compared RoME (as detailed in the main paper) with RoME-SU and NNR-Linear. The differences between these methods are illustrated in Figure. 12. In this specific dataset, both NNR and DML exhibit comparable performance individually, but their combined effect significantly enhances the overall performance of the algorithm.

## D   Additional Details for Algorithm 1

This appendix briefly discusses two details of Algorithm 1. The first is efficient computation of $V^{-1}$. Because $V$ is a large matrix, a full matrix inversion is expensive. Fortunately, however, we can dramatically reduce the necessary computational requirements because each additional $(i, t)$ involves a rank-one perturbation to $V$. Consequently, we can apply the Sherman–Morrison formula to speed up computations. We leveraged this trick in our implementation of RoME for the simulation study by using efficient rank-one updates available in the SuiteSparse library (Davis & Hu, 2011).

The second detail is the requirements for the distribution, $\mathcal{D}^{TS}$:

**Definition 1.** *$\mathcal{D}^{TS}$ is a multivariate distribution on $\mathbb{R}^d$ absolutely continuous with respect to Lebesgue measure which satisfies: 1. (anti-concentration) that there exists a strictly positive probability $p$ such that for any $u \in \mathbb{R}^d$ with $\|u\| = 1$, $\mathbb{P}(u^\top \eta \geq 1) \geq p$; 2. (concentration) there exists $c, c'$ positive constants such that $\forall \delta \in (0, 1)$, $P(\|\eta\| \leq \sqrt{cd\log(c'd/\delta)}) \geq 1 - \delta$; and 3. it possesses a finite second moment, $\mathbb{E}\|\eta\|^2$, where $\eta \sim \mathcal{D}^{TS}$.*

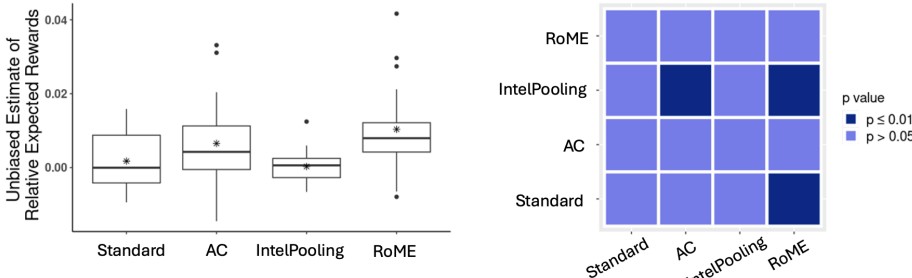

Figure 11: **(left)** Unbiased estimates of the average per-trial reward for all four competing algorithms, relative to the reward obtained under the pre-specified Intern Health Study randomization policy across 20 multiple-imputed data sets. And **(right)** p-values from the pairwise paired t-tests. The dark shade in the last column indicates that the proposed RoME algorithm achieved significantly higher rewards than the other three competing algorithms while demonstrating comparable performance to the AC algorithm.

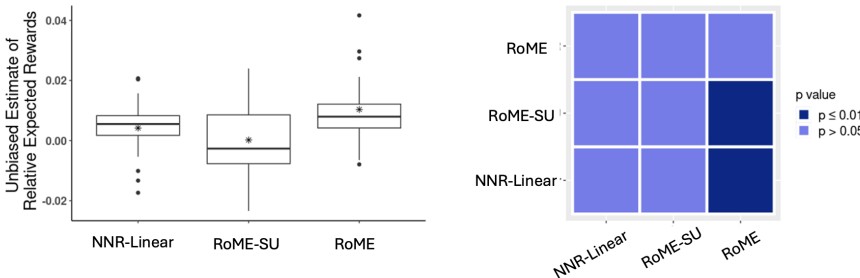

Figure 12: **(left)** Unbiased estimates of the average per-trial reward for all three ablation algorithms, relative to the reward obtained under the pre-specified Intern Health Study randomization policy across 20 multiple-imputed data sets. And **(right)** p-values from the pairwise paired t-tests.

While a Gaussian prior satisfies Definition 1, this approach allows us to move beyond Bayesian posteriors to generic randomized policies. As discussed in the main paper, we recommend setting $\mathcal{D}^{TS}$ to a multivariate Gaussian distribution or multivariate t-distribution. These choices simplify computations because the probability computation in (5) simplifies to an evaluation of the CDF of either (a) a univariate Gaussian or (b) a univariate t-distribution. The mean and variance of the corresponding univariate distribution can easily be worked out using the moments of a linear combination of random variables.

# E   Regret Bound

## E.1   Double Robustness of Pseudo-Reward

Going forward, we use the notation $\Delta_{i,t}^f(s, \bar{a}) \coloneqq f_{i,t}(s, \bar{a}) - f_{i,t}(s, 0)$ to denote the prediction of the differential reward. In the computations in this section, we implicitly condition on $\mathcal{H}_{i,t}$ and $A_{i,t} \in \{0, \bar{a}\}$.

**Lemma 2.** *If either $p_{i,t} = \pi_{i,t}$ or $f_{i,t} = r_{i,t}$, then*

$$\mathbb{E}\left(\tilde{R}_{i,t}^f | s, \bar{a}\right) = \Delta_{i,t}(s, \bar{a}).$$

*That is, the pseudo-reward is an unbiased estimator of the true differential reward.*

*Proof.* Recall that

$$\tilde{R}_{i,t}^f = \frac{R_{it} - f_{i,t}(s, A_{i,t})}{\delta_{A_{i,t}=\bar{a}} - \pi_{i,t}(0|s)} + \Delta_{i,t}^f(s, \bar{a})$$

**Case I: $\pi$'s are correctly specified**

Then

$$\mathbb{E}\left[\frac{R_{it}}{\delta_{A_{i,t}=\bar{a}} - \pi_{i,t}(0|s)}\bigg|s, \bar{a}\right] = r_{i,t}(s, \bar{a}) - r_{i,t}(s, 0)$$
$$= \Delta_{i,t}(s, \bar{a})$$

and

$$\mathbb{E}\left[\frac{f_{i,t}(s, A_{i,t})}{\delta_{A_{i,t}=\bar{a}} - \pi_{i,t}(0|s)}\bigg|s, \bar{a}\right] = f_{i,t}(s, \bar{a}) - f_{i,t}(s, 0)$$
$$= \Delta_{i,t}^f(s, \bar{a})$$

so that

$$\mathbb{E}\left[\frac{R_{it} - f_{i,t}(s, A_{i,t})}{\delta_{A_{i,t}=\bar{a}} - \pi_{i,t}(0|s)} + \Delta_{i,t}^f(s, \bar{a})\bigg|s, \bar{a}\right] = \Delta_{i,t}(s, \bar{a}) - \Delta_{i,t}^f(s, \bar{a}) + \Delta_{i,t}^f(s, \bar{a})$$
$$= \Delta_{i,t}(s, \bar{a})$$

**Case II: $f$ correctly specified**

$$\mathbb{E}\left[\frac{R_{it}}{\delta_{A_{i,t}=\bar{a}} - \pi_{i,t}(0|s)}\bigg|s, \bar{a}\right] = \frac{1 - p_{i,t}(0|s)}{1 - \pi_{i,t}(0|s)}r_{i,t}(s, \bar{a}) - \frac{p_{i,t}(0|s)}{\pi_{i,t}(0|s)}r_{i,t}(s, 0)$$

and

$$\mathbb{E}\left[\frac{f_{i,t}(s, A_{i,t})}{\delta_{A_{i,t}=\bar{a}} - \pi_{i,t}(0|s)}\bigg|s, \bar{a}\right] = \frac{1 - p_{i,t}(0|s)}{1 - \pi_{i,t}(0|s)}f_{i,t}(s, \bar{a}) - \frac{p_{i,t}(0|s)}{\pi_{i,t}(0|s)}f_{i,t}(s, \bar{0})$$
$$= \frac{1 - p_{i,t}(0|s)}{1 - \pi_{i,t}(0|s)}r_{i,t}(s, \bar{a}) - \frac{p_{i,t}(0|s)}{\pi_{i,t}(0|s)}r_{i,t}(s, \bar{0})$$

and

$$\mathbb{E}\left[\Delta_{i,t}^f(s, \bar{a})\bigg|s, \bar{a}\right] = \Delta_{i,t}(s, \bar{a})$$

$$\mathbb{E}\left[\frac{R_{it} - f_{i,t}(s, \bar{A}_{i,t})}{\delta_{A_{i,t}=\bar{a}} - \pi_{i,t}(0|s)} + \Delta_{i,t}^f(s, \bar{a})\bigg|s, \bar{a}\right] = \Delta_{i,t}(s, \bar{a})$$

$\square$

## E.2  Preliminaries

**Lemma 3.** *Let $X$ be a mean-zero sub-Gaussian random variable with variance proxy $v^2$ and $Y$ be a bounded random variable such that $|Y| \leq B$ for some $0 \leq B < \infty$. Then $XY$ is sub-Gaussian with variance proxy $v^2 B^2$.*

*Proof.* Recall that $X$ being mean-zero sub-Gaussian means that

$$P(|X| \geq t) \leq 2\exp\left(-\frac{t^2}{2v^2}\right).$$

Now note that
$$|XY| \leq |X|B$$
so that if $|XY| > t$, then $|X|B > t$. Thus by monotonicity

$$P(|XY| \geq t) \leq P(|X|B \geq t)$$
$$= P\left(|X| \geq \frac{t}{B}\right)$$
$$\leq 2\exp\left(-\frac{t^2}{2B^2 v^2}\right)$$

as desired. $\qquad\square$

**Lemma 4.** *If $X, Y$ are sub-Gaussian with variance proxies $v_x^2$, $v_y^2$, respectively, then $\alpha X + \beta Y$ is sub-Gaussian with variance proxy $\alpha^2 v_x^2 + \beta^2 v_y^2 \forall \alpha, \beta \in \mathbb{R}$.*

*Proof.* Recall the equivalent definition of sub-Gaussianity that $X, Y$ are sub-Gaussian iff for some $a, b > 0$ and all $\lambda > 0$

$$\mathbb{E}\exp\left(\lambda(X - \mathbb{E}X)\right) \leq \exp(\lambda^2 v_x^2/2),$$
$$\mathbb{E}\exp\left(\lambda(Y - \mathbb{E}Y)\right) \leq \exp(\lambda^2 v_y^2/2).$$

Then

$$\mathbb{E}\exp\left(\lambda(\alpha X + \beta Y - \alpha\mathbb{E}X - \beta\mathbb{E}Y)\right) \leq \sqrt{\mathbb{E}\exp\left(2\alpha\lambda(X - \mathbb{E}X)\right)}\sqrt{\mathbb{E}\exp\left(2\beta\lambda(Y - \mathbb{E}Y)\right)}$$
$$\leq \sqrt{\exp(2\alpha^2\lambda^2 v_x^2)}\sqrt{\exp(2\beta^2\lambda^2 v_y^2)}$$
$$= \exp((\alpha^2 v_x^2 + \beta^2 v_y^2)\lambda^2)$$

$\qquad\square$

The following Lemma gives the sub-Gaussianity and variance of the difference between the pseudo-reward and its expectation. We see that in the variance, all terms except those involving the inverse propensity weighted noise variance vanish as $f_{i,t}$ becomes a better estimate of $r_{i,t}$.

**Lemma 5.** *The difference between the pseudo-reward and its expectation (taken wrt the action and noise) is mean zero sub-Gaussian with variance*

$$Var\left(\tilde{R}_{i,t}^f(s, \bar{a})\right) = \frac{(r_{i,t}(s,\bar{a}) - f_{i,t}(s,\bar{a}))^2 \pi_{i,t}(0|s) + Var(\epsilon_{i,t})}{1 - \pi_{i,t}(0|s)}$$
$$+ \frac{(r_{i,t}(s,0) - f_{i,t}(s,0))^2[1 - \pi_{i,t}(0|s)] + Var(\epsilon_{i,t})}{\pi_{i,t}(0|s)}$$
$$- 2(r_{i,t}(s,\bar{a}) - f_{i,t}(s,\bar{a}))(r_{i,t}(s,0) - f_{i,t}(s,0))$$

*Proof.* We need to show that it is sub-Gaussian and upper bound its variance. We write the difference as

$$\tilde{R}_{i,t}^f(s,\bar{a}) - \mathbb{E}[\tilde{R}_{i,t}^f|s,\bar{a}] = \tilde{R}_{i,t}^f(s,\bar{a}) - \Delta_{i,t}(s,\bar{a})$$
$$= \frac{R_{it} - f_{i,t}(s, A_{i,t})}{\delta_{A_{i,t}=\bar{a}} - \pi_{i,t}(0|s)} + \Delta_{i,t}^f(s,\bar{a}) - \Delta_{i,t}(s,\bar{a})$$
$$= \frac{r_{i,t}(s, A_{i,t}) - f_{i,t}(s, A_{i,t}) + \epsilon_{i,t}}{\delta_{A_{i,t}=\bar{a}} - \pi_{i,t}(0|s)} + \Delta_{i,t}^f(s,\bar{a}) - \Delta_{i,t}(s,\bar{a})$$

Note that $|r_{i,t}(s, A_{i,t})| \leq \max\left(|r_{i,t}(s,\bar{a})|, |r_{i,t}(s,0)|\right) \leq M$ and $|f_{i,t}(s, A_{i,t})| \leq \max\left(|f_{i,t}(s,\bar{a})|, |f_{i,t}(s,0)|\right) \leq M$. Thus, since $\left|1/\{\delta_{A_{i,t}=\bar{a}} - \pi_{i,t}(0|s)\}\right|$ is upper bounded (because $\pi_{i,t}(0|s) \in [\pi_{\min}, \pi_{\max}]$), we have that $\frac{r_{i,t}(s, A_{i,t}) - f_{i,t}(s, A_{i,t})}{\delta_{A_{i,t}=\bar{a}} - \pi_{i,t}(0|s)}$ is bounded and thus (not necessarily mean zero) sub-Gaussian. Since $\epsilon_{i,t}$ is sub-Gaussian, its denominator is bounded, and the

remaining terms are deterministic, the entire difference between the pseudo-reward and its mean is sub-Gaussian. Now

$$\mathrm{Var}(\tilde{R}_{i,t}^f(s,\bar{a}) - \mathbb{E}[\tilde{R}_{i,t}^f|s,\bar{a}]) = \mathrm{Var}\left(\tilde{R}_{i,t}^f(s,\bar{a})\right)$$
$$= \mathbb{E}\left[\tilde{R}_{i,t}^f(s,\bar{a})^2\right] - \Delta_{i,t}(s,\bar{a})^2 \qquad (7)$$

since $\mathbb{E}[\tilde{R}_{i,t}^f|s,\bar{a}]$ is not random. Now we expand the first term on the rhs.

$$\mathbb{E}\left[\tilde{R}_{i,t}^f(s,\bar{a})^2\right] = \mathbb{E}\left[\left(\frac{r_{i,t}(s,A_{i,t}) - f_{i,t}(s,A_{i,t}) + \epsilon_{i,t}}{\delta_{A_{i,t}=\bar{a}} - \pi_{i,t}(0|s)} + \Delta_{i,t}^f(s,\bar{a})\right)^2\right]$$
$$= \mathbb{E}\left[\left(\frac{r_{i,t}(s,A_{i,t}) - f_{i,t}(s,A_{i,t}) + \epsilon_{i,t}}{\delta_{A_{i,t}=\bar{a}} - \pi_{i,t}(0|s)}\right)^2\right]$$
$$+ 2\mathbb{E}\left[\frac{r_{i,t}(s,A_{i,t}) - f_{i,t}(s,A_{i,t}) + \epsilon_{i,t}}{\delta_{A_{i,t}=\bar{a}} - \pi_{i,t}(0|s)}\right]\Delta_{i,t}^f(s,\bar{a}) + \Delta_{i,t}^f(s,\bar{a})^2$$
$$= \mathbb{E}\left[\left(\frac{r_{i,t}(s,A_{i,t}) - f_{i,t}(s,A_{i,t}) + \epsilon_{i,t}}{\delta_{A_{i,t}=\bar{a}} - \pi_{i,t}(0|s)}\right)^2\right]$$
$$+ 2(\Delta_{i,t}(s,\bar{a}) - \Delta_{i,t}^f(s,\bar{a}))\Delta_{i,t}^f(s,\bar{a}) + \Delta_{i,t}^f(s,\bar{a})^2 \qquad (8)$$

For the first term on the rhs of (8),

$$\mathbb{E}\left[\left(\frac{r_{i,t}(s,A_{i,t}) - f_{i,t}(s,A_{i,t}) + \epsilon_{i,t}}{\delta_{A_{i,t}=\bar{a}} - \pi_{i,t}(0|s)}\right)^2\right]$$
$$= \mathbb{E}\left[\left(\frac{r_{i,t}(s,A_{i,t}) - f_{i,t}(s,A_{i,t})}{\delta_{A_{i,t}=\bar{a}} - \pi_{i,t}(0|s)}\right)^2\right] + \mathbb{E}\left[\left(\frac{\epsilon_{i,t}}{\delta_{A_{i,t}=\bar{a}} - \pi_{i,t}(0|s)}\right)^2\right]$$
$$= \frac{(r_{i,t}(s,\bar{a}) - f_{i,t}(s,\bar{a}))^2 + \mathbb{E}[\epsilon_{i,t}^2]}{1 - \pi_{i,t}(0|s)} + \frac{(r_{i,t}(s,0) - f_{i,t}(s,0))^2 + \mathbb{E}[\epsilon_{i,t}^2]}{\pi_{i,t}(0|s)}$$

so that plugging this into (8), we have

$$\mathbb{E}\left[\tilde{R}_{i,t}^f(s,\bar{a})^2\right] = \frac{(r_{i,t}(s,\bar{a}) - f_{i,t}(s,\bar{a}))^2 + \mathbb{E}[\epsilon_{i,t}^2]}{1 - \pi_{i,t}(0|s)} + \frac{(r_{i,t}(s,0) - f_{i,t}(s,0))^2 + \mathbb{E}[\epsilon_{i,t}^2]}{\pi_{i,t}(0|s)}$$
$$+ 2(\Delta_{i,t}(s,\bar{a}) - \Delta_{i,t}^f(s,\bar{a}))\Delta_{i,t}^f(s,\bar{a}) + \Delta_{i,t}^f(s,\bar{a})^2$$

and plugging this into (7) we obtain the variance.

$$\mathrm{Var}\left(\tilde{R}_{i,t}^f(s,\bar{a})\right) = \frac{(r_{i,t}(s,\bar{a}) - f_{i,t}(s,\bar{a}))^2 + \mathrm{Var}(\epsilon_{i,t})}{1 - \pi_{i,t}(0|s)} + \frac{(r_{i,t}(s,0) - f_{i,t}(s,0))^2 + \mathrm{Var}(\epsilon_{i,t})}{\pi_{i,t}(0|s)}$$
$$+ 2(\Delta_{i,t}(s,\bar{a}) - \Delta_{i,t}^f(s,\bar{a}))\Delta_{i,t}^f(s,\bar{a}) + \Delta_{i,t}^f(s,\bar{a})^2 - \Delta_{i,t}(s,\bar{a})^2 \qquad (9)$$

Note that

$$2(\Delta_{i,t}(s,\bar{a}) - \Delta_{i,t}^f(s,\bar{a}))\Delta_{i,t}^f(s,\bar{a}) + \Delta_{i,t}^f(s,\bar{a})^2 - \Delta_{i,t}(s,\bar{a})^2$$
$$= 2(\Delta_{i,t}(s,\bar{a}) - \Delta_{i,t}^f(s,\bar{a}))\Delta_{i,t}^f(s,\bar{a}) - (\Delta_{i,t}(s,\bar{a}) - \Delta_{i,t}^f(s,\bar{a}))(\Delta_{i,t}(s,\bar{a}) + \Delta_{i,t}^f(s,\bar{a}))$$
$$= (\Delta_{i,t}(s,\bar{a}) - \Delta_{i,t}^f(s,\bar{a}))(\Delta_{i,t}^f(s,\bar{a}) - \Delta_{i,t}(s,\bar{a}))$$
$$= -(\Delta_{i,t}(s,\bar{a}) - \Delta_{i,t}^f(s,\bar{a}))^2$$

and plugging this into (9),

$$\mathrm{Var}\left(\tilde{R}_{i,t}^f(s,\bar{a})\right) = \frac{(r_{i,t}(s,\bar{a}) - f_{i,t}(s,\bar{a}))^2 + \mathrm{Var}(\epsilon_{i,t})}{1 - \pi_{i,t}(0|s)} + \frac{(r_{i,t}(s,0) - f_{i,t}(s,0))^2 + \mathrm{Var}(\epsilon_{i,t})}{\pi_{i,t}(0|s)}$$
$$- (\Delta_{i,t}(s,\bar{a}) - \Delta_{i,t}^f(s,\bar{a}))^2$$

as desired. Now note that

$$\left(\Delta_{i,t}(s,\bar{a}) - \Delta_{i,t}^f(s,\bar{a})\right)^2 = (r_{i,t}(s,\bar{a}) - r_{i,t}(s,0) - (f_{i,t}(s,\bar{a}) - f_{i,t}(s,0)))^2$$

$$= (r_{i,t}(s,\bar{a}) - f_{i,t}(s,\bar{a}) - (r_{i,t}(s,0) - f_{i,t}(s,0)))^2$$

$$= (r_{i,t}(s,\bar{a}) - f_{i,t}(s,\bar{a}))^2 + (r_{i,t}(s,0) - f_{i,t}(s,0))^2$$

$$\quad - 2(r_{i,t}(s,\bar{a}) - f_{i,t}(s,\bar{a}))(r_{i,t}(s,0) - f_{i,t}(s,0))$$

so that

$$\text{Var}\left(\tilde{R}_{i,t}^f(s,\bar{a})\right) = \frac{(r_{i,t}(s,\bar{a}) - f_{i,t}(s,\bar{a}))^2 \pi_{i,t}(0|s) + \text{Var}(\epsilon_{i,t})}{1 - \pi_{i,t}(0|s)}$$

$$+ \frac{(r_{i,t}(s,0) - f_{i,t}(s,0))^2 [1 - \pi_{i,t}(0|s)] + \text{Var}(\epsilon_{i,t})}{\pi_{i,t}(0|s)}$$

$$- 2(r_{i,t}(s,\bar{a}) - f_{i,t}(s,\bar{a}))(r_{i,t}(s,0) - f_{i,t}(s,0))$$

$\square$

**Corollary 1.** *Let $\tilde{\pi} := \min(\pi_{\min}, 1 - \pi_{\max})$ and $\sigma^2 := Var(\epsilon_{i,t})$. Then*

$$Var\left(\tilde{R}_{i,t}^f(s,\bar{a})\right) \le \frac{2\sigma^2 + 4M^2}{\tilde{\pi}} + 8M^2 =: v_1^2.$$

*Proof.* Substitute $\sigma^2 := \text{Var}(\epsilon_{i,t})$ in Lemma 5 and apply the bounds $M$, $\pi_{\min}$, and $\pi_{\max}$ to obtain

$$\text{Var}\left(\tilde{R}_{i,t}^f(s,\bar{a})\right) \le \frac{4M^2 \pi_{i,t}(0|s) + \sigma^2}{\tilde{\pi}} + \frac{4M^2\{1 - \pi_{i,t}(0|s)\} + \sigma^2}{\tilde{\pi}} + 8M^2$$

$$= \frac{2\sigma^2 + 4M^2}{\tilde{\pi}} + 8M^2.$$

$\square$

We now show that Assumption 3 results in a lower asymptotic variance bound.

**Corollary 2.** *The asymptotic variance of $\tilde{R}_{i,t}^f(s,\bar{a})$ is no greater than*

$$v_2^2 := \frac{\sigma^2}{\tilde{\pi}(1 - \tilde{\pi})}.$$

*Proof.* By Assumption 3, the terms involving $r_{i,t}(s,\bar{a}) - f_{i,t}(s,\bar{a})$ and $r_{i,t}(s,0) - f_{i,t}(s,0)$ converge almost surely to zero. Dropping these terms from the expression in Lemma 5, the asymptotic variance is then less than or equal to

$$\frac{\sigma^2}{1 - \pi_{i,t}(0|s)} + \frac{\sigma^2}{\pi_{i,t}(0|s)} = \frac{\sigma^2}{\pi_{i,t}(0|s)\{1 - \pi_{i,t}(0|s)\}} \le \frac{\sigma^2}{\tilde{\pi}(1 - \tilde{\pi})}.$$

$\square$

**Remark 1.** *Comparing $v_1^2$ and $v_2^2$, we see that $v_1^2 \ge v_2^2$ if*

$$\frac{2\sigma^2}{\tilde{\pi}} \ge \frac{\sigma^2}{\tilde{\pi}(1 - \tilde{\pi})} \iff 1 - \tilde{\pi} \ge 1/2 \iff 1/2 \ge \tilde{\pi},$$

*which must hold by the definition of $\tilde{\pi}$. Consequently, The effect of the DML is to lower the variance bound.*

### E.3 Derivation of Confidence Sets

To facilitate the developments below, we define the following quantities:

- $V_{0,k}, V_{i,t}, b_{i,t}, \hat{\theta}_{i,t}$: The quantities $V_0, V, b, \hat{\theta} := V^{-1}b$ in Algorithms 1 and 2 at decision point $(i, t)$.

- $\mathcal{O}_{i,t}$: The set of user–time pairs up to but excluding $(i, t)$; we can express $\mathcal{O}_{i,t}$ as $\{(1,1),(1,2),(2,1),\ldots,(i,t)\}\backslash\{(i,t)\}$

- $\theta_{i,t}^{\star} := \theta^{\text{shared}} + \theta_i^{\text{user}} + \theta_t^{\text{time}} = C_{i,t}\theta_k^{\star} \in \mathbb{R}^d$.

- $\check{\theta}_{i,t} := C_{i,t}\hat{\theta}_{i,t} \in \mathbb{R}^d$.

- $\Lambda_{i,t}^0 := C_{i,t}V_{i,t}^{-1}V_{0,k}V_{i,t}^{-1}C_{i,t}^{\top} \in \mathbb{R}^{(d \times d)}$.

- $\Lambda_{i,t}^+ := C_{i,t}V_{i,t}^{-1}\left[\sum_{(j,u)\in\mathcal{O}_{i,t}}\tilde{\sigma}_{j,u}^2\phi\{x(S_{ju}, A_{ju})\}\phi\{x(S_{ju}, A_{ju})\}^{\top}\right]V_{i,t}^{-1}C_{i,t}^{\top} \in \mathbb{R}^{(d \times d)}$.

- $\underline{V}_{i,t} := (\Lambda_{i,t}^0 + \Lambda_{i,t}^+)^{-1} = (C_{i,t}V_{i,t}^{-1}C_{i,t}^{\top})^{-1} \in \mathbb{R}^{d \times d}$.

In the proof below, we allow the dimension of $\theta_k^{\star}$ to increase as stages progress so that $\theta_k^{\star}, b_{i,t}, \hat{\theta}_{i,t} \in \mathbb{R}^{(2k+1)d}$ and $V_{0,k}, V_{i,t} \in \mathbb{R}^{(2k+1)d \times (2k+1)d}$. However, the proof is also applicable (with minor modifications) to settings where the final dimension of $\theta_k^{\star}$ is known and set in advance (as in Algorithm 2).

In analogy to Bayesian linear regression, $\underline{V}_{i,t}$ plays the role of a posterior precision matrix for $\theta_{i,t}^{\star}$. Similarly, $\underline{V}_{i,t}^{-1}$ plays the role of a posterior covariance matrix. Assumption 9 ensures that $\underline{V}_{i,t}^{-1}$ is 'decreasing' sufficiently fast. Proving that this assumption holds under more primitive conditions is an interesting area for future work. We conjecture that the following condition is sufficient provided that $\gamma$ is set to a sufficiently large constant.

**Assumption 10.** *Let $\mathcal{O}$ be a set of $(i, t)$ pairs containing at least $K^{\star} \in \mathbb{N}$ elements, and let $\omega > 0$. Then the following ordering holds:*

$$\sum_{(i,t)\in\mathcal{O}} x_{i,t}x_{i,t}^{\top} \succ (\#\mathcal{O})\omega I_d,$$

*where $\#\mathcal{O}$ denotes the cardinality of $\mathcal{O}$.*

This assumption ensures that we continue acquiring information about all elements of $x_{i,t}$ at a linear rate as we progress through stages. We now adapt the classical theory on RLS estimation error to our setting.

**Lemma 6** (Adapted from Theorem 2 in Abbasi-Yadkori et al. (2011)). *Let $\check{\theta}_{i,t}$ be the regularized least squares (RLS) estimate from Algorithm 1 for the $i$-th user at the $t$-th time point, corresponding to stage $k := i + t - 1$ and let $\theta_{i,t}^{\star}$ be the true parameter value. For any $\delta > 0$, with probability at least $1 - \delta$ the estimates $\{\check{\theta}_{i,t}\}_{(i,t)\in\mathcal{O}_K}$ satisfy*

$$\|\check{\theta}_{i,t} - \theta_{i,t}^{\star}\|_{\underline{V}_{i,t}} \leq v\sqrt{2\log\left\{\frac{1}{\delta}\cdot\sqrt{\frac{\det(\underline{V}_{i,t}^{-1})}{\det(\Lambda_{i,t}^0)}}\right\}} + \|C_{i,t}V_{i,t}^{-1}V_{0,k}\theta_k^{\star}\|_{\underline{V}_{i,t}},$$

*where $v^2$ is the variance proxy for the difference between the pseudo-reward and its mean.*

*Proof.* Let $m_{i,t} = \tilde{\sigma}_{i,t}\phi\{x(S_{i,t}, A_{i,t})\}$ and $\rho_{i,t} = \tilde{\sigma}_{i,t}\{\tilde{R}_{i,t}^f(S_{i,t}, \bar{A}_{i,t}) - \mathbb{E}(\tilde{R}_{i,t}^f|S_{i,t}, \bar{A}_{i,t})\}$. Further let

$$\xi_{i,t} := \sum_{(j,u)\in\mathcal{O}_{i,t}} m_{j,u}\rho_{j,u} \in \mathbb{R}^{(2k+1)d}.$$

$$\hat{\theta}_{i,t} = V_{i,t}^{-1} b_{i,t}$$
$$= V_{i,t}^{-1}(\xi_{i,t} + \Phi_{i,t}^{\top} W_{i,t} \Delta_{i,t})$$
$$= V_{i,t}^{-1}\xi_{i,t} + V_{i,t}^{-1}\Phi_{i,t}^{\top} W_{i,t}\Phi_{i,t}\theta_k^{\star}$$
$$= V_{i,t}^{-1}\xi_{i,t} + V_{i,t}^{-1}(\Phi_{i,t}^{\top} W_{i,t}\Phi_{i,t} + V_{0,k})\theta_k^{\star} - V_{i,t}^{-1}V_{0,k}\theta_k^{\star}$$
$$= V_{i,t}^{-1}\xi_{i,t} + \theta_k^{\star} - V_{i,t}^{-1}V_{0,k}\theta_k^{\star}$$

and thus

$$\hat{\theta}_{i,t} - \theta_k^{\star} = V_{i,t}^{-1}(\xi_{i,t} - V_{0,k}\theta_k^{\star}).$$

Premultiplying, we can then obtain

$$(\hat{\theta}_{i,t} - \theta_k^{\star})^{\top} C_{i,t}^{\top}\underline{V}_{i,t}C_{i,t}(\hat{\theta}_{i,t} - \theta_k^{\star}) = (\hat{\theta}_{i,t} - \theta_k^{\star})^{\top}C_{i,t}^{\top}\underline{V}_{i,t}C_{i,t}V_{i,t}^{-1}(\xi_{i,t} - V_{0,k}\theta_k^{\star}).$$

Simplification yields

$$\|\check{\theta}_{i,t} - \theta_{i,t}^{\star}\|_{\underline{V}_{i,t}}^2 = (\check{\theta}_{i,t} - \theta_{i,t}^{\star})^{\top}\underline{V}_{i,t}C_{i,t}V_{i,t}^{-1}(\xi_{i,t} - V_{0,k}\theta_k^{\star}).$$

We can then apply the Cauchy-Schwarz inequality to bound the right-hand side.

$$\|\check{\theta}_{i,t} - \theta_{i,t}^{\star}\|_{\underline{V}_{i,t}}^2 \leq \|\check{\theta}_{i,t} - \theta_{i,t}^{\star}\|_{\underline{V}_{i,t}} \cdot \|C_{i,t}V_{i,t}^{-1}(\xi_{i,t} - V_{0,k}\theta_k^{\star})\|_{\underline{V}_{i,t}}$$

Dividing by $\|\check{\theta}_{i,t} - \theta_{i,t}^{\star}\|_{\underline{V}_{i,t}}$ and applying the triangle inequality gives

$$\|\check{\theta}_{i,t} - \theta_{i,t}^{\star}\|_{\underline{V}_{i,t}} \leq \|C_{i,t}V_{i,t}^{-1}\xi_{i,t}\|_{\underline{V}_{i,t}} + \|C_{i,t}V_{i,t}^{-1}V_{0,k}\theta_k^{\star}\|_{\underline{V}_{i,t}}$$

We can show that

$$(\Lambda_{i,t}^0 + \Lambda_{i,t}^+)^{-1}$$
$$= \left\{ C_{i,t}V_{i,t}^{-1}V_{0,k}V_{i,t}^{-1}C_{i,t}^{\top} + \sum_{(j,u)\in\mathcal{O}_{i,t}} C_{i,t}V_{i,t}^{-1}m_{j,u}m_{j,u}^{\top}V_{i,t}^{-1}C_{i,t}^{\top} \right\}^{-1}$$
$$= \left\{ C_{i,t}V_{i,t}^{-1}\left( V_{0,k} + \sum_{(j,u)\in\mathcal{O}_{i,t}} \tilde{\sigma}_{j,u}^2\phi_{j,u}\phi_{j,u}^{\top} \right) V_{i,t}^{-1}C_{i,t}^{\top} \right\}^{-1}$$
$$= \left( C_{i,t}V_{i,t}^{-1}V_{i,t}V_{i,t}^{-1}C_{i,t}^{\top} \right)^{-1}$$
$$= \left( C_{i,t}V_{i,t}^{-1}C_{i,t}^{\top} \right)^{-1}$$
$$= \underline{V}_{i,t}.$$

Combined with the fact that $\rho_{i,t}$ is sub-Gaussian with variance proxy $v^2$, the above result allows us to apply Theorem 1 of Abbasi-Yadkori et al. (2011). Specifically, to match their notation (setting aside our definition of $S_{i,t}$ for the moment), let $S_{i,t} = C_{i,t}V_{i,t}^{-1}\xi_{i,t} = \sum_{(j,u)\in\mathcal{O}_{i,t}} C_{i,t}V_{i,t}^{-1}\rho_{j,u}m_{j,u}$ and $\bar{V}_{i,t} = C_{i,t}V_{i,t}^{-1}V_{0,k}V_{i,t}^{-1}C_{i,t}^{\top} + C_{i,t}V_{i,t}^{-1}\left( \sum_{(j,u)\in\mathcal{O}_{i,t}} m_{j,u}m_{j,u}^{\top} \right) V_{i,t}^{-1}C_{i,t}^{\top}$, we have

$$\|S_{i,t}\|_{\bar{V}_{i,t}}^2 \leq 2v^2 \log\left( \frac{\det(\underline{V}_{i,t}^{-1})^{1/2}}{\delta \det(\Lambda_{i,t}^0)^{1/2}} \right)$$

$$\|C_{i,t}V_{i,t}^{-1}\xi_{i,t}\|_{\underline{V}_{i,t}} \leq v\sqrt{2\log\left\{ \frac{1}{\delta} \cdot \sqrt{\frac{\det(\underline{V}_{i,t}^{-1})}{\det(\Lambda_{i,t}^0)}} \right\}}$$

Adding this bound to $\|C_{i,t}V_{i,t}^{-1}V_{0,k}\theta_k^{\star}\|_{\underline{V}_{i,t}}$ completes the proof.

$\square$

In the above inequality, we will need to bound $\|C_{i,t}V_{i,t}^{-1}V_{0,k}\theta_k^\star\|_{\underline{V}_{i,t}}$ in order to obtain useful confidence sets. To do so, we first define a new matrix, $\underline{C}_{i,t}$.

**Definition 1.** *Let $\underline{C}_{i,t} \in \mathbb{R}^{3d \times (2k+1)d}$ be a $3 \times (2k+1)$ block matrix where all blocks are $d \times d$ zero matrices except the blocks at entries $(1,1)$, $(2, 1+i)$, and $(3, 1+k+t)$ which are equal to $I_d$.*

Note that $\underline{C}_{i,t}$ selects the components of $\theta_k^\star$ that are relevant to $(i,t)$: $\theta^{\text{shared}}$, $\theta_i^{\text{user}}$, and $\theta_t^{\text{time}}$. We also have the relationship $C_{i,t} = (1_3^\top \otimes I_d)\underline{C}_{i,t}$, which we leverage below.

**Lemma 7.**

$$\|C_{i,t}V_{i,t}^{-1}V_{0,k}\theta_k^\star\|_{\underline{V}_{i,t}} \leq \|V_{0,k}\theta_k^\star\|_{V_{i,t}^{-1}}.$$

*Proof.* To simplify the proof, we reorder the rows and columns of $V_{i,t}$ such that the rows and columns corresponding to $\theta^{\text{shared}}$, $\theta_i^{\text{user}}$, and $\theta_t^{\text{time}}$ appear first. The matrices $\underline{C}_{i,t}$ and $C_{i,t}$ are similarly reordered, resulting in $\underline{C}_{i,t} = [I_{3d} \ 0_{3d,2d(k-1)}]$, $C_{i,t} = [(1_3^\top \otimes I_d) \ 0_{p,2d(k-1)}]$. We now partition $V_{i,t}$ as follows:

$$V_{i,t} = \begin{bmatrix} Z_{11} & Z_{12} \\ Z_{21} & Z_{22}. \end{bmatrix}$$

Using the $2 \times 2$ block-matrix inversion identity, it then follows that

$$V_{i,t}^{-1} = \begin{bmatrix} P^{-1} & -P^{-1}Z_{12}Z_{22}^{-1} \\ -Z_{22}^{-1}Z_{21}P^{-1} & Z_{22}^{-1} + Z_{22}^{-1}Z_{21}P^{-1}Z_{12}Z_{22}^{-1} \end{bmatrix},$$

where $P := Z_{11} - Z_{12}Z_{22}^{-1}Z_{21}$.

Consequently, we have $\underline{C}_{i,t}V_{i,t}^{-1} = P^{-1}\begin{bmatrix} I_{3d} & -Z_{12}Z_{22}^{-1} \end{bmatrix}$ and $\underline{C}_{i,t}V_{i,t}^{-1}\underline{C}_{i,t}^\top = P^{-1}$. Letting $G = [I_{3d} \ -Z_{12}Z_{22}^{-1}]$, we can write the squared norm as

$$\|C_{i,t}V_{i,t}^{-1}V_{0,k}\theta_k^\star\|_{\underline{V}_{i,t}}^2$$
$$= \theta_k^{\star\top}V_{0,k}G^\top P^{-1}(1_3 \otimes I_d)\{(1_3^\top \otimes I_d)P^{-1}(1_3 \otimes I_d)\}^{-1}(1_3^\top \otimes I_d)P^{-1}GV_{0,k}\theta_k^\star$$
$$\leq \theta_k^{\star\top}V_{0,k}G^\top P^{-1}GV_{0,k}\theta_k^\star,$$

where we have used the fact that $P^{-1/2}(1_3 \otimes I_d)\{(1_3^\top \otimes I_d)P^{-1}(1_3 \otimes I_d)\}^{-1}(1_3^\top \otimes I_d)P^{-1/2} \preceq I_d$ because the former is a projection matrix and thus its maximum eigenvalue is one. We can further simplify and bound this quantity as follows:

$$\theta_k^{\star\top}V_{0,k}G^\top P^{-1}GV_{0,k}\theta_k^\star$$
$$= \theta_k^{\star\top}V_{0,k}\begin{bmatrix} I_{3d} \\ -Z_{22}^{-1}Z_{21} \end{bmatrix} P^{-1} \begin{bmatrix} I_{3d} & -Z_{12}Z_{22}^{-1} \end{bmatrix} V_{0,k}\theta_k^\star$$
$$= \theta_k^{\star\top}V_{0,k}\begin{bmatrix} P^{-1} & -P^{-1}Z_{12}Z_{22}^{-1} \\ Z_{22}^{-1}Z_{21}P^{-1} & Z_{22}^{-1}Z_{21}P^{-1}Z_{12}Z_{22}^{-1} \end{bmatrix} V_{0,k}\theta_k^\star$$
$$= \theta_k^{\star\top}V_{0,k}\left(V_{i,t}^{-1} - \begin{bmatrix} 0 & 0 \\ 0 & Z_{22}^{-1} \end{bmatrix}\right) V_{0,k}\theta_k^\star$$
$$\leq \|V_{0,k}\theta_k^\star\|_{V_{i,t}^{-1}}^2.$$

$\square$

**Remark 2.** *We can apply Lemma 7 and Assumption 8 to bound $\|C_{i,t}V_{i,t}^{-1}V_{0,k}\theta_k^\star\|_{\underline{V}_{i,t}}$. Assumption 8 implies that there exists a function $\zeta(\delta) : (0,1) \to \mathbb{R}$ such that*

$$\sup_{(i,t) \in \mathcal{O}_K} \|C_{i,t}V_{i,t}^{-1}V_{0,k}\theta_k^\star\|_{\underline{V}_{i,t}} \leq \sup_{(i,t) \in \mathcal{O}_K} \|V_{0,k}\theta_k^\star\|_{V_{i,t}^{-1}} \leq \zeta(\delta)\max\{\log^{3/4}(K), 1\}$$

*with probability at least $1 - \delta$. This function corresponds with the hyperparameter $\zeta$ given in the main paper. The purpose of the $\max$ operation is to avoid having a bound of zero for $K = 1$.*

We confirmed in simulation that Assumption 8 assumption is reasonable. The left panel of Figure 13 provides empirical evidence that $\|V_{0,k}\theta_k^\star\|_{V_{i,t}^{-1}}/\log^{3/4}(k)$ is bounded in probability in a simulation with values of $k$ ranging from 1 to 200. We ran 400 repetitions with an intercept-only model. Similarly, the right panel suggests that $\sup_{(i,t)\in\mathcal{O}_K}\|V_{0,k}\theta_k^\star\|_{V_{i,t}^{-1}}/\log^{3/4}(K)$ is bounded as assumed. We conjecture that the following assumption is a sufficient condition (in combination with the other assumptions listed in the main paper) for Assumption 8 to hold.

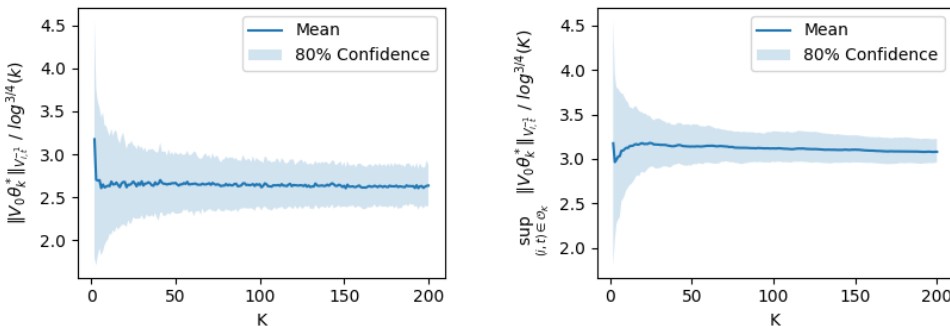

Figure 13: (left) Simulated values of $\|V_{0,k}\theta_k^\star\|_{V_{i,t}^{-1}}/\log^{3/4}(k)$. (right) Simulated values of $\sup_{(i,t)\in\mathcal{O}_K}\|V_{0,k}\theta_k^\star\|_{V_{i,t}^{-1}}/\log^{3/4}(K)$. The plots show evidence that these terms are bounded as required by Assumption 8.

**Assumption 11.** *The random effects $\theta_i^{user}$, $\theta_t^{time}$ for $i, t = 1, 2, \ldots$ are distributed such that*

$$\frac{1}{\sqrt{K}}\sum_{k=1}^K \theta_k^{user} O_p(1), \quad \frac{1}{\sqrt{K}}\sum_{k=1}^K \theta_k^{time} = O_p(1).$$

So, in other words, it would be sufficient for the normalized sums above to converge in distribution. This effectively requires these normalized sums to be approximately mean zero. This requirement is reasonable given the presence of $\theta^{shared}$ in the model.

We now bound the first term from Lemma 6.

**Lemma 8.** *Assuming $\gamma \geq 1$, we have:*

$$\det(\underline{V}_{i,t}^{-1}) \leq \left(\frac{3}{\gamma}\right)^d$$

*Proof.* By definition, we have:
$$\underline{V}_{i,t}^{-1} = C_{i,t}V_{i,t}^{-1}C_{i,t}^\top$$

Taking the determinant of both sides:
$$\det(\underline{V}_{i,t}^{-1}) = \det(C_{i,t}V_{i,t}^{-1}C_{i,t}^\top)$$

Using the property that the determinant of a matrix is less than or equal to the product of its eigenvalues, and letting $\lambda_j(\cdot)$ denote the $j$-th largest eigenvalue, we have:

$$\det(\underline{V}_{i,t}^{-1}) \leq \prod_{j=1}^d \lambda_j(C_{i,t}V_{i,t}^{-1}C_{i,t}^\top)$$
$$\leq \lambda_{\max}(C_{i,t}V_{i,t}^{-1}C_{i,t}^\top)^d$$

Now letting $\sigma_1$ be the largest singular value of a matrix,
$$\lambda_{\max}(C_{i,t}V_{i,t}^{-1}C_{i,t}^\top) \leq \sigma_1(C_{i,t})^2\lambda_{\max}(V_{i,t}^{-1})$$

Since $C_{i,t} = (1_3^\top \otimes I_d)\underline{C}_{i,t}$, we have that

$$
\begin{aligned}
\sigma_1(C_{i,t}) &= \sigma_1((1_3^\top \otimes I_d)\underline{C}_{i,t}) \\
&\leq \sigma_1(1_3^\top \otimes I_d)\sigma_1(\underline{C}_{i,t}) \\
&= \sigma_1(1_3^\top)\sigma_1(I_d)\sigma_1(\underline{C}_{i,t}) \\
&= \sqrt{3} \cdot 1 \cdot 1 \\
&= \sqrt{3}
\end{aligned}
$$

so that $\sigma_1(C_{i,t})^2 = 3$. Therefore,

$$
\begin{aligned}
\lambda_{\max}(C_{i,t}V_{i,t}^{-1}C_{i,t}^\top) &\leq 3\lambda_{\max}(V_{i,t}^{-1}) \\
&= \frac{3}{\lambda_{\min}(V_{i,t})} \\
&\leq \frac{3}{\gamma}
\end{aligned}
$$

and thus

$$
\det(\underline{V}_{i,t}^{-1}) \leq \left(\frac{3}{\gamma}\right)^d.
$$

$\square$

**Lemma 9.**

$$
det(\Lambda_{i,t}^0) \geq 3^d \left(\frac{\lambda_{\min}(V_{0,k})}{\lambda_{\max}(V_{i,t})^2}\right)^d
$$

*Proof.*

$$
\begin{aligned}
\det(\Lambda_{i,t}^0) &\geq \prod_{j=1}^d \sigma_{\min}(1_3^\top \otimes I_d)^2 \lambda_{\min}(C_{i,t}V_{i,t}^{-1}V_{0,k}V_{i,t}^{-1}C_{i,t}^\top) \\
&\geq 3^d \lambda_{\min}(V_{i,t}^{-1}V_{0,k}V_{i,t}^{-1})^d \\
&\geq 3^d \left(\frac{\lambda_{\min}(V_{0,k})}{\lambda_{\max}(V_{i,t})^2}\right)^d
\end{aligned}
$$

$\square$

**Lemma 10.** *Assuming $\gamma \geq 1$,*

$$
\log \frac{\det(\underline{V}_{i,t}^{-1})}{\det(\Lambda_{i,t}^0)} \leq 2d \log \left\{\frac{3k(k+1)}{8} + \gamma k + 2\lambda e_k\right\}.
$$

*Proof.*

$$
\begin{aligned}
\log \frac{\det(\underline{V}_{i,t}^{-1})}{\det(\Lambda_{i,t}^0)} &\leq d \log \frac{3}{\gamma} - d\log 3 + 2d\log \lambda_{\max}(V_{i,t}) - d\log \lambda_{\min}(V_{0,k}) \\
&\leq 2d(\log \lambda_{\max}(V_{i,t}) - \log \gamma)
\end{aligned}
$$

Now we need to upper bound $\lambda_{\max}(V_{i,t})$. Note that

$$
\begin{aligned}
\lambda_{\max}(V_{i,t}) &\leq \operatorname{tr}(V_{i,t}) \\
&\leq \operatorname{tr}\left(V_{0,k} + \sum_{(i,t)\in\mathcal{O}_K} \tilde{\sigma}_{i,t}^2 \phi(x_{i,t})\phi(x_{i,t})^\top\right) \\
&\leq \sum_{(i,t)\in\mathcal{O}_K} \tilde{\sigma}_{i,t}^2 \|\phi(x_{i,t})\|_2^2 + \gamma k + 2\lambda e_k \\
&\leq \frac{3}{4}\frac{k(k+1)}{2} + \gamma k + 2\lambda e_k.
\end{aligned}
$$

This then gives us

$$\log \frac{\det(\underline{V}_{i,t}^{-1})}{\det(\Lambda_{i,t}^0)} \leq 2d \left( \log \frac{3k(k+1)}{8} + \gamma k + 2\lambda e_k - \log \gamma \right)$$

$$\leq 2d \log \left\{ \frac{3k(k+1)}{8} + \gamma k + 2\lambda e_k \right\}$$

$\square$

Note that $e_k$ produces no more than a logarithmic contribution to the above bound due to Assumption 7. We now define a term, $\beta_{i,t}(\delta_1, \delta_2)$, that corresponds with the term $\beta_{i,t}(\delta)$ from the main paper, setting $\delta = 2\delta_1$ with sufficiently large $\zeta$. However, here we make explicit that the term depends on two probabilities, $\delta_1$ and $\delta_2$.

**Definition 2.** *Let $\beta_{i,t}(\delta_1, \delta_2)$ be defined as follows:*

$$\beta_{i,t}(\delta_1, \delta_2) := v \sqrt{2 \log \left( \frac{1}{\delta_1} \right) + \log \left\{ \frac{\det(\underline{V}_{i,t}^{-1})}{\det(\Lambda_{i,t}^0)} \right\}} + \zeta(\delta_2) \max\{\log^{3/4}(K), 1\}.$$

Note that by Lemma 6 and Assumption 8 we have $\|\check{\theta}_{i,t} - \theta_{i,t}^\star\|_{\underline{V}_{i,t}} \leq \beta_{i,t}(\delta_1, \delta_2)$ with probability at least $1 - \delta_1 - \delta_2$.

**Lemma 11.** *Let $\delta_1' = \delta_1/\{K(K+1)\}$. Then*

$$\sup_{(i,t) \in \mathcal{O}_K} \|\check{\theta}_{i,t} - \theta_{i,t}^\star\|_{\underline{V}_{i,t}} \leq \sup_{(i,t) \in \mathcal{O}_K} \beta_{i,t}(\delta_1', \delta_2)$$

*with probability at least $1 - \delta_1/2 - \delta_2$. Further, $\sup_{(i,t) \in \mathcal{O}_K} \beta_{i,t}(\delta_1', \delta_2) \leq \beta_K(\delta_1, \delta_2)$, where $\beta_K(\delta_1, \delta_2)$ is defined as*

$$v \sqrt{2 \log \left\{ \frac{K(K+1)}{\delta_1} \right\} + 2d \log \left\{ \frac{3K(K+1)}{8} + \gamma K + 2\lambda e_K \right\}} + \zeta(\delta_2) \max\{\log^{3/4}(K), 1\}.$$

*Proof.* The first statement follows from applying Lemma 6 with a union bound. Note that the second term in Lemma 6 can be bounded uniformly using Lemma 7 and Assumption 8 as follows:

$$\|C_{i,t} V_{i,t}^{-1} V_{0,k} \theta_k^\star\|_{\underline{V}_{i,t}} \leq \|V_{0,k} \theta_k^\star\|_{V_{i,t}^{-1}} \leq \zeta(\delta_2) \max\{\log^{3/4}(K), 1\}$$

with probability at least $1 - \delta_2$. The second statement follows from bounding the first term in $\beta_{i,t}(\delta_1', \delta_2)$ using Lemma 10:

$$v \sqrt{2 \log \left\{ \frac{K(K+1)}{\delta_1} \right\} + \log \frac{\det(\underline{V}_{i,t}^{-1})}{\det(\Lambda_{i,t}^0)}}$$

$$\leq v \sqrt{2 \log \left\{ \frac{K(K+1)}{\delta_1} \right\} + 2d \log \left\{ \frac{3K(K+1)}{8} + \gamma K + 2\lambda e_K \right\}}.$$

$\square$

Lastly, we define several events and bound the probabilities with which they occur.

**Proposition 1.** *Let*

- $\delta_1' = \delta_1/\{K(K+1)\}$

- $\hat{E}_K = \{\forall(i,t) \in \mathcal{O}_K, \|\check{\theta}_{i,t} - \theta_{i,t}^\star\|_{\underline{V}_{i,t}} \leq \beta_{i,t}(\delta_1', \delta_2)\}$

- $\gamma_{i,t}(\delta_1, \delta_2) = \beta_{i,t}(\delta_1, \delta_2) \sqrt{cd \log(c'd/\delta_1)}$

- $\tilde{E}_K = \{\forall(i,t) \in \mathcal{O}_K, \|\tilde{\theta}_{i,t} - \check{\theta}_{i,t}\|_{\underline{V}_{i,t}} \leq \gamma_{i,t}(\delta_1', \delta_2)\}$

- $E_K = \hat{E}_K \cap \tilde{E}_K$

*Then* $\Pr(E_K) \geq 1 - \delta_1 - \delta_2$.

*Proof.* First, Lemma 11 shows that $\Pr(\hat{E}_k) \geq 1 - \delta_1/2 - \delta_2$.

Next, applying the TS sampling distribution and $\tilde{\theta}_{i,t} = \check{\theta}_{i,t} + \beta_k(\delta_1', \delta_2)\underline{V}_{i,t}^{-1/2}\eta_{i,t}$ where $\eta_{i,t}$ is drawn i.i.d. from $\mathcal{D}^{TS}$ we have

$$\Pr\left(\|\tilde{\theta}_{i,t} - \check{\theta}_{i,t}\|_{\underline{V}_{i,t}} \leq \beta_{i,t}(\delta_1', \delta_2)\sqrt{cd\log\left(\frac{c'd}{\delta_1'}\right)}\right)$$

$$= \Pr\left(\|\eta_{i,t}\| \leq \sqrt{cd\log\left(\frac{c'd}{\delta_1'}\right)}\right)$$

$$\geq 1 - \delta_1'$$

by the concentration property in Definition 1. A union bound then gives $\Pr(\tilde{E}_K) \geq 1 - \delta_1/2$. Applying a union bound to $\tilde{E}_K$ and $\hat{E}_K$ yields the conclusion. $\qquad\square$

## E.4 Treatment Probability Estimation Errors

The proof relies on bounding terms of the form $\{\pi_{i,t}(\bar{A}_{i,t}|S_{i,t}) - \pi_{i,t}^{\star}(\bar{A}_{i,t}^{\star}|S_{i,t})\}x(S_{i,t}, \bar{A}_{i,t})^{\top}\theta_{i,t}^{\star}$. To simplify the proof, we denote these terms as $(\pi_{i,t} - \pi_{i,t}^{\star})x_{i,t}^{\top}\theta_{i,t}^{\star}$. We begin by defining two filtrations.

**Definition 2.** *We define the filtration $\mathcal{F}_{i,t}^{S}$ as the information accumulated up to decision point $(i,t)$ including the sampled context, that is, $\mathcal{F}_{i,t}^{S} = (\mathcal{F}_1, \sigma(S_{1,1}, \bar{A}_{1,1}, A_{1,1}, R_{1,1}S_{2,1}, \ldots, S_{i,t}))$. Similarly, we define the filtration $\mathcal{F}_{i,t}$ as the information accumulated up to decision point $(i,t)$ before sampling $S_{i,t}$. Thus, $\mathcal{F}_{i,t}$ can be expressed in the same form as that of $\mathcal{F}_{i,t}^{S}$ above, removing the term $S_{i,t}$.*

We now leverage $\mathcal{F}_{i,t}^{S}$ to conditionally bound $(\pi_{i,t} - \pi_{i,t}^{\star})x_{i,t}^{\top}\theta_{i,t}^{\star}$.

**Lemma 12.** *Conditional on $\mathcal{F}_{i,t}^{S}$, the term $(\pi_{i,t} - \pi_{i,t}^{\star})x_{i,t}^{\top}\theta_{i,t}^{\star}$ is nonnegative and bounded as*

$$(\pi_{i,t} - \pi_{i,t}^{\star})x_{i,t}^{\top}\theta_{i,t}^{\star} < \sqrt{\frac{\|\check{\theta}_{i,t} - \theta_{i,t}^{\star}\|_{\underline{V}_{i,t}}^2 + \beta_{i,t}^2(\delta_1, \delta_2)\mathbb{E}\|\eta_{i,t}\|^2}{\min(\pi_{\min}, 1 - \pi_{\max})\alpha\min(i,t)}}.$$

*Proof.* We begin by conditioning on $\mathcal{F}_{i,t}^{S}$ so that $x_{i,t}^{\top}\theta_{i,t}^{\star}$ is nonrandom. There are three cases.

**Case 1:** $x_{i,t}^{\top}\theta_{i,t}^{\star} = 0$.

In this simple case, we have $(\pi_{i,t} - \pi_{i,t}^{\star})x_{i,t}^{\top}\theta_{i,t}^{\star} = 0$.

**Case 2:** $x_{i,t}^{\top}\theta_{i,t}^{\star} > 0$.

In this case, the optimal policy is to apply the control arm with probability $\pi_{i,t}^{\star} = \pi_{\min}$ because the treatment arm has a positive expected effect on the reward. Because $\pi_{i,t} \in [\pi_{\min}, \pi_{\max}]$ by construction, we have $\pi_{i,t} \geq \pi_{\min} = \pi_{i,t}^{\star}$, which implies that $\pi_{i,t} - \pi_{i,t}^{\star} \geq 0$, proving the first statement in the proof for this case.

Now the probability $\pi_{i,t}$ is based on the following probability calculation:

$$
\Pr\{x_{i,t}^\top \tilde{\theta}_{i,t} < 0 \,|\, \mathcal{F}_{i,t}^S\}
$$

$$
= \Pr\left[x_{i,t}^\top \left\{\theta_{i,t}^\star + (\check{\theta}_{i,t} - \theta_{i,t}^\star) + \beta_{i,t}(\delta_1, \delta_2)\underline{V}_{i,t}^{-1/2}\eta_{i,t}\right\} < 0 \,\Big|\, \mathcal{F}_{i,t}^S\right]
$$

$$
= \Pr\left[x_{i,t}^\top \left\{(\check{\theta}_{i,t} - \theta_{i,t}^\star) + \beta_{i,t}(\delta_1, \delta_2)\underline{V}_{i,t}^{-1/2}\eta_{i,t}\right\} < -x_{i,t}^\top \theta_{i,t}^\star \,\Big|\, \mathcal{F}_{i,t}^S\right]
$$

$$
\leq \Pr\left[\left|x_{i,t}^\top \left\{(\check{\theta}_{i,t} - \theta_{i,t}^\star) + \beta_{i,t}(\delta_1, \delta_2)\underline{V}_{i,t}^{-1/2}\eta_{i,t}\right\}\right| \geq x_{i,t}^\top \theta_{i,t}^\star \,\Big|\, \mathcal{F}_{i,t}^S\right]
$$

$$
\leq \Pr\left[\|x_{i,t}\|_{\underline{V}_{i,t}^{-1}} \left\|\left\{(\check{\theta}_{i,t} - \theta_{i,t}^\star) + \beta_{i,t}(\delta_1, \delta_2)\underline{V}_{i,t}^{-1/2}\eta_{i,t}\right\}\right\|_{\underline{V}_{i,t}} \geq x_{i,t}^\top \theta_{i,t}^\star \,\Big|\, \mathcal{F}_{i,t}^S\right]
$$

$$
= \Pr\left[\left\|(\check{\theta}_{i,t} - \theta_{i,t}^\star) + \beta_{i,t}(\delta_1, \delta_2)\underline{V}_{i,t}^{-1/2}\eta_{i,t}\right\|_{\underline{V}_{i,t}} \geq \frac{x_{i,t}^\top \theta_{i,t}^\star}{\|x_{i,t}\|_{\underline{V}_{i,t}^{-1}}} \,\Big|\, \mathcal{F}_{i,t}^S\right]
$$

$$
\leq \mathbb{E}\left(\left\|(\check{\theta}_{i,t} - \theta_{i,t}^\star) + \beta_{i,t}(\delta_1, \delta_2)\underline{V}_{i,t}^{-1/2}\eta_{i,t}\right\|_{\underline{V}_{i,t}}^2 \,\Big|\, \mathcal{F}_{i,t}^S\right) \frac{\|x_{i,t}\|_{\underline{V}_{i,t}^{-1}}^2}{(x_{i,t}^\top \theta_{i,t}^\star)^2}
$$

$$
\leq \mathbb{E}\left(\left\|(\check{\theta}_{i,t} - \theta_{i,t}^\star) + \beta_{i,t}(\delta_1, \delta_2)\underline{V}_{i,t}^{-1/2}\eta_{i,t}\right\|_{\underline{V}_{i,t}}^2 \,\Big|\, \mathcal{F}_{i,t}^S\right) \frac{1}{(x_{i,t}^\top \theta_{i,t}^\star)^2\,\alpha\min(i,t)},
$$

where the last two lines follow from Markov's inequality and Assumption 9. Working directly with the expectation, we can show that

$$
\mathbb{E}\left(\left\|(\check{\theta}_{i,t} - \theta_{i,t}^\star) + \beta_{i,t}(\delta_1, \delta_2)\underline{V}_{i,t}^{-1/2}\eta_{i,t}\right\|_{\underline{V}_{i,t}}^2 \,\Big|\, \mathcal{F}_{i,t}^S\right)
$$

$$
\leq \mathbb{E}\left(\left\|\check{\theta}_{i,t} - \theta_{i,t}^\star\right\|_{\underline{V}_{i,t}}^2 \,\Big|\, \mathcal{F}_{i,t}^S\right) + \mathbb{E}\left(\left\|\beta_{i,t}(\delta_1, \delta_2)\underline{V}_{i,t}^{-1/2}\eta_{i,t}\right\|_{\underline{V}_{i,t}}^2 \,\Big|\, \mathcal{F}_{i,t}^S\right)
$$

$$
= \left\|\check{\theta}_{i,t} - \theta_{i,t}^\star\right\|_{\underline{V}_{i,t}}^2 + \beta_{i,t}^2(\delta_1, \delta_2)\mathbb{E}\left\|\eta_{i,t}\right\|^2.
$$

This result implies that

$$
\Pr\{x_{i,t}^\top \tilde{\theta}_{i,t} < 0\} \leq \frac{\left\|\check{\theta}_{i,t} - \theta_{i,t}^\star\right\|_{\underline{V}_{i,t}}^2 + \beta_{i,t}^2(\delta_1, \delta_2)\mathbb{E}\left\|\eta_{i,t}\right\|^2}{(x_{i,t}^\top \theta_{i,t}^\star)^2\alpha\min(i,t)} =: M_{i,t}.
$$

Provided $M_{i,t} \leq \pi_{\min}$, we have $(\pi_{i,t} - \pi_{i,t}^\star)x_{i,t}^\top \theta_{i,t}^\star = 0$ because $\pi_{i,t} = \pi_{i,t}^\star = \pi_{\min}$. Alternatively, when $M_{i,t} > \pi_{\min}$, we have

$$
x_{i,t}^\top \theta_{i,t}^\star < \sqrt{\frac{\left\|\check{\theta}_{i,t} - \theta_{i,t}^\star\right\|_{\underline{V}_{i,t}}^2 + \beta_{i,t}^2(\delta_1, \delta_2)\mathbb{E}\left\|\eta_{i,t}\right\|^2}{\pi_{\min}\alpha\min(i,t)}}.
$$

Thus, in both cases we have

$$
(\pi_{i,t} - \pi_{i,t}^\star)x_{i,t}^\top \theta_{i,t}^\star < \sqrt{\frac{\left\|\check{\theta}_{i,t} - \theta_{i,t}^\star\right\|_{\underline{V}_{i,t}}^2 + \beta_{i,t}^2(\delta_1, \delta_2)\mathbb{E}\left\|\eta_{i,t}\right\|^2}{\pi_{\min}\alpha\min(i,t)}}.
$$

**Case 3:**

This case is essentially identical to **Case 2**, except both $\pi_{i,t} - \pi_{i,t}^\star$ and $x_{i,t}^\top \theta_{i,t}^\star$ are negative. The resulting bound is

$$
(\pi_{i,t} - \pi_{i,t}^\star)x_{i,t}^\top \theta_{i,t}^\star < \sqrt{\frac{\left\|\check{\theta}_{i,t} - \theta_{i,t}^\star\right\|_{\underline{V}_{i,t}}^2 + \beta_{i,t}^2(\delta_1, \delta_2)\mathbb{E}\left\|\eta_{i,t}\right\|^2}{(1 - \pi_{\max})\alpha\min(i,t)}}.
$$

**Conclusion:**

Combining the three cases yields

$$(\pi_{i,t} - \pi^\star_{i,t})x^\top_{i,t}\theta^\star_{i,t} < \sqrt{\frac{\left\|\check{\theta}_{i,t} - \theta^\star_{i,t}\right\|^2_{\underline{V}_{i,t}} + \beta^2_{i,t}(\delta_1, \delta_2)\mathbb{E}\left\|\eta_{i,t}\right\|^2}{\min(\pi_{\min}, 1 - \pi_{\max})\alpha \min(i, t)}}.$$

$\square$

**Lemma 13.** *Under event* $E_K$, *the following bound holds for all* $(i, t) \in \mathcal{O}_{i,t}$:

$$(\pi_{i,t} - \pi^\star_{i,t})x^\top_{i,t}\theta^\star_{i,t} < \frac{\tau\beta_K(\delta_1, \delta_2)}{\sqrt{\min(i, t)}}, \quad \tau := \sqrt{\frac{1 + \mathbb{E}\left\|\eta_{i,t}\right\|^2}{\alpha\min(\pi_{\min}, 1 - \pi_{\max})}}.$$

*Proof.* Let $(i^\star, t^\star)$ denote the final decision point at stage $K$ and let $\delta'_1 = \delta_1/\{K(K+1)\}$. Conditional on $\mathcal{F}^S_{i^\star, t^\star}$, Lemma 12 implies that

$$(\pi_{i,t} - \pi^\star_{i,t})x^\top_{i,t}\theta^\star_{i,t} < \sqrt{\frac{\left\|\check{\theta}_{i,t} - \theta^\star_{i,t}\right\|^2_{\underline{V}_{i,t}} + \beta^2_{i,t}(\delta'_1, \delta_2)\mathbb{E}\left\|\eta_{i,t}\right\|^2}{\min(\pi_{\min}, 1 - \pi_{\max})\alpha \min(i, t)}}$$

for all $(i, t) \in \mathcal{O}_K$ because $\mathcal{F}^S_{i,t} \subseteq \mathcal{F}^S_{i^\star, t^\star}$. We can then apply Lemma 11 to argue that

$$(\pi_{i,t} - \pi^\star_{i,t})x^\top_{i,t}\theta^\star_{i,t} < \sqrt{\frac{\beta^2_K(\delta_1, \delta_2) + \beta^2_K(\delta_1, \delta_2)\mathbb{E}\left\|\eta_{i,t}\right\|^2}{\min(\pi_{\min}, 1 - \pi_{\max})\alpha \min(i, t)}}$$

$$= \frac{\beta_K(\delta_1, \delta_2)}{\sqrt{\min(i, t)}}\sqrt{\frac{1 + \mathbb{E}\left\|\eta_{i,t}\right\|^2}{\alpha\min(\pi_{\min}, 1 - \pi_{\max})}}$$

under event $E_K$ for all $(i, t) \in \mathcal{O}_{i,t}$. $\square$

### E.5 Proof of Theorem 1

We first decompose the regret

$$\sum_{k=1}^K \frac{1}{k} \sum_{(i,t)\in\mathcal{O}_k\backslash\mathcal{O}_{k-1}} \left[\pi^\star_{i,t}(\bar{A}^\star_{i,t}|S_{i,t})x(S_{i,t}, \bar{A}^\star_{i,t})^\top\theta^\star_{i,t} - \pi_{i,t}(\bar{A}_{i,t}|S_{i,t})x(S_{i,t}, \bar{A}_{i,t})^\top\theta^\star_{i,t}\right]$$

$$= \sum_{k=1}^K \frac{1}{k} \sum_{(i,t)\in\mathcal{O}_k\backslash\mathcal{O}_{k-1}} \Big[\{\pi^\star_{i,t}(\bar{A}^\star_{i,t}|S_{i,t}) - \pi_{i,t}(\bar{A}_{i,t}|S_{i,t})\}x(S_{i,t}, \bar{A}_{i,t})^\top\theta^\star_{i,t}$$

$$+ \pi^\star_{i,t}(\bar{A}^\star_{i,t}|S_{i,t})\left\{x(S_{i,t}, \bar{A}^\star_{i,t})^\top\theta^\star_{i,t} - x(S_{i,t}, \bar{A}_{i,t})^\top\theta^\star_{i,t}\right\}\Big]$$

$$\leq \sum_{k=1}^K \frac{1}{k} \sum_{(i,t)\in\mathcal{O}_k\backslash\mathcal{O}_{k-1}} \left[\{\pi^\star_{i,t}(\bar{A}^\star_{i,t}|S_{i,t}) - \pi_{i,t}(\bar{A}_{i,t}|S_{i,t})\}x(S_{i,t}, \bar{A}_{i,t})^\top\theta^\star_{i,t}\right]$$

$$+ \sum_{k=1}^K \frac{1}{k} \sum_{(i,t)\in\mathcal{O}_k\backslash\mathcal{O}_{k-1}} \left\{x(S_{i,t}, \bar{A}^\star_{i,t})^\top\theta^\star_{i,t} - x(S_{i,t}, \bar{A}_{i,t})^\top\theta^\star_{i,t}\right\}.$$

We can bound the (absolute value of) the first term using Lemma 13:

$$\sum_{k=1}^{K} \frac{1}{k} \sum_{(i,t) \in \mathcal{O}_k \setminus \mathcal{O}_{k-1}} \left[ \{\pi_{i,t}^{\star}(\bar{A}_{i,t}^{\star}|S_{i,t}) - \pi_{i,t}(\bar{A}_{i,t}|S_{i,t})\} x(S_{i,t}, \bar{A}_{i,t})^{\top} \theta_{i,t}^{\star} \right]$$

$$< \sum_{k=1}^{K} \frac{1}{k} \sum_{(i,t) \in \mathcal{O}_k \setminus \mathcal{O}_{k-1}} \frac{\tau \beta_K(\delta_1, \delta_2)}{\sqrt{\min(i,t)}}$$

$$< \tau \beta_K(\delta_1, \delta_2) \sum_{k=1}^{K} \frac{1}{k} \int_{u=0}^{k/2} \frac{2}{\sqrt{u}} du$$

$$= \tau \beta_K(\delta_1, \delta_2) \sum_{k=1}^{K} \frac{2\sqrt{2}}{\sqrt{k}}$$

$$< \tau \beta_K(\delta_1, \delta_2) 2\sqrt{2} \int_{u=0}^{K} \frac{1}{\sqrt{u}} du$$

$$= \tau \beta_K(\delta_1, \delta_2) 4\sqrt{2K},$$

under event $E_K$.

The proof for the second term follows closely from Abeille & Lazaric (2017) with several adjustments. Assumption 5 implies that we only need to consider the unit ball $\mathcal{X} = \{x \in \mathbb{R}^d : \|x\| \le 1\}$. Then the second term of the regret can be decomposed into

$$\underbrace{\sum_{k=1}^{K} \frac{1}{k} \sum_{(i,t) \in \mathcal{O}_k \setminus \mathcal{O}_{k-1}} \left( x_{i,t}^{\star\top} \theta_{i,t}^{\star} - x_{i,t}^{\top} \tilde{\theta}_{i,t} \right)}_{R^{TS}(K)} + \underbrace{\sum_{k=1}^{K} \frac{1}{k} \sum_{(i,t) \in \mathcal{O}_k \setminus \mathcal{O}_{k-1}} \left( x_{i,t}^{\top} \tilde{\theta}_{i,t} - x_{i,t}^{\top} \theta_{i,t}^{\star} \right)}_{R^{RLS}(K)}$$

where $x_{i,t}^{\star} = x(S_{i,t}, \bar{A}_{i,t}^{\star})$ and $x_{i,t} = x(S_{i,t}, \bar{A}_{i,t})$. The first term is the regret due to the random deviations caused by sampling $\tilde{\theta}_{i,t}$ and whether it provides sufficient useful information about the true parameter $\theta_{i,t}^{\star}$. The second term is the concentration of the sampled term around the true linear model for the advantage function.

**Bounding $R^{RLS}(T)$.** We decompose the second term into the variation of the point estimate and the variation of the random sample around the point estimate:

$$\sum_{k=1}^{K} \frac{1}{k} \sum_{(i,t) \in \mathcal{O}_k \setminus \mathcal{O}_{k-1}} \left( x_{i,t}^{\top} \tilde{\theta}_{i,t} - x_{i,t}^{\top} \check{\theta}_{i,t} \right) + \sum_{k=1}^{K} \frac{1}{k} \sum_{(i,t) \in \mathcal{O}_k \setminus \mathcal{O}_{k-1}} \left( x_{i,t}^{\top} \check{\theta}_{i,t} - x_{i,t}^{\top} \theta_{i,t}^{\star} \right)$$

The first term describes the deviation of the TS linear predictor from the RLS one, while the second term describes the deviation of the RLS linear predictor from the true linear predictor. The first term is controlled by the construction of the sampling distribution $\mathcal{D}^{TS}$, while the second term is controlled by the RLS estimate being a minimizer of the regularized cumulative squared error in (3). In particular, the first term will be small when the TS estimate concentrates around the RLS one, while the second will be small when the RLS estimate concentrates around the true parameter vector. Under event $E_K$, we can bound $R^{RLS}(K)$ by leveraging Lemma 6 and decomposing the error via

$$R^{RLS}(K) \le \sum_{k=1}^{K} \frac{1}{k} \left[ \sum_{(i,t) \in \mathcal{O}_k \setminus \mathcal{O}_{k-1}} |x_{i,t}^{\top}(\tilde{\theta}_{i,t} - \check{\theta}_{i,t})| \right] + \sum_{k=1}^{K} \frac{1}{k} \left[ \sum_{(i,t) \in \mathcal{O}_k \setminus \mathcal{O}_{k-1}} |x_{i,t}^{\top}(\check{\theta}_{i,t} - \theta_{i,t}^{\star})| \right].$$

Under event $E_K$, we have

$$|x_{i,t}^{\top}(\tilde{\theta}_{i,t} - \check{\theta}_{i,t})| \le \|x_{i,t}\|_{\underline{V}_{i,t}^{-1}} \gamma_K(\delta_1, \delta_2), \quad |x_{i,t}^{\top}(\check{\theta}_{i,t} - \theta_{i,t}^{\star})| \le \|x_{i,t}\|_{\underline{V}_{i,t}^{-1}} \beta_K(\delta_1, \delta_2),$$

where $\gamma_K(\delta_1, \delta_2) := \beta_K(\delta_1, \delta_2)\sqrt{cd\log\{K(K+1)c'd/\delta_1\}}$ and $\beta_K(\delta_1, \delta_2)$ is defined in Lemma 11. Then, from Assumption 9, we have

$$\sum_{k=1}^{K} \frac{1}{k} \sum_{(i,t)\in\mathcal{O}_k\backslash\mathcal{O}_{k-1}} |x_{i,t}^\top(\tilde{\theta}_{i,t} - \check{\theta}_{i,t})|$$

$$\leq \sum_{k=1}^{K} \frac{1}{k} \sum_{(i,t)\in\mathcal{O}_k\backslash\mathcal{O}_{k-1}} \|x_{i,t}\|_{\underline{V}_{i,t}^{-1}} \cdot \|\tilde{\theta}_{i,t} - \check{\theta}_{i,t}\|_{\underline{V}_{i,t}}$$

$$\leq \sum_{k=1}^{K} \frac{1}{k} \sum_{(i,t)\in\mathcal{O}_k\backslash\mathcal{O}_{k-1}} \|x_{i,t}\|_{\underline{V}_{i,t}^{-1}} \cdot \gamma_K(\delta_1, \delta_2)$$

$$= \gamma_K(\delta_1, \delta_2) \sum_{k=1}^{K} \frac{1}{k} \sum_{(i,t)\in\mathcal{O}_k\backslash\mathcal{O}_{k-1}} \|x_{i,t}\|_{\underline{V}_{i,t}^{-1}}$$

$$< \gamma_K(\delta_1, \delta_2) \sum_{k=1}^{K} \frac{1}{k} \sum_{(i,t)\in\mathcal{O}_k\backslash\mathcal{O}_{k-1}} \frac{1}{\sqrt{\alpha\min(i,t)}}$$

$$< \frac{\gamma_K(\delta_1, \delta_2)}{\sqrt{\alpha}} \sum_{k=1}^{K} \frac{1}{k} \int_{u=0}^{k/2} \frac{2}{\sqrt{u}} du$$

$$= \frac{\gamma_K(\delta_1, \delta_2)}{\sqrt{\alpha}} \sum_{k=1}^{K} 2\sqrt{\frac{2}{k}}$$

$$< \gamma_K(\delta_1, \delta_2) 2\sqrt{2/\alpha} \int_{u=0}^{K} \frac{1}{\sqrt{u}} du$$

$$= \gamma_K(\delta_1, \delta_2) 4\sqrt{2K/\alpha}$$

under event $E_K$. Using a similar derivation for the $\beta_K(\delta_1, \delta_2)$ case, the following inequality holds under event $E_K$:

$$R^{RLS}(K) < \{\beta_K(\delta_1, \delta_2) + \gamma_K(\delta_1, \delta_2)\} 4\sqrt{2K/\alpha}$$
$$= \beta_K(\delta_1, \delta_2)\left[1 + \sqrt{cd\log\{K(K+1)c'd/\delta_1\}}\right] 4\sqrt{2K/\alpha}$$

**Bounding $R^{TS}(T)$.** Leveraging Abeille & Lazaric (2017, pages 6–7), Definition 1 lets us bound $R^{TS}(K)$ under the event $E_K$:

$$R^{TS}(K) \leq \frac{4\gamma_K(\delta_1, \delta_2)}{p} \sum_{k=1}^{K} \frac{1}{k} \sum_{(i,t)\in\mathcal{O}_k\backslash\mathcal{O}_{k-1}} \mathbb{E}\left(\|\operatorname{argmax}_{x\in\mathcal{X}} x^\top\tilde{\theta}_{i,t}\|_{\underline{V}_{i,t}^{-1}}|\mathcal{F}_{i,t}\right).$$

We can bound the expectation under Assumption 9 as

$$\mathbb{E}\left(\|\operatorname{argmax}_{x\in\mathcal{X}} x^\top\tilde{\theta}_{i,t}\|_{\underline{V}_{i,t}^{-1}}|\mathcal{F}_{i,t}\right) < \frac{1}{\sqrt{\alpha\min(i,t)}}.$$

Using a derivation similar to that of $R^{RLS}(K)$, it then follows that

$$R^{TS}(K) < \frac{4\gamma_K(\delta_1, \delta_2)}{p} 4\sqrt{2K/\alpha}$$

under event $E_K$.

**Overall bound.** Composing the three bounds yields the following upper bound:

$$4\sqrt{2K}\left\{\beta_K(\delta_1, \delta_2)\left(\tau + \frac{1}{\sqrt{\alpha}}\right) + \frac{\gamma_K(\delta_1, \delta_2)}{\sqrt{\alpha}}\left(1 + \frac{4}{p}\right)\right\}$$

with probability at least $1 - \delta_1 - \delta_2$. With $\delta = 2\delta_1$ and sufficiently large $\zeta$, the bound holds with probability at least $1 - \delta$ as stated in Theorem 1. The rate in the main paper follows by noting that

$$\beta_K(\delta_1, \delta_2) = O\left\{\sqrt{d\log(K)} + \log^{3/4}(K)\right\},$$

$$\gamma_K(\delta_1, \delta_2) = \beta_K(\delta_1, \delta_2) \cdot O\left\{\sqrt{d\log(K^2 d)}\right\},$$

$$\frac{1}{\sqrt{\alpha}} = O(\sqrt{d}).$$

## F   Notation Guide

For convenience, we summarize below some key notation used in the main paper.

- $a = 0$: Control action
- $q$: Number of non-baseline treatment arms
- $i = 1, 2, \ldots$: Index for users
- $t = 1, 2, \ldots$: Index for time points
- $S_{i,t} \in \mathcal{S}$: Context vector observed for user $i$ at time point $t$
- $A_{i,t} \in \{0, \ldots, q\}$: Action chosen for user $i$ at time point $t$
- $R_{i,t} \in \mathbb{R}$: Reward observed for user $i$ at time point $t$
- $r_{i,t}(s, a) := \mathbb{E}[R_{i,t}|S_{i,t} = s, A_{i,t} = a]$: Conditional mean for the observed reward given the state and context
- $\Delta_{i,t}(s, a) := r_{i,t}(s, a) - r_{i,t}(s, 0)$: Linear differential reward for any action $a > 0$ and state $s$
- $\mathcal{H}_{i,t}$: History up to time point $t$ for user $i$
- $\pi_{i,t}(a|s)$: Probability of action $a \in [K]$ given context $s \in \mathcal{S}$ for a fixed (implicit) history
- $\pi_{\min}, \pi_{\max}$: Lower and upper limits for $\pi_{i,t}(0|s)$, respectively
- $A_{i,t}^\star \in [K]$: The optimal arm
- $\bar{A}_{i,t}^\star \in [K]\backslash\{0\}$: The optimal non-baseline arm
- $\bar{A}_{i,t} \in [K]\backslash\{0\}$: Potential non-baseline arm that may be chosen if the baseline arm is not chosen
- $\mathcal{O}_k = \{(i, t) : i \le k, t \le k + 1 - i\}$: The set of observed time points across all users at stage $k$ of the sequential requirement setting
- $x(s, a) \in \mathbb{R}^d$: Feature vector depending on the state and action
- $\theta_{i,t} \in \mathbb{R}^d$: Vector of parameters that may depend on the user $i$ and time point $t$. Later $\theta_{i,t}$ is written as $\theta_{i,t} = \theta^{\text{shared}} + \theta_i^{\text{user}} + \theta_t^{\text{time}}$, where $\theta_i^{\text{user}}$ is the user-specific but time-invariant term and $\theta_t^{\text{time}}$ is a shared time-specific term
- $g_t(s)$: Baseline reward function (conditional mean) that is observed when users are randomized to receive no treatment
- $\delta_{a>0}$: Indicator function that takes the value 1 if $a > 0$ and 0 otherwise
- $\epsilon_{i,t}$: Conditionally mean-zero, sub-Gaussian error term
- $f_{i,t}(s, a)$: Working model for the true conditional mean $r_{i,t}(s, a)$
- $\tilde{R}_{i,t}^f(s, \bar{a})$: Pseudo-reward given state $S_{i,t} = s$ and potential arm $\bar{a}$, which has the same expectation as the differential reward, $\Delta_{i,t}(s_{i,t}, \bar{a}_{i,t})$; this term is written as $\tilde{R}_{i,t}^f$ in some points in the main paper, with the state and action implied
- $W_j$: The $j$-th fold, of $J$ total folds for sampling splitting
- $\phi_{i,t}$: The enlarged feature vector with $x_{i,t}$ placed in positions corresponding to $\theta^{\text{shared}}, \theta_i^{\text{user}}$, and $\theta_t^{\text{time}}$: $\phi_{i,t} = \phi(x_{i,t}) = C_{i,t}^\top x_{i,t}$

- $\tilde{\sigma}_{i,t}^2 = \pi_{i,t}(0|s_{i,t}) \cdot \{1 - \pi_{i,t}(0|s_{i,t})\}$: Weights used in the penalized regression estimation, which are inversely proportional to $\text{Var}(\tilde{R}_{i,t}^f)$

- $C_{i,t}$: Matrix defined as $[1, \ 0_{i-1}^\top, \ 1, \ 0_{K-i+t-1}^\top, \ 1, \ 0_{K-t}^\top] \otimes I_d$

- $G_{\text{user}} = (V_{\text{user}}, E_{\text{user}})$, $G_{\text{time}} = (V_{\text{time}}, E_{\text{time}})$: Graphs for the nearest-neighbor networks across users and time, respectively

- Q: Incidence matrix where the element $Q_{v,e}$ corresponds to the $v$-th vertex (user) and $e$-th edge

- $L_{\text{user}}, L_{\text{time}}$: Laplacian matrices for users and time, respectively

- $\lambda$: Hyperparameter for Laplacian penalization

- $\gamma$: Hyperparameter for ridge penalization

- $\beta_{i,t}(\delta)$: Term used to scale confidence sets; see (4)

