# OpenReview forum: "RoME: A Robust Mixed-Effects Bandit Algorithm for Optimizing Mobile Health Interventions"
_NeurIPS.cc/2024/Conference — NeurIPS 2024 poster_

### Official Review · Reviewer_JBD9 · 2024-07-10

**Soundness:** 4
**Presentation:** 3
**Contribution:** 3
**Rating:** 7
**Confidence:** 4

**Summary:**

The paper introduces a robust contextual bandit algorithm to optimize personalized mobile health interventions. The proposed algorithm leverages Thompson sampling, mixed-effect models, debiased machine learning, and nearest-neighbor regularization techniques to address the problems of user and time heterogeneity, information pooling, and complex baseline rewards. The paper establishes a high-probability regret bound and demonstrates the algorithm’s effectiveness compared to existing methods through a simulation and two off-policy evaluation studies.

**Strengths:**

-	The paper proposes a method to effectively address three common issues in mobile health: treatment effect heterogeneity across users and time, the need to pool information across users, and the possibility of complex baseline rewards.
-	The simulation and real data study sufficiently demonstrate the advantages of the proposed method in the presence of heterogeneous users and nonlinear baseline rewards, and discuss its limitations when users are homogeneous.
-	The paper is well-organized and easy to follow.

**Weaknesses:**

The proposed method combines several existing techniques, including Thompson sampling, mixed-effect models, debiased machine learning, and nearest-neighbor regularization. Could the authors elaborate on the specific challenges associated with the proposed method, whether in terms of methodology, computation, or theoretical proof?

**Questions:**

-	It is mentioned in the appendix that the nearest neighbor network is assumed to be known in the simulation study, and that the number of nearest neighbors k is set to 5 in the Valentine Study. Could the authors provide details on how the nearest neighbor network is defined in the simulation study and how the number of nearest neighbors is chosen in the Valentine Study? Is the algorithm's performance sensitive to the construction of the nearest neighbor network?
-	The paper lists several feasible choices for the working model f_{i,t}. However, its construction in the simulation and real data analysis was not detailed. It would be helpful if the authors could discuss how the working model is chosen in practice.
-	Assumption 1 ensures that the randomization probability for action 0 is always positive, but it does not specify this for other actions. Is a positivity assumption required for all actions in the proposed method? If so, the authors should discuss how this is ensured for other actions.
-	Computational efficiency is crucial for online algorithms. It would be beneficial if the authors could provide the computation time of the proposed method, e.g., in the simulation study or the Valentine Study. How does it compare with the baseline methods?
-	In the single-column vector \theta, user-specific parameters are numbered from 1 to K. Should they instead be numbered from 1 to N, as in the \Theta_{user} matrix? Besides, it seems that N has not been defined in the main text.

**Limitations:**

The authors have discussed the limitations of the proposed method.

---

> ### Author Rebuttal · Authors · 2024-08-06
>
> Thank you for the thoughtful review. We address the listed weaknesses and questions below.
>
> # Weaknesses
>
> Yes, there are three primary challenges. We will include these descriptions below in the final revision.
>
> The first is the methodological challenge of figuring out how to bring all of the pieces together. DML, for instance, requires independent subsets of data, so we had to develop a data-splitting technique that is appropriate for dynamic settings. As another example, NNR is typically applied to distinct users in stationary settings, so we had to adapt the technique to handle nonlinear time trends.
>
> The second challenge is extending the standard proof techniques for bandit regret bounds to handle (a) the nonlinear baseline reward and Graph Laplacian regularization and (b) our method’s growing number of parameters. For (a), we need to adapt the standard proof bounding the regularized least squares (RLS) linear predictor error to handle the use of weighted least squares, the non-linear baseline and Laplacian regularization (our Lemma 6). This requires careful analysis of how the RLS estimator relates to the double robustness of the pseudo-reward and bounding the variance of the pseudo-reward. For (b), our final regret bound involves a sum of such regularized least squares linear predictor errors, which is bounded by a sum of weighted norms of feature vectors inverse scaled by stages. Bounding this requires a simple but non-obvious application of Cauchy-Schwarz to decouple the weighted feature norms from inverse stage scaling, which allows us to apply standard techniques to the sum of norms and bound the sum of inverse stages by the harmonic number.
>
> The third challenge was creating an efficient implementation. Re-computing inverses and log determinants at every stage would have been prohibitively slow, so we had to experiment with efficient rank-one update procedures (implemented via the SuiteSparse package).
>
> # Question 1: Network
>
> Our main rebuttal contains a discussion of these points, including some recently added sensitivity analyses. To summarize, the quality of the network and number of neighbors does influence the performance, but these differences are small relative to the differences between methods.
>
> To form the network, we created a distance metric using four baseline covariates: gender, age, device, and baseline average daily steps. Then we connected users to their 5 closest neighbors. We had to move this information to the appendix due to space restrictions.
>
> # Question 2: Supervised Model Selection
>
> Because the simulation required many repetitions over three settings and eight methods (including those in the Appendix), our goal was to find a nonlinear regression method that had an easy-to-use and computationally efficient online implementation. The River library contains several regression methods that meet these criteria. We chose an ensemble of decision trees (essentially a random forest) because we have found that random forests typically perform well with limited tuning.
>
> We also used a random forest model for the case study, largely for the reason stated above. However, the computational demands were lower, so we were actually able to refit it at each stage using standard (not online) decision trees.
>
> In general, we recommend using nonlinear regression methods. When computation time is not a large concern, we recommend choosing a method with high predictive accuracy for the problem, which could be assessed using previous studies or in an adaptive fashion via cross validation. When computation time is a major concern, we recommend selecting methods that have efficient online updates. In particular, linear models with flexible basis functions (e.g., splines) are particularly well suited because they have efficient rank-one updates.
>
> # Question 3: Assumption 1
>
> This is somewhat of a subtle point. We sample actions in a hierarchical fashion. We first select a non-baseline action and then we randomize between that action and the baseline (“do-nothing”) action. The regret bound is based only on the second randomization. It bounds the regret of a policy that is restricted to two actions: the baseline action and the given non-baseline arm. Consequently, the $K$ appearing in the regret bound is really the number of stages in which the given non-baseline arm is selected. Our analysis mimics that of the action-centered (AC) paper in this regard (Greenewald et al., 2017).
>
> To summarize, we do not list a positivity assumption for other actions because we define and bound the regret in a way that restricts the randomization to a single non-baseline arm. In practice, however, introducing more arms will generally lead to higher overall regret (with a fixed total study length) because less data will be available for each arm.
>
> # Question 4: Computation Time
>
> Thank you for the good suggestion. We reported these timings in the main rebuttal.
>
> # Question 5: $N$ vs. $K$
>
> Thank you for catching this. In our asymptotic regime, $N = K$, so either is correct. For consistency and simplicity, however, we have replaced the $N$’s with $K$’s in the main text. The algorithm could still be applied under a more realistic regime in which $N \neq K$, but the matrices would need to be resized based on the number of users ($N$) and time points ($T$).

---

> > ### Comment · Reviewer_JBD9 · 2024-08-12
> >
> > Thank you for your thoughtful and detailed response. I particularly appreciate the in-depth discussion of the technical challenges, which I found highly informative. Accordingly, I have adjusted my rating upward.

---

> > > ### Author Response · Authors · 2024-08-12
> > >
> > > Thank you once again for your insightful comments—we really appreciate your support! We will be sure to incorporate these discussions into our revised manuscript.

---

### Official Review · Reviewer_HodX · 2024-07-12

**Soundness:** 2
**Presentation:** 3
**Contribution:** 2
**Rating:** 7
**Confidence:** 5

**Summary:**

The paper "A Robust Mixed-Effects Bandit Algorithm for Assessing Mobile Health Interventions" introduces the DML-TS-NNR (Debiased Machine Learning Thompson Sampling with Nearest-Neighbor Regularization) algorithm. This novel contextual bandit algorithm is designed to address challenges in mobile health (mHealth) interventions, such as participant heterogeneity, nonstationarity, and nonlinearity in rewards. The algorithm incorporates user- and time-specific incidental parameters, network cohesion penalties, and debiased machine learning to flexibly estimate baseline rewards.

**Strengths:**

### Originality:

The DML-TS-NNR algorithm introduces a novel approach by combining debiased machine learning, network cohesion penalties, and Thompson sampling to address the unique challenges in mHealth interventions. The use of user- and time-specific incidental parameters and the flexible estimation of baseline rewards via debiased machine learning are innovative contributions that enhance the adaptability and robustness of the algorithm.

### Quality:

The methodology is rigorously developed, with comprehensive theoretical analysis and detailed proofs provided for the high-probability regret bounds. Extensive experimental validation includes simulations and real-world mHealth studies, demonstrating the algorithm's effectiveness and practical viability.

### Clarity:

The paper is well-structured, with clear explanations of the problem statement, related work, methodology, experimental setup, and results. Figures and tables effectively illustrate the proposed method and its performance improvements. Technical terms and concepts are explained thoroughly, ensuring accessibility to a broad audience, including those not deeply familiar with contextual bandit algorithms or mHealth interventions.

### Significance:

By addressing critical challenges in mHealth, such as participant heterogeneity and nonstationarity in rewards, the DML-TS-NNR algorithm has significant implications for improving personalized treatment strategies and patient outcomes.

**Weaknesses:**

### Novelty:

While the combination of debiased machine learning, network cohesion penalties, and Thompson sampling is innovative, a more detailed comparison with existing methods, particularly in terms of theoretical and practical advantages, would further highlight the unique contributions of the proposed DML-TS-NNR algorithm.


### Experimental Validation:

The experimental validation primarily relies on controlled datasets. Including more diverse real-world testing scenarios, such as various health conditions or treatment modalities, would provide a more comprehensive assessment of the DML-TS-NNR algorithm's effectiveness and generalizability.


### Technical Details:

Some aspects of the algorithm, such as the optimization process for debiased machine learning and the derivation of the network cohesion penalties, could be explained in greater detail to enhance clarity and understanding. The choice of evaluation metrics and their suitability for different types of mHealth interventions could be discussed more extensively.

**Questions:**

1. Can the authors elaborate on why specific baseline methods (e.g., Standard Thompson Sampling, Action-Centered contextual bandit) were chosen for comparison? What advantages do these techniques offer for evaluating the effectiveness of the DML-TS-NNR algorithm in mHealth interventions?

2. How does the DML-TS-NNR algorithm perform in more diverse and dynamic real-world settings, such as different health conditions or treatment modalities? Are there plans to test the approach in more varied environments, including chronic diseases or mental health applications?

3. What potential challenges could arise regarding the scalability of the DML-TS-NNR algorithm for very large datasets or real-time applications? How does the algorithm handle computational and communication overhead in such scenarios?

4. Can the authors provide more details on the optimization process for debiased machine learning and its impact on overall performance and accuracy of reward estimation?

5. What are the potential ethical considerations or privacy concerns associated with the proposed method, particularly in using sensitive health data for mHealth interventions? How does the DML-TS-NNR algorithm address these concerns?

**Limitations:**

1. Potential challenges and mitigation strategies for deploying the DML-TS-NNR algorithm in real-world settings, particularly regarding data diversity and environmental variability.
2. Discussing the ethical implications and privacy concerns related to continuous monitoring and intervention in high-stakes healthcare applications. This includes addressing the implications of using sensitive patient data and the potential impact of intervention decisions on different demographic groups.
3. Further exploring the trade-offs between computational efficiency and intervention accuracy, particularly in scenarios where rapid decision-making is crucial for clinical outcomes.

---

> ### Author Rebuttal · Authors · 2024-08-06
>
> Thank you for your insightful comments and questions. We address the latter below.
>
> # Q 1
>
> We included a wide variety of competing algorithms (see appendix for the full set) to (1) assess how well DML-TS-NNR performs relative to simple baselines and (2) understand which aspects of DML-TS-NNR contribute most to its performance.
>
> 1. The comparison to Standard and AC shows that there is a large benefit to pooling data across users.
> 2. The comparison to NNR-Linear shows that the benefit of DML-TS-NNR is not entirely due to the additional regularization (though, it is helpful).
> 3. The comparison to DML-TS-SU shows that ignoring participant heterogeneity leads to suboptimal performance.
> 4. The comparison to DML-TS-NNR-BLM (which uses bagged linear models instead of an ensemble of online decision trees) shows that DML-TS-NNR is robust to the choice of supervised learning algorithm but that a more flexible algorithm can lead to better performance when the baseline rewards are nonlinear.
> 5. The comparison to IntelPooling, the most relevant previous method, shows that there is additional benefit to the new components of our method: NNR and DML (IntelPooling uses neither).
> 6. The comparison to Neural-Linear shows that the performance of DML-TS-NNR is not entirely (or even largely) due to the flexible baseline model; the time effects and NNR also play an important role.
>
> # Q 2
>
> The appendix includes an additional off-policy evaluation study for the Intern Health Study (IHS). IHS is a mental health study for medical interns. Similar to the case study in the main paper, we again found that DML-TS-NNR outperformed competing algorithms.
>
> Thus, to answer your question, we have applied DML-TS-NNR to two different areas already and DML-TS-NNR achieved state-of-the-art performance in each. We are involved in applied collaborations in multiple sub-areas of medical and behavioral science and plan to use DML-TS-NNR across these domains. While we do not foresee any particular difficulties for specific domains, we expect that these collaborations will lead to an improved understanding of when DML-TS-NNR performs particularly well (or poorly) and how to address practical issues, such as specification of the context and reward variables.
>
> # Q 3
>
> Our main rebuttal addresses this point. In its current form, we think the algorithm could be used without modification/approximation up to about 1,000 stages—much more than most academic mobile health studies. At that point, some of the strategies suggested in the main rebuttal could be used to enable scaling to even larger data sets.
>
> Real-time implementation of the algorithm across many mobile devices would require communication between each user’s device and a central server. While this carries numerous engineering difficulties, the methodological difficulties are fairly limited. The system could be set up such that devices send information to the server each time a decision is required, which would require essentially no methodological changes. Alternatively, the algorithm parameters ($V, b$ in particular) could be updated in batches, and the mobile devices would make decisions using the most up-to-date cached parameter values available to them. In this case, the mechanics of the algorithm would remain essentially unchanged; however, the regret bound would need to be adapted to account for the batched updates.
>
> # Q 4
>
> One of the benefits of the DML-TS-NNR algorithm is that it remains agnostic to the choice of the specific supervised learning model and its corresponding optimization algorithm. The only theoretical requirement is the consistency condition of Assumption 3. In fact, as pointed out in our response to reviewer cFGH, we could still obtain a regret bound of the same asymptotic order as that given in Theorem 1 (albeit with larger constants) under violations of Algorithm 3, provided the supervised learning model converges to a bounded function.
>
> That said, we generally expect highly predictive supervised learning models to produce the lowest regret. Consequently, it would be reasonable to select a supervised learning algorithm via standard cross validation techniques (though, adaptive strategies like this are not necessarily compatible with our regret bound).
>
> We also note that our additional simulations in Appendix B include a comparison of two supervised learning algorithms: an ensemble of online decision trees (used in the main paper) and a bagged ensemble of linear models. The former performed slightly better in the Nonlinear Setting (beating DML-TS-NNR in 30/50 simulations), presumably because it is flexible enough to model the nonlinear baseline rewards.
>
> # Q 5
>
> To address these privacy considerations, we would need to ensure that data is transmitted from mobile devices to the server in a secure fashion. We would also need to ensure that the server meets the relevant requirements (e.g., HIPAA in the United States) for collecting, storing, and processing that data. The data transmission from the server to the devices would be less of a concern because the data is aggregated (in fact, we would not need to send the parameters for user $i$ to user $j$ [with $i \neq j$]).
>
> Regarding the treatment of different demographic groups, we believe the personalized, network-based nature of DML-TS-NNR has the potential to ameliorate these concerns. Allowing users to have their own parameters means that decisions will be directly adapted to users’ own actions—not the conglomerate of some larger population. If practitioners believe that treatment effects will vary substantially by demographic group, then demographic information could be used to design the network structure. The result would be that the algorithm would suggest treatments that are effective for “similar” (based on the network) users rather than treatments that are effective for the “average user,” particularly in the early stages for a given user when subject-specific data is minimal.

---

> > ### Comment · Reviewer_HodX · 2024-08-11
> >
> > Thank you for your responses. I currently don't have any additional questions. I will make my final decision after further discussions with the other reviewers and the AC.

---

> > > ### Author Response · Authors · 2024-08-11
> > >
> > > Thank you very much for your feedback; we will be sure to incorporate your comments into the revised manuscript.

---

### Official Review · Reviewer_SKyx · 2024-07-13

**Soundness:** 3
**Presentation:** 2
**Contribution:** 2
**Rating:** 3
**Confidence:** 4

**Summary:**

The authors propose a novel contextual bandit algorithm that addresses individual heterogeneity, nonstationarity, and nonlinearity of the reward function. This algorithm involves three distinct steps to manage these challenges.

**Strengths:**

The current paper is easy to follow and addresses a complicated scenario that has not been frequently discussed in the literature.

The methods are well organized and the theoretical discussions are helpful.

**Weaknesses:**

The selection of tuning parameters plays a crucial role but is not discussed in the proposed method. It is unclear whether the promising performance of the proposed algorithm is due to the choice of a specific tuning parameter. Providing some robustness checks would enhance the soundness of the simulation experiments.

I also need some motivation for Assumption 4. For example, how well does the dataset under investigation support this assumption? In its current form, Assumption 4 appears to be arbitrary.

I also found the assumption of a known network structure to be quite restrictive, as network structures are typically unknown in practice. This limitation hinders the practicality of the proposed algorithm.

**Questions:**

I found the comments on page 4 about the doubly robustness concerning. As the methods discussed in Section 4.1 are all nonparametric estimators (hence, the model misspecification is not an issue), the benefit of incorporating DML is to improve estimation efficiency.

**Limitations:**

not noted.

---

> ### Author Rebuttal · Authors · 2024-08-07
>
> Thank you for your helpful comments. We address the outlined weaknesses and questions below.
>
> # Hyperparameter Tuning
>
> Hyperparameter selection plays an important role in the performance of bandit algorithms. While virtually all such algorithms require specification of some hyperparameters (at a minimum, a “prior” mean and variance), DML-TS-NNR does require more than a basic parametric bandit algorithm (e.g., the Standard method in our simulations). As detailed in the main rebuttal above, we added 8 sensitivity analyses to assess the impact of our method’s hyperparameters on its performance. The main finding is that DML-TS-NNR consistently outperforms the other methods in the Nonlinear Setting (the most realistic) even when we specify poor hyperparameters. The reason for the superior performance is that DML-TS-NNR is the **only one that accounts for all of the important problem structure**, including heterogeneity across users and time, explainable variation in baseline rewards, and network information. These results suggest that correctly accounting for the problem structure is relatively more important than hyperparameter tuning. This is analogous to the fact that out-of-the-box ML models often outperform tuned linear models (e.g., ridge regression / LASSO) on nonlinear regression tasks.
>
> We also re-emphasize that there are adaptive strategies that could be explored for setting many of the hyperparameters. The primary challenge with these strategies is that the regret bound is not guaranteed to apply when the hyperparameters are not specified in advance. This is an interesting and important avenue for future research.
>
> # Assumption 4
>
> Assumption 4 is best understood by comparing it to corresponding assumptions made by competing approaches. We provide this context below, beginning with simple bandit algorithms:
>
> - **Standard / AC:** Were we to apply Standard / AC to the mHealth setting, we would effectively be assuming that $\theta_{i,t} = \theta^{\text{shared}}$ for all $i, t$. This assumption is clearly very limiting and partially explains the poor performance of these methods in the simulation / case studies.
> - **NNR-Linear:** Graph bandit approaches, such as the NNR-Linear method from our simulation, allow all users to have their own parameters, effectively assuming $\theta_{i,t} = \theta_{i}^{\text{user}}$. By regularizing these parameters toward each other and zero (i.e., including penalties analogous to $\lambda$ and $\gamma$, respectively), we are essentially expressing a prior belief that the $\theta_{i}^{\text{user}}$ values are small and similar between neighbors. A slight improvement to this approach is to set $\theta_{i,t} = \theta^{\text{shared}} + \theta_{i}^{\text{user}}$ so that estimates are regularized toward the (learned) shared parameter instead of zero. These graph bandit approaches are much more effective than the Standard / AC approach in heterogeneous settings; however, they still assume constant effects over time, which is why NNR-Linear is not competitive in our Nonlinear simulation setting.
> - **Mixed Model Approaches:** A common solution to this problem in mixed modeling is to assume a random slopes model: $\theta_{i,t} = \theta_{i0}^{\text{user}} + t \cdot \theta_{i1}^{\text{user}}$; i.e., we assume a subject-specific linear trend over time. This assumption improves on the static assumption of graph bandit approaches, but it is still highly parametric and likely to be misspecified.
>
> With this information as background, we can see that Assumption 4 improves on simpler (and commonly used) alternatives by allowing for variation across both users and time. Unlike the random slopes model, we allow for non-parametric trends over time by including a separate parameter for all members of the (infinitely increasing) set of time points. We also note that IntelPooling employs a version of Assumption 4 by including both user- and time-specific random effects.
>
> We conducted an exploratory analysis of the Valentine Study to motivate the user-level and temporal effect heterogeneity. For this analysis, we formed pseudo-rewards according to Equation (3) and conducted an ANOVA test. The results (Table 2 of the rebuttal PDF) show strong evidence of both forms of heterogeneity. Figure 1 in the rebuttal PDF shows a Loess fit of the causal effects over time, indicating strong nonstationarity in the causal effects.
>
> # Network Structure Assumptions
> See our main rebuttal for a discussion of this point. To the best of our knowledge, all work in this literature, including Cesa-Bianchi et al. 2013, Herbster et al 2021, Vaswani et al. 2017, Choi et al. 2023 and many other papers use the same assumption of a known network structure. In many cases where contextual bandits are useful, such as in social networks, there are known relationships between users.
>
> That said, learning the network structure is a potentially important avenue for future work: thank you for mentioning this.
>
> # Double Robustness and DML
>
> We see the point being raised. The regret bound does indeed require the ML model to be a consistent estimator of $f$ (Assumption 3); hence, a nonparametric estimator of $f$ is needed in general for the regret bound to apply.
>
> In the proof of the regret bound, however, the only impact of consistently estimating $f$ is that it reduces the size of certain confidence sets, resulting in a tighter regret bound. As long as $f$ converges to a bounded function, DML-TS-NNR still produces a regret bound with the same asymptotic order, but the constants in the regret bound could be improved by consistently estimating $f$. Thus, the robustness to misspecification of $f$ in the pseudo-reward (lines 136 – 137) does have important theoretical implications when Assumption 3 fails. It is for this reason that we listed parametric models in lines 151-152. This is a subtle point that we could briefly explain in the final version of the paper.

---

### Official Review · Reviewer_cFGH · 2024-07-13

**Soundness:** 4
**Presentation:** 3
**Contribution:** 4
**Rating:** 7
**Confidence:** 4

**Summary:**

The paper introduces a novel robust mixed-effects bandit algorithm, named "DML-TS-NNR", designed to optimize mobile health (mHealth) interventions. mHealth aims to deliver personalized and contextually tailored notifications to promote healthier behaviors. The proposed algorithm addresses key challenges in mHealth, such as participant heterogeneity, nonstationarity, and nonlinearity in rewards. The main contributions of the paper are:
- Modeling Differential Rewards: Incorporates user- and time-specific parameters.
- Network Cohesion Penalties: Uses penalties to pool information across users and time.
- Debiased Machine Learning: Employs this technique for flexible baseline reward estimation.

The algorithm's high-probability regret bound is solely dependent on the differential reward model's dimension. The effectiveness of DML-TS-NNR is demonstrated through simulations and two off-policy evaluation studies.

**Strengths:**

- The integration of user- and time-specific incidental parameters for modeling differential rewards.
- The novel application of network cohesion penalties and debiased machine learning in the context of mHealth interventions.
- The empirical validation through simulations and real-world mHealth studies demonstrates the practical applicability and effectiveness of the proposed algorithm.
 - The paper is well-structured and clearly explains the problem, the proposed solution, and the results.
 - The algorithm addresses significant challenges in mHealth, potentially improving the effectiveness of personalized health interventions. By achieving robust regret bounds and superior empirical performance, the algorithm can contribute to advancements in personalized healthcare technologies.

**Weaknesses:**

- The algorithm's reliance on complex calculations, such as log-determinants and matrix inverses, may limit its scalability for large datasets.
- The assumption of a known network with binary edges may not always be practical. Real-world networks can be more complex, and this assumption might limit the algorithm's applicability.

**Questions:**

- How robust is the algorithm to violations of the assumptions regarding the network structure and hyperparameters? Can the authors provide guidance on tuning these parameters in practice?
- How can the algorithm be extended to consider long-term effects and treatment fatigue? Are there plans to address these aspects in future work?

**Limitations:**

While the authors have made strides in addressing some key limitations, such as computational demand and network assumptions, there are still areas that may need further attention:
- The high computational demand due to log-determinants and matrix inverses is a limitation. Future work should explore more efficient computational methods or approximations.
- The algorithm focuses on immediate rewards and does not consider long-term effects or treatment fatigue. Addressing these aspects is crucial for real-world applicability and effectiveness.
- The need for correctly specified hyperparameters can be challenging in practice. Providing more robust methods for hyperparameter tuning and addressing potential misspecifications would improve the algorithm's robustness and ease of use.

---

> ### Author Rebuttal · Authors · 2024-08-06
>
> # Weaknesses & Hyperparameters
>
> Thank you for raising these concerns. Other reviewers asked about them as well. Our main rebuttal contains a detailed discussion of all three. Below, we include a brief summary of the main points and a few additional details for your specific questions:
>
> 1. **Computation:** We made our implementation extremely efficient using sparse rank-one updates to Cholesky factors. In its current form, DML-TS-NNR is about three orders of magnitude faster than is required for a typical mHealth research study. Several optimizations and approximations could be used to speed up the algorithm even more for large data sets.
> 2. **Known Network:** This limitation is shared with all or nearly all of the related work in graph bandits. Our new sensitivity analysis shows that the algorithm is fairly robust to specification of the network. The algorithm can easily be extended to include edge weights; the only required change is scaling the corresponding entries in the Laplacian matrix.
> 3. **Hyperparameters:** We added eight sensitivity analyses showing that the simulation results are robust to variations in the hyperparameters. While hyperparameters do affect the performance of DML-TS-NNR, the differences between methods are much larger than the differences due to changing the hyperparameters. Empirical Bayes strategies can be used to set the hyperparameters. The penalty hyperparameters ($\lambda, \gamma$) can also be interpreted as prior precisions (inverse variances), which can provide some intuition on an appropriate order of magnitude.
>
> # Long-term Treatment Effects
>
> We see two main approaches to address long-term effects.
>
> The first approach is specifically designed to address treatment fatigue. It involves modifying the observed rewards by subtracting a penalty from rewards at treated time points (i.e., when $A_{i,t} > 0$). Or, equivalently, we could modify the algorithm such that individuals are treated with probability equal to the “posterior” probability that the differential reward is above a positive threshold as opposed to zero; see Equation (10). The threshold could be set based on a moderation analysis using the weighted, centered least squares method of Boruvka et al., 2018 [2]. Specifically, we would estimate treatment moderation according to a model including a linear term for the number of previous treatment occasions (perhaps over some moving window, such as the last 10 time points). Then we would set the threshold equal to the corresponding estimate for this linear moderation term, effectively anticipating the impact of current treatment on future outcomes.
>
> The second approach would be to generalize some of the ideas from this paper to more flexible RL methods, such as Q-learning. These methods directly model long-term effects, but uncertainty quantification is generally more challenging relative to bandits and may require some form of Markovian assumption, which would likely be unrealistic in mobile health.
>
> We do not have immediate plans to pursue these directions. However, we are actively involved in designing and analyzing mobile health studies, and we plan to use DML-TS-NNR to optimize online decision making in these studies. If we observe empirically that DML-TS-NNR consistently sacrifices long-term performance for small short-term improvements, then we will revisit this problem and the approaches outlined above.
>
> [2] Boruvka, A., Almirall, D., Witkiewitz, K., & Murphy, S. A. (2018). Assessing time-varying causal effect moderation in mobile health. Journal of the American Statistical Association, 113(523), 1112-1121.

---

### Author Rebuttal · Authors · 2024-08-06

We thank the reviewers for their helpful and positive comments, including “the paper is well-structured and clearly explains the problem, the proposed solution, and the results” and “The methodology is rigorously developed, with comprehensive theoretical analysis and detailed proofs.” Here we address points made by several reviewers, and we provide individualized responses to comments that are unique to specific reviewers.

# Hyperparameter Tuning & Robustness

In the paper’s main simulation study, we chose hyperparameters for the fairest comparison possible, employing the same regularization parameters across methods. To assess robustness, we recently added 8 sensitivity analyses that alter hyperparameters for our methods while fixing those of the other methods; this approach gives the competing algorithms an (arguably unfair) advantage. Our proposed algorithm, DML-TS-NNR (and the variant DML-TS-NNR-BLM) still dramatically outperform the other methods in the Nonlinear Setting. We provide detailed results below.

Three analyses led to no meaningful changes to Figure 1 or Table 2. These involved rescaling (1) $\gamma$, (2) $\lambda$, (3) and the bounds $B, D$ by a factor of 10: . The remaining simulations resulted in minor performance differences, especially in the Heterogenous Setting. Below, we summarize these five analyses:

- **Adding (low and medium) noise to the network:** NNR-Linear slightly outperforms our methods in the Heterogeneous Setting, winning 50-60% of simulations (compared to 40-42% without noise) because NNR-Linear has access to a higher quality network. Table 1 of the attached PDF displays pairwise comparisons for the medium noise level.
- **Increasing to 10 neighbors:** DML-TS-NNR, DML-TS-NNR-BLM perform much better than DML-TS-SU (the Single-User ablation) in the Heterogeneous Setting, presumably because this change enforced stronger network cohesion among highly similar users.
- **Rescaling $\sigma$ by 10:** NNR-Linear outperforms our methods in the Heterogeneous setting. NNR-Linear and IntelPooling outperform DML-TS-SU (but not DML-TS-NNR) in the Nonlinear Setting. Both occur because NNR-Linear and IntelPooling use the true value of $\sigma$.
- **Setting delta to 0.05:** NNR-Linear outperforms DML-TS-SU in the Nonlinear Setting, likely because this change led to insufficient exploration; see line 220 and Equation (8).

A heuristic strategy to set $\lambda, \gamma, \sigma$ is to use an empirical Bayes approach like that of the IntelPooling paper (Tomkins et al., 2021). Similarly, we could adaptively set the bounds $B, D$ using the parameter estimates. Deriving a regret bound under these techniques is an interesting future challenge.

# Scalability

Several reviewers expressed concern about computational efficiency when computing matrix inverses and log-determinants over a large parameter space. As suggested by Reviewer JBD9, we timed the algorithms in the simulation study. The average wall times (in seconds) in the Nonlinear setting were as follows (in increasing order):
- **AC:** 1.45
- **Standard:** 4.90
- **DML-TS-NNR-BLM (Bagged Linear Models):** 52.46
- **IntelPooling:** 66.15
- **NNR-Linear:** 71.39
- **DML-TS-SU:** 403.69
- **DML-TS-NNR (online decision trees):** 452.57
- **Neural-Linear:** 3013.04

As the simulation involved 200 stages, the timings produced 200 * 201 / 2 = 20,100 decisions for each method. Neural-Linear was the most time consuming because it requires neural network predictions for each reward (without a GPU). As expected, our methods were generally slower than the competitors. However, the difference between DML-TS-NNR-BLM and DML-TS-NNR indicates that much of the additional computation time is due to ML model fitting. From a computational standpoint, the reason that our method remained competitive is that (1) V is sparse and (2) we used efficient rank-one updates to Cholesky factors to avoid recomputing large matrix inverses and determinants. Real scientific mHealth applications typically require fewer than 1,000 decisions per day, so the computation time above is $\frac{20,100 / 452.57}{1,000 / (24 * 60 * 60)} \approx 3,837$ times faster than necessary.

In a large-data setting, the performance could be further optimized by:
- Updating $V, b$ in batches
- Parallelization
- Taylor approximations for the matrix inverse (see Yang et al. 2020)
- Fast approximate log-determinant computation [1]
- Forming networks with “closed” clusters, which would allow us to compute the inverse and determinant in blocks

# Known Network Assumption

Assuming a known network can indeed be limiting in practice. However, this limitation is shared with all or nearly all of the graph bandit literature, including Cesa-Bianchi et al. (2013), Vaswani et al. (2017), and Yang et al. (2020).

We assessed the importance of this assumption in our sensitivity analyses by adding noise in the network construction. We originally set the network structure by connecting users / time points to the 5 other users / time points with the most similar parameter values. In the sensitivity analyses, we added artificial noise to the parameter values before network construction, resulting in a lower quality network. As explained above, this modification resulted in only a slight decrease in performance.

In some cases, as in the Intern Health Study in our Appendix, there is an a priori known network structure. In other cases, it may be reasonable to propose a distance metric using baseline covariates (or proximity of time points). An interesting future direction would be to set the network in an adaptive fashion, perhaps using some form of sample splitting. Again, a primary challenge with an adaptive strategy like this is showing that the regret bound (or a modified version of it) is still valid.

[1] Boutsidis, Christos, et al. "A randomized algorithm for approximating the log determinant of a symmetric positive definite matrix." Linear Algebra and its Applications 533 (2017): 95-117.

---

### Decision · Program_Chairs · 2024-09-25

**Decision:**

Accept (poster)

**Comment:**

The paper proposes a new contextual bandit algorithm, namely DML-TS-NNR, for mHealth which can effectively incorporate the nonstationarity, nonlinearity and user heterogeneity of the health outcomes.
DML-TS-NNR allows the baseline outcome (outcome under "no action") to be highly complex and imposes a parametric structure only to the differential reward. The differential reward model parameter includes user specific and time specific parameters, with minimum assumptions that parameters of proximal users (according to a known network) and proximal time are similar. Laplacian regularization is employed to efficiently pool information across users and time. Finally, debiased machine learning methods are employed in the construction of a doubly robust estimate of the differential reward, which reduces the variance of existing IPW estimates. The paper also provides a regret upper bound which solely depends on the dimension of the differential reward model.

The paper is well-motivated from mHealth applications and provides a novel algorithm that is robust to the frequently occurring challenges such as nonstationarity, heterogeneity, and complexity of baseline rewards. These points are well supported by theory and extensive experiments.

Some concerns raised by reviewers were as follows,
- robustness to hyperparameters
- scalability
- the fact that the network structure should be known.

As for the first point, authors have provided in their rebuttal an extensive sensitivity analysis to show the robustness of the algorithm to the hyperparameter values, and also suggested to use the Empirical Bayes method to set the hyperparameters in practice.

As for the second point, authors provided the exact computation time in their rebuttal, showing that the speed is competitive to other existing methods and is fast enough in mHealth scenarios.

As for the third point, as authors claimed, it is true that there exist cases where the network structure is known, and most existing works exploiting the network structure assume the knowledge of the network. Authors also provided some experiments for the case where the network is not known perfectly and showed robustness of their method.